# Geometric Imbalance in Semi-Supervised Node Classification

**Liang Yan**[1], **Shengzhong Zhang**[1], **Bisheng Li**[1], **Menglin Yang**[4], **Chen Yang**[1],
**Min Zhou**[5], **Weiyang Ding**[1], **Yutong Xie**[3,*] **Zengfeng Huang**[1,2*]
[1]Fudan University     [2]Shanghai Innovation Institute     [3]MBZUAI
[4]Hong Kong University of Science and Technology (Guangzhou)     [5]Logs AI

Project Page: https://divinyan.com/UNREAL
Code: https://github.com/yanliang3612/UNREAL

## Abstract

Class imbalance in graph data presents a significant challenge for effective node classification, particularly in semi-supervised scenarios. In this work, we formally introduce the concept of geometric imbalance, which captures how message passing on class-imbalanced graphs leads to geometric ambiguity among minority-class nodes in the riemannian manifold embedding space. We provide a rigorous theoretical analysis of geometric imbalance on the riemannian manifold and propose a unified framework that explicitly mitigates it through pseudo-label alignment, node reordering, and ambiguity filtering. Extensive experiments on diverse benchmarks show that our approach consistently outperforms existing methods, especially under severe class imbalance. Our findings offer new theoretical insights and practical tools for robust semi-supervised node classification.

## 1   Introduction

Class imbalance is a prevalent challenge in real-world graph datasets, where the disproportionate distribution of class labels often hinders the effectiveness and fairness of Graph Neural Networks (GNNs) [73, 37]. While GNNs have achieved remarkable success in various graph-based learning tasks [25, 54], their message passing mechanisms can amplify the negative impact of class imbalance, especially for minority classes. This challenge is further exacerbated in semi-supervised settings, where only a small fraction of nodes are labeled, making the reliable propagation of information even more difficult.

Self-training (ST) [63, 48], which iteratively incorporates high-confidence pseudo-labeled nodes into the training set, offers a simple yet effective approach for leveraging unlabeled data to address class imbalance. Compared to oversampling [73, 37] or loss function engineering [6, 47], ST avoids the complexities of synthesizing new nodes or edges and thus is naturally well-suited for graph-structured data. However, despite its practical success, existing ST-based work for imbalanced node classification [75, 71, 27] lacks a principled theoretical and empirical understanding of self-training in imbalanced node classification scenarios, especially regarding its inherent limitations under severe imbalance.

In this work, we for the first time formally introduce and analyze the notion of *geometric imbalance* in the context of semi-supervised imbalanced node classification. We rigorously characterize geometric imbalance on riemannian manifold based on the von Mises-Fisher (vMF) distribution, and establish its direct relationship with prediction uncertainty and class imbalance ratio. Our theoretical results show that message passing, especially on class-imbalanced graphs, induces geometric ambiguity

---

[*]Corresponding author.

in the embedding space, making minority-class nodes particularly vulnerable to unreliable pseudo-labeling. This previously overlooked phenomenon motivates the need for targeted solutions beyond existing self-training strategies. It is worth noting that our proposed concept of geometric imbalance is fundamentally distinct from previously studied *topology imbalance* [6, 37]. Specifically, geometric imbalance is defined in the riemannian manifold node embedding space, whereas topology imbalance is defined on the original graph structure. Moreover, topology imbalance focuses on the influence of *labeled* minority nodes in the message-passing process on the raw graph, while our work emphasizes the positional ambiguity of *unlabeled* nodes in the riemannian manifold space (A detailed discussion of this distinction can be found in Appendix G.).

Building on this foundation, we propose a unified and modular framework that explicitly addresses geometric imbalance through three complementary components: (1) a *DualPath PseudoLabeler* that aligns clustering and classification perspectives for more reliable pseudo-labels; (2) a *Node-Reordering* strategy that fuses geometric proximity with classifier confidence; and (3) a lightweight mechanism for discarding geometrically ambiguous nodes. Extensive experiments on nine benchmark datasets—including both synthetic and naturally imbalanced settings—demonstrate that our framework consistently outperforms state-of-the-art baselines, especially under severe imbalance. Detailed ablation and sensitivity studies further validate the effectiveness of each module. Our contributions advance the theoretical understanding of geometric imbalance in GNN-based self-training and offer a robust solution for real-world semi-supervised imbalanced node classification.

## 2 Preliminaries

**Notations and Task Definitions.** In this work, we focus on the problem of semi-supervised imbalanced node classification (SINC) on an undirected and unweighted graph, denoted as $\mathcal{G} = (\mathcal{V}, \mathcal{E}, \mathcal{L})$. Here, $\mathcal{V}$ represents the set of nodes, $\mathcal{E}$ denotes the set of edges, and $\mathcal{L}$ is the set of node labels. The labeled nodes form a subset $\mathcal{D}^{\text{label}} \subset \mathcal{V}$, while the unlabeled nodes are denoted as $\mathcal{D}^{\text{unlabel}} = \mathcal{V} \setminus \mathcal{D}^{\text{label}}$. The feature matrix $\mathcal{X} \in \mathbb{R}^{n \times d}$ contains node features. Here, $n = |\mathcal{V}|$ is the number of nodes, and $d$ is the feature dimensionality. To represent the graph structure, we use the adjacency matrix $\mathcal{A} \in \{0, 1\}^{n \times n}$. For each node $v$, its 1-hop neighbors are denoted by $\mathcal{N}(v)$. Nodes in the labeled set are divided into $C$ classes, denoted as $(\mathcal{C}_1, \mathcal{C}_2, \ldots, \mathcal{C}_C)$ with the number of nodes in each class represented as $(n_1, n_2, \ldots, n_C)$. To quantify class imbalance, we define the imbalance ratio as $\rho = \frac{\max_i(n_i)}{\min_i(n_i)}$, where $i \in \{1, 2, \ldots, C\}$. The objective is to design a robust model $f_\theta$ that effectively utilizes the imbalanced labeled set to perform accurate and fair node classification. Table 8 in Appendix A summarizes all notations used throughout the paper.

**Message Passing Neural Networks.** Message Passing Neural Networks (MPNNs) form the foundation of many graph-based learning tasks. An MPNN typically comprises three components: a message function $m_l$, an aggregation function $\theta_l$, and a node feature update function $\psi_l$. At each layer, the features of a node $v$ are updated. Let $h_v^{(l)}$ represent the feature of node $v$ at the $l$-th layer. Then, the feature update for the $(l + 1)$-th layer is given by $h_v^{(l+1)} = \psi_l \left( h_v^{(l)}, \theta_l \left( \left\{ m_l \left( h_v^{(l)}, h_{\hat{v}}^{(l)}, e_{v, \hat{v}} \right) \mid \hat{v} \in \mathcal{N}(v) \right\} \right) \right)$, where $\mathcal{N}(v)$ denotes the set of neighbors of node $v$, and $e_{v, \hat{v}}$ represents the edge weight between $v$ and $\hat{v}$. To complete the node classification task, a classification layer, typically implemented as a fully connected or softmax layer, is appended after the final GNN layer to map the node embeddings to class probabilities.

**Graph Self-Training Paradigm.** In graph semi-supervised learning [25, 54], graphs encode relationships between data points as edges, enabling information sharing across the graph structure. This facilitates the propagation of labels from labeled to unlabeled nodes. Self-training methods [26, 63, 48] align well with this framework by iteratively incorporating pseudo-labeled data into the training process, thereby expanding the labeled set and improving the model's performance on unlabeled data. The self-training pipeline can be summarized as follows. Let $\mathcal{D}^{\text{label}} = \{(v^j, y^j)\}_{j=1}^{|\mathcal{D}^{\text{label}}|}$ denote the labeled dataset and $\mathcal{D}^{\text{unlabel}} = \{u^j\}_{j=1}^{|\mathcal{D}^{\text{unlabel}}|}$ the unlabeled dataset. Initially, a GNN $f_\theta$ is trained on $\mathcal{D}^{\text{label}}$. At each iteration $t$, the model predicts labels $\hat{y}^j$ for nodes in $\mathcal{D}^{\text{unlabel}}$. A subset of pseudo-labeled nodes $\mathcal{D}_t^{\text{pseudo}} = \{(u^j, \hat{y}^j)\}$, selected based on prediction confidence (e.g., selecting predictions with probabilities above a predefined threshold), is added to $\mathcal{D}^{\text{label}}$. The model is then retrained iteratively with the augmented labeled set, gradually incorporating more pseudo-labeled data in each step.

# 3 Geometric Imbalance

In this part, we provide a novel theoretical analysis for SINC and introduce the geometric imbalance problem induced by class imbalance through message passing.

**Theoretical Background.** We conduct the analysis on the unit hyperspher (a simple riemannian manifold) because GNN embeddings by message passing are typically normalized before classification to ensure stability and improve training dynamics [25, 29]. This normalization maps embeddings onto a riemannian manifold, which is consistent with the assumptions of von Mises-Fisher (vMF) distribution [14]. The riemannian manifold further allows us to rigorously define and analyze geometric imbalance in terms of angular distances and class separation [28, 55]. Following the notation introduced in Section 2, the class prior distribution is given by $p_{\mathcal{D}}^{\text{label}}(i) = \frac{n_i}{|\mathcal{D}^{\text{label}}|}$. Given a node $v$ sampled from the node set $\mathcal{V}$, the feature vector $h_v$ is extracted by a message passing neural network. Specifically, we adopt a simple message passing neural network, which computes node embeddings iteratively. For node $v$, its representation after one layer of GNN message passing can be written as $h_v^{(l)} = \sigma(W^{(l)} \cdot (\sum_{\hat{v} \in \mathcal{N}(v)} \alpha_{v\hat{v}} h_{\hat{v}}^{(l-1)} + \alpha_{vv} h_v^{(l-1)}))$, where $W^{(l)} \in \mathbb{R}^{d \times d}$ is the trainable weight matrix at the $l$-th layer, $\sigma(\cdot)$ denotes a nonlinear activation function (e.g., ReLU), and $\alpha_{v\hat{v}}$ is the edge weight, representing the contribution of node $v$ to node $\hat{v}$. Through message passing, the representation $h_v^{(l)}$ is a weighted combination of information from its neighbors.

The embeddings $h_v^{(l)}$ are then projected onto the riemannian manifold $\mathbb{S}^{d-1}$ using $\tilde{h}_v^{(l)} = \frac{h_v^{(l)}}{\|h_v^{(l)}\|_2}$, and subsequently fed into a von Mises-Fisher (vMF) classifier [2]. The latent representations corresponding to the $C$ classes are modeled as a mixture of $C$ von Mises-Fisher (vMF) distributions defined on the riemannian manifold $\mathbb{S}^{d-1}$. Each class is parameterized by a compactness $\kappa_i \in \mathbb{R}^+$ and a unit orientation vector $\tilde{\boldsymbol{\mu}}_i \in \mathbb{R}^{1 \times d}$. The vMF classifier is defined as $\boldsymbol{\Phi}(\cdot; \mathcal{K}, \mathcal{M})$, where $\mathcal{K} = \{\kappa_1, \ldots, \kappa_C\}$ and $\mathcal{M} = \{\tilde{\boldsymbol{\mu}}_1, \ldots, \tilde{\boldsymbol{\mu}}_C\}$ are the sets of learnable compactness and orientation vectors for the $C$ classes, respectively. The probability density function (PDF) of the $i$-th class, denoted as $p(\tilde{h}_v^{(l)} | \kappa_i, \tilde{\boldsymbol{\mu}}_i)$, is given by $p(\tilde{h}_v^{(l)} | \kappa_i, \tilde{\boldsymbol{\mu}}_i) = C_d(\kappa_i) e^{\kappa_i \cdot \tilde{h}_v^{(l)} \tilde{\boldsymbol{\mu}}_i^\top}$, where $C_d(\kappa) = \frac{\kappa^{\frac{d}{2}-1}}{(2\pi)^{\frac{d}{2}} \cdot I_{\frac{d}{2}-1}(\kappa)}$ is the normalization constant for the $d$-dimensional von Mises-Fisher distribution, ensuring that the density integrates to one. Here, $d$ is the dimensionality of the embedding space, and $I_r(\kappa)$ denotes the modified Bessel function of the first kind of order $r$ [23]. Using Bayes' theorem [21], the posterior probability $p(y^l = i | \tilde{h}_v^{(l)})$ is:

$$p_v^i = p(y^v = i | \tilde{h}_v^{(l)}) = \frac{p_{\mathcal{D}}^{\text{label}}(i) \cdot p(\tilde{h}_v^{(l)} | \kappa_i, \tilde{\boldsymbol{\mu}}_i)}{\sum_{j=1}^C p_{\mathcal{D}}^{\text{label}}(j) \cdot p(\tilde{h}_v^{(l)} | \kappa_j, \tilde{\boldsymbol{\mu}}_j)}.$$

If $v \in \mathcal{D}^{\text{unlabel}}$, the posterior probability $p_v^i$ also represents the classifier's confidence in assigning the pseudo-label $i$ to the unlabeled node $v$. For a node $u^j$ in the unlabeled set $\mathcal{D}^{\text{unlabel}}$, the information entropy of the predictions for $u^i$ is defined as $H(u^j) = -\sum_{i=1}^C p_{u^j}^i \log p_{u^j}^i$, and the average entropy over all unlabeled nodes is $\hat{H} = \frac{1}{|\mathcal{D}^{\text{unlabel}}|} \sum_{j=1}^{|\mathcal{D}^{\text{unlabel}}|} H(u^j)$.

To further analyze the uncertainty introduced by class imbalance through message passing in the riemannian manifold embedding space, we introduce the notion of *geometric imbalance*. This concept captures the ambiguity of a node's embedding position with respect to class centers on the hypersphere, especially for minority class nodes. Intuitively, a node that is nearly equidistant from multiple class centers is geometrically ambiguous and more likely to be assigned a wrong pseudo-label. We formalize this phenomenon by defining an $\epsilon$-geometric imbalanced sample and provide a theoretical connection between geometric imbalance and prediction uncertainty in terms of entropy. The following definition and theorem establish the quantitative foundation of our analysis.

**Definition 1** ($\epsilon$-Geometric Imbalance on the Riemannian Manifold). *To characterize directional ambiguity in hyperspherical embeddings obtained by message passing, we define the geometric imbalance score for unlabeled nodes as follows. Let $u^j \in \mathcal{D}^{unlabel}$ be an unlabeled node with*

---

[2] This choice is motivated by the fact that the von Mises-Fisher (vMF) distribution is a natural analogue of Gaussian distribution on the riemannian manifold. It has been widely used to model directional data and has shown effectiveness in normalized embedding spaces [14, 1]. Compared to Euclidean Gaussian modeling, vMF offers better geometric interpretability when embeddings are $\ell_2$-normalized.

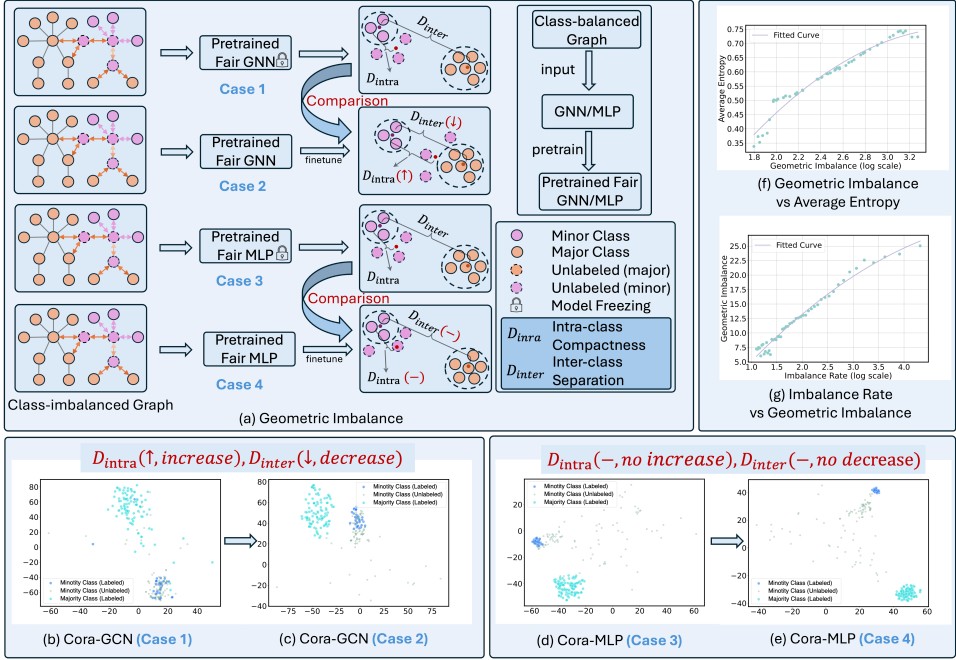

Figure 1: Illustration and quantitative analysis of Geometric Imbalance (GI). (a) Conceptual illustration of GI under different pretrained and fine-tuned settings. (b)-(e): t-SNE visualizations of node embeddings in the Cora dataset under four representative cases, showing intra-class compactness and inter-class separation patterns. (f)-(g): Quantitative relationships between GI and (f) average entropy, and (g) class imbalance ratio.

*normalized embedding $\tilde{h}_{u^j}^{(l)} \in \mathbb{S}^{d-1}$ produced by the l-th layer of a GNN. Let $\{\tilde{\boldsymbol{\mu}}_c \in \mathbb{S}^{d-1}\}_{c=1}^C$ denote the class center embeddings on the hypersphere for C classes, and let $y^{u^j} \in \{1, \ldots, C\}$ be the ground-truth label of $u^j$. We define the geometric imbalance score of $u^j$ as:*

$$G(u^j) := \frac{\left\| \tilde{h}_{u^j}^{(l)} - \tilde{\boldsymbol{\mu}}_{y^{u^j}} \right\|^2}{\sum_{\mathcal{C}_1 \neq \mathcal{C}_2} \left\| \tilde{\boldsymbol{\mu}}_{\mathcal{C}_1} - \tilde{\boldsymbol{\mu}}_{\mathcal{C}_2} \right\|^2}.$$

*We say that $u^j$ is an ε-geometrically imbalanced sample if $G(u^j) > \epsilon$, where $\epsilon > 0$ is a user-defined threshold.*

We define $V_{\text{minor}} := \{v \in \mathcal{V} \mid y_v \in \mathcal{C}_{\text{minor}}\}$ as the set of all nodes whose ground-truth labels belong to minority classes, where $\mathcal{C}_{\text{minor}} \subset \{1, \ldots, C\}$ denotes the set of minority class indices. The following theorem establishes a direct relationship between geometric imbalance and the prediction entropy of pseudo-labels. It shows that higher geometric imbalance is associated with greater uncertainty, especially among minority-class nodes.

**Theorem 1** (Geometric Imbalance vs. Information Entropy). *Let $\mathcal{D}^{unlabel} \subset \mathcal{V}$ be the set of unlabeled nodes, $V_{minor} := \{v \in \mathcal{V} \mid y_v \in \mathcal{C}_{minor}\}$ denote the set of all nodes whose ground-truth labels belong to the minority class set $\mathcal{C}_{minor} \subset \{1, \ldots, C\}$. Define the intra-class compactness and inter-class separation as:*

$$D_{intra} := \sum_{u^j \in V_{minor} \cap \mathcal{D}^{unlabel}} \left\| \tilde{h}_{u^j}^{(l)} - \tilde{\boldsymbol{\mu}}_{y^{u^j}} \right\|^2, \quad D_{inter} := \sum_{\mathcal{C}_1 \neq \mathcal{C}_2} \left\| \tilde{\boldsymbol{\mu}}_{\mathcal{C}_1} - \tilde{\boldsymbol{\mu}}_{\mathcal{C}_2} \right\|^2.$$

*Then the average information entropy $\hat{H}$ of pseudo-label predictions over $\mathcal{D}^{unlabel}$ satisfies $\hat{H} \propto \frac{D_{intra}}{D_{inter}}$, i.e., the expected prediction uncertainty increases with greater intra-class compactness and decreases with greater inter-class separation. The proof is in Appendix C.*

Definition 1 and Theorem 1 offer a quantitative lens through which to understand geometric imbalance. Specifically, we define the intra-class compactness of minority class nodes as $D_{\text{intra}} = \sum_{u^j \in V_{\text{minor}} \cap \mathcal{D}^{\text{unlabel}}} \left\| \tilde{h}_{u^j}^{(l)} - \tilde{\mu}_{\mathcal{C}_i} \right\|^2$, and the inter-class separation as $D_{\text{inter}} = \sum_{\mathcal{C}_1 \neq \mathcal{C}_2} \left\| \tilde{\mu}_{\mathcal{C}_1} - \tilde{\mu}_{\mathcal{C}_2} \right\|^2$. From Theorem 1, it follows that reducing $D_{\text{intra}}$ or increasing $D_{\text{inter}}$ leads to a decrease in average entropy $\hat{H}$, thus improving pseudo-label reliability. This underscores the importance of mitigating

geometric imbalance in embedding space. Furthermore, we investigate how the degree of geometric imbalance is affected by the class distribution in the training set. The next theorem shows that the geometric imbalance metric grows with the class imbalance ratio.

**Theorem 2** (Imbalance Ratio vs. Geometric Imbalance). *Let $\rho$ denote the class imbalance ratio, defined as the ratio between the number of samples in the majority class and the number in the minority class. Let the geometric imbalance of unlabeled minority nodes be measured by:*

$$\bar{G}_{minor} := \frac{1}{|V_{minor} \cap \mathcal{D}^{unlabel}|} \sum_{u^j \in V_{minor} \cap \mathcal{D}^{unlabel}} G(u^j),$$

*where $G(u^j)$ is the geometric imbalance score as defined in Definition 1. Then, under fixed feature extraction and class centers, the expected geometric imbalance $\bar{G}_{minor}$ increases monotonically with the class imbalance ratio $\rho$: $\bar{G}_{minor} \propto \rho$. The proof is in Appendix D.*

This result characterizes the empirical observation that datasets with greater class imbalance exhibit higher average geometric ambiguity among minority-class unlabeled nodes, highlighting the structural difficulty in correctly pseudo-labeling minority nodes when training data is skewed. Importantly, this theoretical formulation is particularly relevant to GNNs, where message passing mechanisms tend to amplify the effects of class imbalance, resulting in more pronounced geometric confusion for minority-class nodes compared to non-graph-based models such as MLPs. As we will later demonstrate, MLPs—lacking structural aggregation—are far less affected by class imbalance in this manner. Together, Theorem 1 and Theorem 2 highlight the dual challenges posed by geometric ambiguity and class imbalance in semi-supervised GNN settings. These insights motivate the development of a principled approach to detect and mitigate geometric imbalance, which we introduce in the next section.

**Empirical Illustration.** The above theoretical analysis is visually and quantitatively substantiated in Figure 1. Figure 1(a) presents a conceptual overview of geometric imbalance under different pretraining and fine-tuning scenarios for both GNNs and MLPs. Figure 1(b)–(e) use t-SNE visualizations to directly contrast the evolution of node embeddings: for GNNs (Case 1 and 2), fine-tuning on imbalanced data leads to a substantial increase in intra-class dispersion and a reduction in inter-class separation for minority classes, resulting in pronounced geometric imbalance. In contrast, MLPs (Case 3 and 4) preserve compact and well-separated clusters, showing minimal change regardless of class imbalance. This empirical contrast highlights that geometric imbalance is a distinctive and aggravated issue in GNNs, primarily due to their message passing mechanisms, while it remains marginal in MLPs. Furthermore, Figure 1(f) and (g) quantitatively validate our theoretical results, demonstrating a strong positive correlation between geometric imbalance and prediction entropy (as predicted by Theorem 1), as well as a monotonic relationship between geometric imbalance and the class imbalance ratio (as characterized by Theorem 2). Detailed experimental setups and result analyses for Figure 1(b)-(e) and Figure 1(f)-(g) are provided in Appendix E and F, respectively. Taken together, these empirical observations in Figure 1 not only substantiate our theoretical claims but also underscore the need for GNN-specific mitigation strategies, which we propose in the following section.

## 4  Method

A straightforward strategy to mitigate geometric imbalance is to compute the geometric imbalance score for each unlabeled node and discard those with high scores during self-training. However, this approach suffers from two major limitations: (1) it is computationally expensive, as it requires evaluating pairwise angular distances between each unlabeled node and all class centroids; and (2) it relies on ground-truth labels to determine which class a node should be close to, or alternatively, depends on potentially noisy pseudo-labels—making the approach unreliable and indirect in practice. To address these challenges more efficiently and effectively, we propose a unified and modular framework, UNREAL, named to reflect its nature as a pseudo-labeling algorithm. Our UNREAL framework mitigates geometric imbalance through three flexible and complementary components: (1) a *DualPath PseudoLabeler* that enhances pseudo-label quality via alignment between clustering and classification; (2) a *Node-Reordering* mechanism that jointly considers geometric proximity and classifier confidence; and (3) a *Discarding Geometrically Imbalanced Nodes (DGIS)* module that filters out samples with ambiguous geometric positioning. Notably, DGIS can be viewed as a lightweight, approximate alternative to direct geometric imbalance scoring, enabling scalable filtering without requiring true labels or dense computations. These components can be used independently or in combination, allowing the framework to adapt to various scenarios and computational budgets.

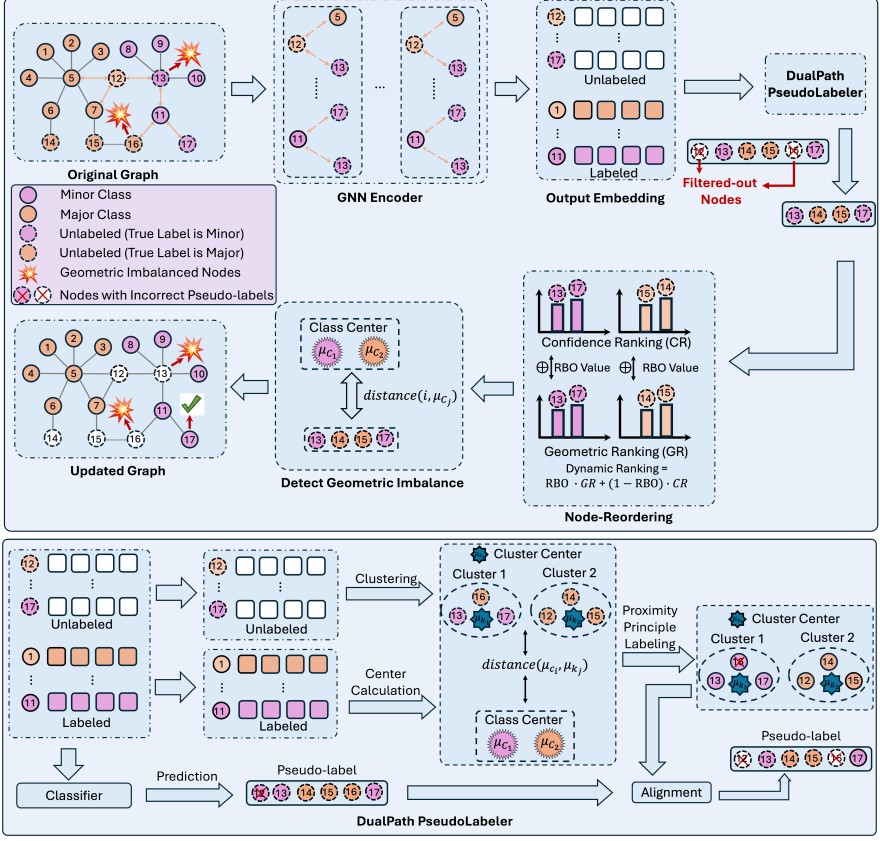

Figure 2: The pipeline of our UNREAL framework.

The overall pipeline is illustrated in Figure 2, and we elaborate on each component in the following subsections.

### 4.1 Mitigating Geometric Imbalance with Pseudo-label Alignment

Traditional node pseudo-labeling methods rely on embeddings generated by a graph encoder, which are finaly fed into a classifier to get pseudo-labels. However, the geometric imbalance problem we address originates in the embedding space itself, creating a mismatch between mitigating geometric imbalance and improving pseudo-label generation. To address this, we propose a novel approach, the DualPath PseudoLabeler, as illustrated in Figure 2. This method generates pseudo-labels via two pathways: (1) unsupervised clustering (using k-means to segment the embedding space) and (2) supervised classification (using a dual-layer MLP for node prediction).

**DualPath PseudoLabeler Overview.** Let the embeddings of labeled nodes are represented as $\mathcal{H}^{\text{label}} \in \mathbb{R}^{|\mathcal{D}^{\text{label}}| \times d}$, and the embeddings of unlabeled nodes as $\mathcal{H}^{\text{unlabel}} \in \mathbb{R}^{|\mathcal{D}^{\text{unlabel}}| \times d}$. Each row of these matrices represents a node embedding, denoted as $h_v$ or $h_u$, which corresponds to a point in the $d$-dimensional Euclidean space. (1) *Unsupervised Clustering.* We apply k-means clustering [3] to the embeddings of unlabeled nodes $\mathcal{H}^{\text{unlabel}}$, partitioning them into $k'$ clusters, where $K > C$ [4] and $C$ denotes the number of classes. This clustering process produces cluster centers and cluster assignments: $f_{\text{cluster}}(\mathcal{H}^{\text{unlabel}}) \to \{\mathcal{K}_1, \mu_{\mathcal{K}_1}, \mathcal{K}_2, \mu_{\mathcal{K}_2}, \dots, \mathcal{K}_K, \mu_{\mathcal{K}_K}\}$, where $\mathcal{K}_i$ is the $i$-th cluster and $\mu_{\mathcal{K}_i}$ is its center. For labeled nodes, we compute the embedding center for each class: $\boldsymbol{\mu}_{\mathcal{C}_i} = \text{mean}(\{h_v \mid y^v \in c_i\})$, where $\text{mean}(\cdot)$ is the mean function. We assign pseudo-label to the cluster $\mathcal{K}_i$ by minimizing the distance between cluster centers and class centers as $\tilde{y}^i = \arg\min_j \text{distance}(\boldsymbol{\mu}_{\mathcal{C}_j}, \mu_{\mathcal{K}_i})$. Nodes with the same predicted label $i$ are grouped into sets $\tilde{\mathcal{U}}_i$, such that $\mathcal{D}^{\text{unlabel}} = \bigcup_{i=1}^{C} \tilde{\mathcal{U}}_i$. (2) *Supervised Classification.* Simultaneously, the GNN model

---

[3] Although embeddings in our model are approximately distributed on a hypersphere, we employ K-Means clustering for its computational efficiency and robustness (See Appendix J.6.3 for extended discussion.).

[4] This overclustering strategy allows for finer-grained partitioning of the embedding space, which is particularly beneficial in imbalanced or geometrically entangled regions. By assigning more clusters than classes, we aim to capture local structures within each class and reduce the impact of overlapping or ambiguous boundaries. The strategy ($K > C$) has been commonly adopted in deep clustering and pseudo-labeling to improve separation in ambiguous regions [20, 53, 18, 10, 36].

generates predictions $\hat{y}^u$ for each node in $\mathcal{D}^{\text{unlabel}}$. Nodes with the same predicted label $i$ are grouped into sets $\mathcal{U}_i$, such that $\mathcal{D}^{\text{unlabel}} = \bigcup_{i=1}^{C} \mathcal{U}_i$.

**Dual Pseudo-label Alignment Mechanism (DPAM).** To address the geometric ambiguity of clustering-based pseudo-labels and the majority bias of classifier predictions, DPAM aligns both sources by retaining only nodes with consistent labels from clustering and classification, i.e., $\mathcal{U}_i^{\text{final}} = \tilde{\mathcal{U}}_i \cap \mathcal{U}_i$. This intersection acts as a filter, improving pseudo-label reliability and mitigating geometric imbalance, while remaining broadly applicable to different clustering and classification models.

### 4.2 Node-Reordering

While classifier confidence is often used for pseudo-label selection, it does not explicitly address geometric imbalance—especially in early training stages where node embeddings may overlap or drift between class regions. On the other hand, geometric proximity to class centroids provides valuable structural information, but lacks semantic certainty. To reconcile these two perspectives, we propose *Node-Reordering (NR)*, a dynamic ranking strategy that adaptively fuses geometric and confidence-based rankings for more robust pseudo-label selection.

**Ranking Definitions.** We define two types of rankings over the candidate pseudo-labeled nodes $\tilde{\mathcal{U}}_i \cap \mathcal{U}_i$:

**Definition 2** (Confidence Rankings (CR)). *For each node $u$, the classifier outputs a confidence score* $\text{confidence}(u) = \max(\text{softmax}(\text{logits}_u))$, *where* $\text{logits}_u$ *is the output vector of the final classification layer. Nodes are then ranked in descending order of their confidence scores, yielding per-class rankings* $\{\mathcal{T}_1, \mathcal{T}_2, \ldots, \mathcal{T}_L\}$.

**Definition 3** (Geometric Rankings (GR)). *For each node $u$, let* $\boldsymbol{\mu}_{\mathcal{C}_i}$ *denote the centroid of class $i$. The geometric distance is defined as* $\delta_u = \text{distance}(h_u, \boldsymbol{\mu}_{\mathcal{C}_i})$, *where $h_u$ is the embedding of node $u$. Nodes are ranked in ascending order of their distances, forming the geometric rankings* $\{\mathcal{S}_1, \mathcal{S}_2, \ldots, \mathcal{S}_L\}$.

**Fusing Rankings via RBO.** To combine the two rankings, we adopt *Rank-Biased Overlap (RBO)* [62], a metric that measures the agreement between two ranked lists, assigning higher weights to top-ranked elements. For each class $m$, we compute the similarity score $r_m = \text{RBO}(\mathcal{S}_m, \mathcal{T}_m)$ between its geometric and confidence rankings.

**Weighted Node-Reordering.** Based on the computed RBO score, we construct a fused ranking $\mathcal{N}_m^{\text{New}}$ by adaptively weighting the two sources, $\mathcal{N}_m^{\text{New}} = \max\{r_m, 1-r_m\}\cdot\mathcal{S}_m + \min\{r_m, 1-r_m\}\cdot\mathcal{T}_m$. This formulation ensures that when geometric and confidence rankings diverge (i.e., low $r_m$), the dominant weight is assigned to the geometric perspective, which is more reliable in early stages (Detailed ablation results are shown in Appendix J.3 and J.4.). As training progresses and $r_m$ increases, the classifier's confidence gradually plays a greater role in node selection.

**Theoretical Justification.** The following theorem characterizes the contribution of geometric rankings in the fused list:

**Theorem 3.** *The proposed ranking $\mathcal{N}_m^{\text{New}}$ guarantees that geometric rankings dominate the node selection process when the disagreement between confidence and geometry is high, thereby reducing the effect of geometric imbalance in early iterations. The proof is provided in Appendix H.*

**Empirical Dynamics.** We empirically observe that as training proceeds, the similarity between CR and GR improves. Figure 3 illustrates the evolution of RBO values over iterations, confirming that $\mathcal{S}_m$ and $\mathcal{T}_m$ become increasingly aligned. This observation supports our design: geometric criteria dominate early on to stabilize selection, while confidence scores gain influence as the classifier matures. Additional analyses are provided in Appendix J.4.

### 4.3 Mitigating Learning Bias by Discarding Geometric Imbalanced Samples

While the previous components (DPAM and NR) effectively alleviate geometric imbalance by enhancing pseudo-label quality and selection, a more direct strategy is to explicitly identify geometrically imbalanced nodes based on the formal definition in Definition 1 and discard them from training. Although intuitive, this approach incurs high computational cost if applied naively to all unlabeled nodes, as it requires evaluating distances to multiple class centroids. Moreover, it depends on reliable pseudo-labels or ground-truth labels to determine geometric deviation, which may not be available or accurate during early

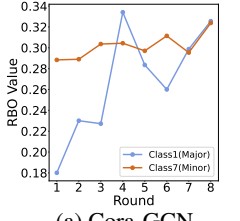 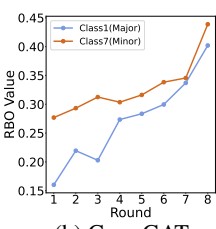

(a) Cora-GCN     (b) Cora-GAT

Figure 3: Fluctuation of RBO values between rankings as iterations progress.

training stages. To overcome these limitations, we defer this operation until after DPAM and Node-Reordering have been applied, thereby reducing the candidate pool and improving label quality. In practice, we adopt a simplified and computationally efficient version of the geometric imbalance score—serving as a lightweight surrogate for full hyperspherical analysis.

**Discarding Geometric Imbalanced Samples (DGIS).** We define a *Geometric Imbalance (GI) index* to approximate the degree of ambiguity in a node's embedding. Let $\delta_u$ be the distance between node $u$'s embedding and its closest class centroid, and let $\beta_u$ denote the distance to its second-closest centroid. Since $\delta_u \leq \beta_u$, a smaller gap between them indicates that the node is nearly equidistant from multiple centroids, and thus more likely to be geometrically ambiguous. The GI index is defined as $\text{GI}_u = \frac{\beta_u - \delta_u}{\delta_u}$. Nodes with lower GI indices are more prone to geometric imbalance. We therefore introduce a threshold on the GI index and discard nodes whose values fall below this threshold. This filtering mechanism allows us to eliminate geometric imbalanced samples with minimal computational cost, further reducing learning bias and improving model robustness.

## 5 Experiment

In this section, we conduct comprehensive experiments to address the following research questions: **(RQ1)** How does our model perform under varying levels of class imbalance? **(RQ2)** How does it perform on datasets with naturally occurring label imbalance? **(RQ3)** Is the model scalable and effective on large-scale datasets with severe imbalance? **(RQ4)** How sensitive is the model to its key components and hyperparameters?

### 5.1 Experimental Setups

**(I) Datasets.** We conduct evaluations under various benchmarking settings on 8 datasets—*Cora* [68], *Citeseer* [68], *Pubmed* [68], *Amazon-Computers* [43], *Computers-Random* [67], *CS-Random* [67], *Flickr* [69], and *Ogbn-arxiv* [17]—capturing a wide range of class imbalance scenarios. To answer **RQ1**, we construct *varying levels of class imbalance (ρ = 10, 20, 50, 100)* settings on four datasets (*Cora*, *Citeseer*, *Pubmed*, and *Amazon-Computers*), following the methodology from [73, 37, 47]. Specifically, we designate half of the classes as minorities and convert a portion of labeled nodes into unlabeled ones to achieve the desired imbalance ratio ($\rho$). For the citation networks (*Cora*, *Citeseer*, and *Pubmed*), we use the standard splits from [68] to create imbalance settings with $\rho = 10$ and $\rho = 20$. For more extreme imbalances ($\rho = 50$ and $100$), which require more labeled nodes per class, we adopt random splits. For *Amazon-Computers*, we generate splits with varying degrees of class imbalance ($\rho = 10, 20, 50, 100$) based on the procedure in [67].

To answer **RQ2** and **RQ3**, we extend our experiments to naturally imbalanced and large-scale datasets, *Computers-Random*, *CS-Random*, *Flickr*, *Ogbn-arxiv*, reflecting real-world conditions. Here, random sampling is employed to construct training sets that reflect the original label distributions for *Computers-Random* and *CS-Random*, following the protocol in [67]. For *Flickr* and *Ogbn-arxiv*, we adopt their publicly available splits, as the settings are inherently highly imbalanced. Appendix K details our experimental framework, including label distributions, evaluation protocols, and algorithm implementations.

Table 1: Ablation analysis for UNREAL.

| Exps \ Modules | CR | GR | NR | DGIS | F1 |
|---|:--:|:--:|:--:|:--:|---|
| **Cora+GCN** ($\rho = 10$) | × | ✓ | ✓ | ✓ | $73.93_{\pm 0.95}$ |
| | × | ✓ | ✓ | × | $72.74_{\pm 0.63}$ |
| | ✓ | × | ✓ | ✓ | $75.85_{\pm 0.82}$ |
| | ✓ | × | ✓ | × | $75.34_{\pm 0.63}$ |
| | ✓ | ✓ | × | ✓ | $75.00_{\pm 0.97}$ |
| | ✓ | ✓ | × | × | $\mathbf{76.44_{\pm 1.06}}$ |
| **CiteSeer+SAGE** ($\rho = 20$) | × | ✓ | ✓ | ✓ | $46.09_{\pm 4.08}$ |
| | × | ✓ | ✓ | × | $47.76_{\pm 1.06}$ |
| | ✓ | × | ✓ | ✓ | $50.32_{\pm 3.75}$ |
| | ✓ | × | ✓ | × | $53.32_{\pm 3.75}$ |
| | ✓ | ✓ | × | ✓ | $55.43_{\pm 2.14}$ |
| | ✓ | ✓ | × | × | $\mathbf{57.51_{\pm 4.92}}$ |
| **PubMed+GAT** ($\rho = 50$) | × | ✓ | ✓ | ✓ | $76.34_{\pm 0.39}$ |
| | × | ✓ | ✓ | × | $75.42_{\pm 0.39}$ |
| | ✓ | × | ✓ | ✓ | $77.32_{\pm 0.21}$ |
| | ✓ | × | ✓ | × | $76.89_{\pm 1.43}$ |
| | ✓ | ✓ | × | ✓ | $76.12_{\pm 2.63}$ |
| | ✓ | ✓ | × | × | $\mathbf{77.38_{\pm 0.39}}$ |
| **Computers+GAT** ($\rho = 100$) | × | ✓ | ✓ | ✓ | $70.86_{\pm 1.73}$ |
| | × | ✓ | ✓ | × | $68.86_{\pm 1.42}$ |
| | ✓ | × | ✓ | ✓ | $72.32_{\pm 2.43}$ |
| | ✓ | × | ✓ | × | $73.65_{\pm 0.67}$ |
| | ✓ | ✓ | × | ✓ | $74.03_{\pm 2.53}$ |
| | ✓ | ✓ | × | × | $\mathbf{75.83_{\pm 0.74}}$ |

**(II) Baselines.** We evaluate our framework against several classic techniques and state-of-the-art methods for addressing imbalanced node classification, including GraphSMOTE (GS) [73], GraphENS (GE) [37], ReNode (RN) [6], TAM [47], GraphSR (GSR) [75], BIM [71]. We also compare our method with cross-entropy loss with Re-Weighting (RW) [19], PC Softmax (PS) [16], and Balanced Softmax (BS) [40] . GS and GE are representative over-sampling methods, while RN and TAM modify the loss function. GSR and BIM are state-of-the-art self-training based models for imbalance node classification. We also test the performance of TAM when combined with different base models, including GE, RN, and BS, as described in [47]. See Appendix K.5 for implementation details of the baselines. **(III) Evaluation.** We evaluate the performance of our method on several mainstream GNN architectures, including GCN [25], GAT [54], and GraphSAGE [12]. We report the averaged balanced accuracy (bAcc., %) and Macro-F1 score (%), along with the standard errors over 5 repetitions on the GNN architectures. The reported metrics include: balanced accuracy (bAcc.) and Macro-F1 (F1).

Table 2: Experimental results of our method and other baselines on three class-imbalanced node classification benchmark datasets with $\rho = 10$.

| Dataset | Cora | | CiteSeer | | PubMed | |
|---|---|---|---|---|---|---|
| Metric | bAcc. | F1 | bAcc. | F1 | bAcc. | F1 |
| Vanilla | 62.82±1.43 | 61.67±1.59 | 38.72±1.88 | 28.74±3.21 | 65.64±1.72 | 56.97±3.17 |
| RW | 65.36±1.15 | 64.97±1.39 | 44.69±1.78 | 38.61±2.37 | 69.06±1.84 | 64.08±2.97 |
| PS | 68.04±0.82 | 67.84±0.81 | 50.18±0.55 | 46.14±0.14 | 72.46±0.80 | 70.27±0.94 |
| GS | 66.39±0.56 | 65.49±0.93 | 44.87±1.12 | 39.20±1.62 | 67.91±0.64 | 62.68±1.92 |
| BS | 69.98±0.58 | 68.68±0.55 | 55.52±0.97 | 53.74±1.42 | 73.73±0.89 | 71.53±1.06 |
| BS(w TAM) | 69.94±0.45 | 69.54±0.47 | 56.73±0.71 | 56.15±0.78 | 74.62±0.97 | 72.25±1.30 |
| RN | 67.03±1.41 | 67.16±1.67 | 43.47±2.22 | 37.52±3.10 | 71.40±1.42 | 67.27±2.96 |
| RN(w TAM) | 68.26±1.84 | 68.11±1.97 | 46.20±1.17 | 39.96±2.76 | 72.63±2.03 | 68.28±3.30 |
| GE | 70.89±0.71 | 70.90±0.81 | 56.57±0.98 | 55.29±1.33 | 72.13±1.04 | 70.72±1.07 |
| GE(w TAM) | 71.69±0.36 | 72.14±0.51 | 58.01±0.68 | 56.32±1.03 | 74.14±1.42 | 72.42±1.39 |
| GSR | 70.85±0.44 | 71.37±0.63 | 59.28±0.72 | 55.96±0.95 | 73.61±1.25 | 71.88±1.33 |
| BIM | 72.19±0.42 | 72.67±0.48 | 58.54±0.61 | 56.81±0.98 | 74.62±1.15 | 72.93±1.21 |
| **Ours** | **78.33±1.04** | **76.44±1.06** | **65.63±1.38** | **64.94±1.38** | **75.35±1.41** | **73.65±1.43** |
| Δ | +6.14 | +3.77 | +6.35 | +8.13 | +0.73 | +0.72 |

(Model group: GCN)

Table 3: Experimental results of our method and other baselines on three class-imbalanced node classification benchmark datasets with $\rho = 100$.

| Dataset | Cora | | CiteSeer | | PubMed | |
|---|---|---|---|---|---|---|
| Metric | bAcc. | F1 | bAcc. | F1 | bAcc. | F1 |
| Vanilla | 52.65±0.24 | 43.79±0.47 | 36.63±0.09 | 24.12±0.09 | 62.29±0.25 | 47.02±0.38 |
| RW | 59.42±2.88 | 55.26±4.40 | 36.24±1.30 | 27.07±2.88 | 63.33±0.75 | 55.11±1.62 |
| PS | 64.01±1.15 | 60.74±1.68 | 44.74±1.41 | 37.61±1.69 | 72.62±1.42 | 70.95±1.70 |
| BS | 63.43±2.12 | 62.30±2.27 | 49.33±1.12 | 44.58±1.64 | 70.68±0.92 | 69.15±0.84 |
| BS(w TAM) | 66.58±1.53 | 64.56±2.49 | 53.33±1.06 | 50.15±1.45 | 72.59±2.06 | 72.22±2.08 |
| RN | 62.42±0.90 | 60.08±1.19 | 39.61±2.66 | 30.13±3.86 | 67.11±1.12 | 61.09±3.50 |
| RN(w TAM) | 62.06±2.08 | 60.72±3.32 | 42.08±1.88 | 33.19±3.45 | 69.95±1.01 | 65.99±2.28 |
| GE | 63.09±0.97 | 61.20±1.74 | 42.03±1.88 | 36.71±2.99 | 69.71±1.87 | 63.47±3.87 |
| GE(w TAM) | 65.95±2.25 | 63.88±1.78 | 51.03±1.51 | 50.49±1.88 | 73.58±2.01 | 72.44±1.77 |
| GSR | 66.45±2.10 | 64.42±1.83 | 53.52±1.17 | 53.01±1.75 | 74.09±2.12 | 72.97±1.90 |
| BIM | 67.75±2.13 | 64.68±1.95 | 53.83±1.62 | 53.29±1.80 | 74.38±2.08 | 73.24±1.85 |
| **Ours** | **73.47±2.31** | **68.30±2.11** | **59.77±2.98** | **58.92±3.07** | **77.11±0.59** | **74.03±0.81** |
| Δ | +5.72 | +3.62 | +6.04 | +5.63 | +2.73 | +0.79 |

(Model group: SAGE)

Table 4: Experimental results of our method and other baselines on naturally imbalanced setting Computers-Random ($\rho \approx 17.7$).

| Model | GCN | | GAT | | SAGE | |
|---|---|---|---|---|---|---|
| Metric | bAcc. | F1 | bAcc. | F1 | bAcc. | F1 |
| Vanilla | 78.43±0.41 | 77.14±0.39 | 71.35±1.18 | 69.60±1.11 | 65.30±1.07 | 64.77±1.19 |
| RW | 80.49±0.44 | 75.07±0.58 | 71.95±0.80 | 70.67±0.51 | 66.50±1.47 | 66.10±1.46 |
| PS | 81.34±0.55 | 75.17±0.57 | 70.56±1.46 | 67.26±1.48 | 69.73±0.53 | 67.03±0.60 |
| GS | 80.50±1.11 | 73.79±0.14 | 71.98±0.21 | 67.98±0.31 | 72.69±0.82 | 68.73±1.01 |
| BS | 81.39±0.25 | 74.54±0.64 | 72.09±0.31 | 68.38±0.69 | 73.80±1.06 | 69.74±0.60 |
| BS(w TAM) | 81.64±0.48 | 75.59±0.83 | 74.00±0.77 | 70.72±0.50 | 73.77±1.26 | 71.03±0.69 |
| RN | 81.64±0.34 | 76.87±0.32 | 72.80±0.94 | 71.40±0.97 | 70.94±1.50 | 70.04±1.16 |
| RN(w TAM) | 80.50±1.11 | 75.79±0.14 | 71.98±0.21 | 70.98±0.31 | 72.69±0.82 | 70.73±1.01 |
| GE | 82.66±0.61 | 76.55±0.17 | 75.25±0.85 | 71.49±0.54 | 77.64±0.52 | 72.65±0.35 |
| GE(w TAM) | 82.83±0.68 | 76.76±0.39 | 75.81±0.72 | 72.62±0.57 | 78.98±0.60 | 73.59±0.55 |
| GSR | 83.82±0.74 | 77.78±0.42 | 76.79±0.68 | 73.61±0.63 | 77.63±0.32 | 72.56±0.51 |
| BIM | 84.03±0.73 | 77.96±0.45 | 77.01±0.70 | 73.82±0.60 | 77.76±0.65 | 72.09±0.37 |
| **Ours** | **85.32±0.22** | **80.43±0.56** | **82.52±0.35** | **78.90±0.38** | **75.81±1.86** | **71.86±1.86** |
| Δ | +1.29 | +2.47 | +5.51 | +5.08 | -3.17 | -1.73 |

Table 5: Experimental results of our method and other baselines on naturally imbalanced setting CS-Random ($\rho \approx 41.0$).

| Model | GCN | | GAT | | SAGE | |
|---|---|---|---|---|---|---|
| Metric | bAcc. | F1 | bAcc. | F1 | bAcc. | F1 |
| Vanilla | 84.85±0.16 | 87.12±0.14 | 82.47±0.36 | 84.21±0.31 | 83.76±0.27 | 86.22±0.19 |
| RW | 87.42±0.17 | 88.70±0.10 | 83.55±0.39 | 84.73±0.32 | 85.76±0.24 | 87.32±0.16 |
| PS | 88.36±0.12 | 88.94±0.04 | 85.22±0.31 | 85.54±0.33 | 87.18±0.14 | 88.00±0.19 |
| GS | 85.76±1.73 | 87.31±1.32 | 84.65±1.32 | 85.63±1.01 | 85.76±1.98 | 87.34±0.98 |
| BS | 87.72±0.07 | 88.67±0.07 | 84.38±0.20 | 84.53±0.41 | 86.78±0.10 | 88.05±0.09 |
| BS(w TAM) | 88.22±0.11 | 89.22±0.08 | 85.48±0.24 | 85.77±0.50 | 87.83±0.13 | 88.77±0.07 |
| RN | 87.53±0.11 | 88.91±0.06 | 85.98±0.19 | 86.97±0.09 | 86.13±0.10 | 87.89±0.09 |
| RN(w TAM) | 87.55±0.06 | 89.03±0.05 | 86.61±0.30 | 87.42±0.24 | 85.21±0.33 | 87.01±0.31 |
| GE | 85.97±0.29 | 86.68±0.20 | 85.86±0.19 | 86.51±0.32 | 85.39±0.26 | 86.41±0.24 |
| GE(w TAM) | 86.34±0.12 | 87.36±0.08 | 86.29±0.20 | 87.28±0.13 | 85.99±0.13 | 87.25±0.07 |
| GSR | 86.73±0.22 | 85.91±0.21 | 85.34±0.13 | 86.56±0.29 | 85.44±0.27 | 86.46±0.23 |
| BIM | 86.89±0.22 | 85.99±0.21 | 85.63±1.87 | 86.65±0.35 | 85.65±0.28 | 86.73±0.22 |
| **Ours** | **88.94±0.09** | **89.87±0.06** | **87.65±0.12** | **87.65±0.11** | **88.03±0.21** | **88.65±0.07** |
| Δ | + 0.58 | + 0.65 | + 1.04 | + 0.23 | + 0.20 | - 0.12 |

## 5.2 Experimental Results and Analysis

(1) **For RQ1: Results Under Varying Levels of Class Imbalance (Table 2, Table 3, Table 20, Table 21, Table 22, Table 23).** We evaluate our method under different class imbalance ratios ($\rho = 10, 20, 50, 100$) across multiple benchmark datasets and GNN architectures. As shown in Table 20–23, our method consistently outperforms all baselines in both balanced accuracy and F1 score, regardless of the imbalance severity. The performance advantage remains stable even as the imbalance ratio increases, demonstrating the robustness of our method under skewed label distributions. In contrast, some baselines suffer from drastic drops in performance under severe imbalance or even become inapplicable due to data sparsity in minority classes. These results highlight the effectiveness and generalizability of our method across diverse imbalance settings. Owing to space limitations, we include partial results from Table 20 and Table 23 (aligned with Table 2 and Table 3) in the main text as representative examples.

(2) **For RQ2: Results On Naturally Imbalanced Datasets (Table 4, Table 5).** Additionally, we validate our model on three intrinsically imbalanced datasets: Computers-Random ($\rho \approx 17.7$) and CS-Random ($\rho \approx 41.0$), where the unlabeled data also exhibits imbalance (refer to Table 34). The composition of the training, validation, and testing sets is detailed in Appendix K. We present the experimental results in Table 4 and Table 5. More importantly, on these two datasets, our method consistently outperforms other approaches. (3) **For RQ3: Results On Large-Scale Naturally Imbalanced Datasets (Table 6, Table 7).** We further evaluate our method on two large-scale datasets with naturally occurring class imbalance—Flickr and Ogbn-arxiv. As shown in Table 6 and Table 7, our method achieves competitive or superior performance across all GNN backbones. Despite the high imbalance ratio (especially in Ogbn-arxiv), our model maintains stable performance and outperforms existing methods in most cases. Notably, several baselines encounter out-of-memory issues, while our method remains computationally feasible. These results highlight the scalability and practicality of our approach in real-world imbalanced settings. (4) **For RQ4: Ablation Studies and Sensitivity Analysis (Table 1, Figure 4).** Here we investigates the individual contributions of the key components within our method. We examine three ranking strategies—CR, GR, and GR—as well as the impact of the DGIS module, which targets geometrically imbalanced nodes. As shown in Table 1, the combination of NR and DGIS consistently achieves the best F1 scores in three out of four settings, highlighting the effectiveness of our design. We also conduct a sensitivity analysis on two key hyperparameters: the number of clusters $k'$ in K-Means and the

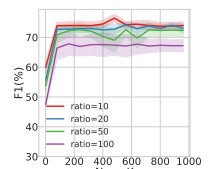

(a) Sensitivity performance of $k'$ of K-Means.

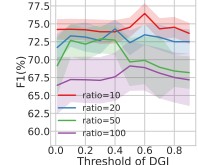

(b) Sensitivity performance of the threshold $\gamma$ of DGIS.

Figure 4: Sensitivity analysis.

Table 6: Experimental results of our method and other baselines on large-scale naturally imbalanced setting Flickr ($\rho \approx 10.8$).

| Model | GCN | | GAT | | SAGE | |
|---|---|---|---|---|---|---|
| Metric | bAcc. | F1 | bAcc. | F1 | bAcc. | F1 |
| Vanilla | $24.62_{\pm 0.07}$ | $24.53_{\pm 0.11}$ | $25.87_{\pm 0.30}$ | $25.32_{\pm 0.44}$ | $25.29_{\pm 0.18}$ | $24.16_{\pm 0.27}$ |
| RW | $28.31_{\pm 1.64}$ | $24.06_{\pm 1.16}$ | $30.66_{\pm 0.76}$ | $27.12_{\pm 0.34}$ | $27.39_{\pm 1.84}$ | $22.62_{\pm 1.04}$ |
| PS | $29.21_{\pm 2.16}$ | $25.81_{\pm 1.75}$ | $30.20_{\pm 0.46}$ | $27.24_{\pm 0.37}$ | $25.40_{\pm 2.49}$ | $21.08_{\pm 1.73}$ |
| GS | OOM | OOM | OOM | OOM | OOM | OOM |
| BS | $27.61_{\pm 0.61}$ | $23.70_{\pm 0.77}$ | $26.01_{\pm 2.81}$ | $23.50_{\pm 3.07}$ | $28.24_{\pm 2.10}$ | $24.98_{\pm 1.59}$ |
| BS(w TAM) | $27.06_{\pm 1.03}$ | $23.97_{\pm 0.60}$ | $28.24_{\pm 0.99}$ | $25.52_{\pm 0.89}$ | $29.79_{\pm 0.37}$ | $27.56_{\pm 0.25}$ |
| RN | OOM | OOM | OOM | OOM | OOM | OOM |
| RN(w TAM) | OOM | OOM | OOM | OOM | OOM | OOM |
| GE | OOM | OOM | OOM | OOM | OOM | OOM |
| GE(w TAM) | OOM | OOM | OOM | OOM | OOM | OOM |
| GSR | $27.63_{\pm 0.59}$ | $23.73_{\pm 0.81}$ | $26.03_{\pm 2.75}$ | $23.53_{\pm 3.15}$ | $28.26_{\pm 2.18}$ | $25.01_{\pm 1.62}$ |
| BIM | $27.87_{\pm 0.65}$ | $23.75_{\pm 0.73}$ | $26.15_{\pm 2.70}$ | $23.74_{\pm 3.10}$ | $28.34_{\pm 2.00}$ | $25.03_{\pm 1.66}$ |
| **Ours** | $30.76_{\pm 0.27}$ | $30.60_{\pm 0.29}$ | $29.45_{\pm 0.72}$ | $28.21_{\pm 0.76}$ | $30.68_{\pm 0.63}$ | $31.01_{\pm 1.34}$ |
| Δ | +1.55 | +4.79 | -1.21 | +0.97 | +0.89 | +3.45 |

Table 7: Experimental results of our method and other baselines on large-scale naturally imbalanced setting Ogbn-arxiv ($\rho \approx 775.4$).

| Model | GCN | | GAT | | SAGE | |
|---|---|---|---|---|---|---|
| Metric | bAcc. | F1 | bAcc. | F1 | bAcc. | F1 |
| Vanilla | $50.21_{\pm 0.65}$ | $49.60_{\pm 0.14}$ | $51.21_{\pm 0.87}$ | $49.23_{\pm 0.33}$ | $50.76_{\pm 0.21}$ | $49.43_{\pm 0.29}$ |
| RW | $50.24_{\pm 0.40}$ | $49.71_{\pm 0.12}$ | $51.12_{\pm 0.80}$ | $49.65_{\pm 0.25}$ | $50.81_{\pm 0.19}$ | $49.78_{\pm 0.22}$ |
| PS | $50.20_{\pm 0.58}$ | $49.64_{\pm 0.12}$ | $51.18_{\pm 0.77}$ | $49.16_{\pm 0.28}$ | $50.82_{\pm 0.19}$ | $49.65_{\pm 0.24}$ |
| GS | OOM | OOM | OOM | OOM | OOM | OOM |
| BS | $50.34_{\pm 0.41}$ | $49.73_{\pm 0.13}$ | $51.35_{\pm 0.69}$ | $49.36_{\pm 0.22}$ | $50.89_{\pm 0.19}$ | $49.56_{\pm 0.18}$ |
| BS(w TAM) | $50.34_{\pm 0.48}$ | $49.72_{\pm 0.10}$ | $51.36_{\pm 0.72}$ | $49.98_{\pm 0.26}$ | $50.94_{\pm 0.17}$ | $49.95_{\pm 0.22}$ |
| RN | OOM | OOM | OOM | OOM | OOM | OOM |
| RN(w TAM) | OOM | OOM | OOM | OOM | OOM | OOM |
| GS | OOM | OOM | OOM | OOM | OOM | OOM |
| GS(w TAM) | OOM | OOM | OOM | OOM | OOM | OOM |
| GSR | $50.31_{\pm 0.24}$ | $49.70_{\pm 0.17}$ | $51.31_{\pm 0.41}$ | $49.33_{\pm 0.26}$ | $50.86_{\pm 0.30}$ | $49.53_{\pm 0.20}$ |
| BIM | $50.33_{\pm 0.42}$ | $49.71_{\pm 0.19}$ | $51.35_{\pm 0.60}$ | $49.36_{\pm 0.28}$ | $50.87_{\pm 0.18}$ | $49.56_{\pm 0.23}$ |
| **Ours** | $51.21_{\pm 0.32}$ | $50.65_{\pm 0.32}$ | $51.84_{\pm 0.87}$ | $51.28_{\pm 0.42}$ | $51.34_{\pm 0.32}$ | $51.36_{\pm 0.27}$ |
| Δ | +0.87 | +0.92 | +0.48 | +1.30 | +0.40 | +0.41 |

threshold $\gamma$ in DGIS, as illustrated in Figure 4. Model performance tends to stabilize when $k'$ is sufficiently large, but degrades sharply when $k'$ is too small, likely due to increased pseudo-label noise from coarse clustering. Similarly, performance remains stable at low values of $\gamma$, suggesting that overly conservative filtering risks discarding useful nodes. In contrast, excessively high $\gamma$ introduces substantial noise into the training process.

**Additional Analyses in the Appendix.** Appendix J.2 presents experiments verifying the effectiveness of DPAM, while Appendix J.4 details further analyses on Node-Reordering and DGIS. The variation in RBO similarity across iterations is discussed in Appendix J.3. Moreover, we provide extensive discussion and analyses on geometric imbalance, self-training, and our framework, please refer to the appendix for details.

## 6 Related Work

Class imbalance in graphs poses unique challenges due to the interplay between topology and feature distribution [44, 61, 73, 31, 39, 6, 37, 47, 67]. Existing methods can be broadly categorized into three lines of work: (1) *Node generation methods*, which synthesize nodes to balance class distributions. For example, GraphSMOTE [73] interpolates minority class embeddings and predicts new edges; ImGAGN [39] jointly generates node features and topology by modeling global minority distributions; and GraphENS [37] enhances diversity via ego-network mixing. While effective, these methods often incur high computational cost and struggle to ensure structural consistency across generated nodes. (2) *Topology-aware adjustment methods* exploit structural priors to refine model training. ReNode [6] reweights nodes based on their topological distances to class decision boundaries, while TAM [47] calibrates logits using local topology and class statistics. However, such approaches rely heavily on labeled nodes to infer structural bias and lack a general framework for quantifying or mitigating topological imbalance. (3) *Self-training methods* attempt to exploit unlabeled nodes in imbalanced settings by generating pseudo-labels to supplement minority classes. GraphSR [75] combines similarity-based filtering with reinforcement learning to select reliable pseudo-labeled nodes; BIM [71] frames the task as an influence maximization problem to balance class influence across receptive fields; and IceBerg [27] proposes a Double Balancing mechanism that simultaneously adjusts pseudo-label distributions and loss calibration, while disentangling GNN propagation for better supervision in few-shot regimes. Despite their promise, these methods often rely on heuristic selection strategies and lack theoretical guarantees or standardized metrics for pseudo-label quality, which may lead to instability under severe class imbalance.

## 7 Conclusion and Future Work

In this work, we present a self-training framework that addresses geometric imbalance in graph-structured data under imbalanced settings. By redefining pseudo-labeling and filtering unreliable nodes, our method improves performance on real-world benchmarks and outperforms existing self-training techniques. Future work includes extending the model to imbalance problems in link prediction tasks.

### Acknowledgements

This work was supported by National Natural Science Foundation of China No. U2241212, No. 62276066. W. Ding was supported by the National Natural Science Foundation of China (No. 12471481, U24A2001), the Science and Technology Commission of Shanghai Municipality (No.

23ZR1403000), and the Open Foundation of Key Laboratory Advanced Manufacturing for Optical Systems, CAS (No. KLMSKF202403). Liang Yan would like to thank Xun Qian from Shanghai AI Lab for his very valuable discussions and insightful comments.

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

# Appendix

## Table of Contents

# A   Notations

Table 8: Notation table used throughout the paper, categorized by roles.

| Indices | |
|---|---|
| $n$ | Number of nodes, $n = \|\mathcal{V}\|$. |
| $d$ | Dimension of node features. |
| $C$ | Number of classes. |
| $i, j$ | Indices of labeled and unlabeled nodes. |
| $t$ | Index of self-training iterations. |
| $l$ | Index of GNN/MPNN layers. |
| $v, \hat{v}, u$ | Nodes in the graph. |
| $k'$ | Number of clusters in k-means clustering ($k' > C$). |
| **Parameters** | |
| $\mathcal{G} = (\mathcal{V}, \mathcal{E}, \mathcal{L})$ | Undirected and unweighted graph: nodes $\mathcal{V}$, edges $\mathcal{E}$, and labels $\mathcal{L}$. |
| $\mathcal{X} \in \mathbb{R}^{n \times d}$ | Node feature matrix. |
| $\mathcal{A} \in \{0, 1\}^{n \times n}$ | Adjacency matrix of the graph. |
| $\mathcal{N}(v)$ | Set of 1-hop neighbors of node $v$. |
| $\mathcal{D}^{\text{label}}$ | Labeled node set: $\mathcal{D}^{\text{label}} \subset \mathcal{V}$. |
| $\mathcal{D}^{\text{unlabel}}$ | Unlabeled node set: $\mathcal{V} \setminus \mathcal{D}^{\text{label}}$. |
| $\mathcal{D}_t^{\text{pseudo}}$ | Pseudo-labeled nodes added at iteration $t$. |
| $v^i, u^j$ | Nodes in the labeled/unlabeled datasets. |
| $y^i$ | Ground-truth label of node $v^i$. |
| $\hat{y}^j$ | Predicted label of node $u^j$ by the model. |
| $n_i$ | Number of labeled nodes in class $i$. |
| $\rho$ | Imbalance ratio, $\rho = \frac{\max_i(n_i)}{\min_i(n_i)}$. |
| $e_{v, \hat{v}}$ | Edge weight between nodes $v$ and $\hat{v}$. |
| $\hat{d}_v$ | Degree of node $v$ including self-loop. |
| $\Phi^l$ | Trainable parameter matrix in the $l$-th GCN layer. |
| $h_v^{(l)}$ | Feature of node $v$ at GNN layer $l$. |
| $\tilde{h}_v^{(l)}$ | $\ell_2$-normalized embedding of node $v$ on the unit hypersphere. |
| $\kappa_i$ | Compactness parameter of class $i$ in vMF distribution. |
| $\tilde{\boldsymbol{\mu}}_i$ | Unit mean direction vector of class $i$ on the hypersphere. |
| $C_d(\kappa)$ | Normalization constant in vMF distribution. |
| $p_v^i$ | Posterior probability of node $v$ being in class $i$. |
| $H(u^j)$ | Entropy of prediction over all classes for unlabeled node $u^j$. |
| $\hat{H}$ | Average prediction entropy over all unlabeled nodes. |
| $V_{\text{minor}}$ | Set of nodes with labels in the minority class. |
| $\mathcal{C}_{\text{minor}}$ | Set of minority class indices. |
| $D_{\text{intra}}$ | Intra-class compactness: sum of squared distances to class centers. |
| $D_{\text{inter}}$ | Inter-class separation: sum of squared distances between class centers. |
| $G(u^j)$ | Geometric imbalance score of node $u^j$. |
| $\bar{G}_{\text{minor}}$ | Average geometric imbalance for minority unlabeled nodes. |
| $\delta_u$ | Distance from node $u$ to its closest class centroid. |
| $\beta_u$ | Distance from node $u$ to its second-closest class centroid. |
| $\text{GI}_u$ | Geometric Imbalance index: $\text{GI}_u = \frac{\beta_u - \delta_u}{\delta_u}$. |
| **Functions and Sets** | |
| $f_\theta$ | GNN model parameterized by $\theta$. |
| $m_l(\cdot)$ | Message function at layer $l$. |
| $\theta_l(\cdot)$ | Aggregation function at layer $l$. |
| $\psi_l(\cdot)$ | Node update function at layer $l$. |
| $\mathcal{H}^{\text{label}}, \mathcal{H}^{\text{unlabel}}$ | Embeddings of labeled and unlabeled nodes. |
| $\mathcal{U}_i, \tilde{\mathcal{U}}_i$ | Pseudo-labeled sets from classifier and clustering for class $i$. |
| $\mathcal{U}_i^{\text{final}}$ | Intersection of classifier and cluster predictions for class $i$. |
| $\mathcal{K}_i$ | Cluster $i$ from unsupervised clustering. |
| $\mu_{\mathcal{K}_i}$ | Center of cluster $\mathcal{K}_i$. |
| $\boldsymbol{\mu}_{\mathcal{C}_i}$ | Embedding center of labeled nodes in class $i$. |
| $\mathcal{S}_m, \mathcal{T}_m$ | Geometric and confidence rankings for class $m$. |
| $r_m$ | Rank-Biased Overlap (RBO) score between $\mathcal{S}_m$ and $\mathcal{T}_m$. |
| $\mathcal{N}_m^{\text{New}}$ | Fused node ranking for class $m$ via RBO-weighted fusion. |

## B  Comprehensive Related Works

**Imbalanced Learning in General Machine Learning.** Real-world datasets often exhibit inherent class imbalances, making it challenging to train fair models that avoid bias towards majority classes. To address this issue, a variety of approaches have been developed. Ensemble learning methods [11, 30, 74, 59, 33, 3] combine the outputs of multiple weak classifiers to improve overall model performance on imbalanced data. Data resampling techniques [5, 13, 45, 42, 22, 56] address imbalance by synthesizing or duplicating minority class samples to modify the label distribution in the training set. A third category of solutions involves modifying the loss function, where class-specific weights or adjusted class margins are used to emphasize minority classes during training [76, 51, 4, 50, 65, 40, 57]. Also, post-hoc correction methods [22, 52, 35, 16] aim to mitigate imbalances during the inference stage, after model training is completed. Finally, there is also some work focusing on developing effective generative models [2, 34] under class imbalance situations [38, 66]. While these techniques have demonstrated effectiveness in i.i.d. data scenarios, extending them to graph-structured data introduces unique challenges due to the interdependence between nodes and the graph topology.

**Pseudo-labeling Methods in Graph Learning.** Recent research has predominantly focused on leveraging pseudo-labeling techniques to train graph neural networks (GNNs) when confronted with limited labeled data. Notably, co-training [26] has emerged as a prominent approach that utilizes Parwalks [63] to generate confident pseudo-labels, thereby facilitating the training of GNNs. Conversely, self-training [26] expands the label set by acquiring pseudo-labels from previously trained GNNs. Moreover, the M3S [48] method employs a clustering technique to enhance the accuracy of pseudo-labeling, effectively filtering out labels that do not align with the clustering assignments.

**Semi-supervised Imbalance Node Classification.** Recent advances have tackled the challenge of imbalanced node classification in graphs, with particular emphasis on leveraging topological structures to improve performance [44, 61, 73, 31, 39, 6, 37, 47, 67]. These methods can be broadly grouped into three categories: (1) *Node generation methods*, which synthesize new nodes to balance class distributions; (2) *Topology-aware adjustment methods*, which exploit graph structure to adjust model weights or decision boundaries; and (3) *Self-training methods*, which incorporate pseudo-labeled nodes to enhance generalization. *Node generation methods* include GraphSMOTE [73], which interpolates minority class nodes in the embedding space and constructs edges via link prediction; ImGAGN [39], which generates both features and topology by modeling global minority distributions; and GraphENS [37], which augments diversity by mixing ego-networks. Despite their effectiveness, these approaches often suffer from high computational cost and poor scalability on large graphs. Furthermore, generating semantically meaningful and structurally consistent topology remains a non-trivial, dataset-dependent task. *Topology-aware adjustment methods* reweight or reshape training signals using structural information. For instance, ReNode [6] assigns weights based on nodes' topological distances to decision boundaries, while TAM [47] calibrates prediction logits by integrating local topology and class distribution statistics. However, these methods typically rely on labeled data to estimate structural bias and lack a unified framework for quantifying or mitigating topological imbalance. *Self-training methods* aim to exploit unlabeled data in class-imbalanced settings by generating pseudo-labels. GraphSR [75] adopts a two-step strategy that combines similarity-based filtering with reinforcement learning to select reliable pseudo-labeled nodes for minority classes. BIM [71] formulates the problem through the lens of influence maximization and selects nodes to balance class influence within receptive fields. IceBerg [27] introduces a simple yet effective Double Balancing mechanism that simultaneously calibrates pseudo-label distribution and the loss function, while also disentangling GNN propagation to improve supervision in few-shot scenarios. While promising, these methods still lack theoretical guarantees and standardized metrics for evaluating pseudo-label quality. Their performance often hinges on heuristics—such as confidence thresholds or influence approximations—which can be unreliable under severe class imbalance.

**Topology Bias in Graph Learning.** Topology-aware learning in graphs has recently emerged as a crucial research direction, especially in the context of class-imbalanced node classification. Traditional class rebalancing techniques often overlook structural biases rooted in the non-uniform distribution of labeled nodes across the graph topology. A growing body of work has started to explicitly address this issue, proposing both theoretical formulations and practical remedies for *topological bias*. Renode [6] first formalizes the notion of topology-imbalance, identifying structural

asymmetries among labeled nodes as a new source of bias in semi-supervised node classification. Their method, ReNode, detects node influence conflicts and adaptively re-weights node influence to mitigate the imbalance. Following this, several works proposed augmentation-based solutions. GraphENS [37] generates ego-networks for minor-class nodes to simulate balanced topological contexts, while TAM [47] introduces a topology-aware margin loss that adapts to local connectivity patterns to improve representation learning under imbalance. RGE [46] approaches the problem from the perspective of noisy topology. It introduces a Repulsive edge Group Elimination strategy that selectively removes groups of edges with consistently harmful influence, enhancing robustness against structure-induced bias. Beyond these, [49] proposes PASTEL, a structure learning framework that mitigates topology-imbalance by optimizing node positions to improve intra-class information flow, particularly addressing under-reaching and over-squashing effects. Recent work by [32] challenges the necessity of class rebalancing altogether. They propose BAT, a lightweight topological augmentation technique that mitigates class imbalance by enhancing local structure, without altering label distributions. [72] explores topology-level imbalance at a coarser granularity. They argue that dominant sub-topology groups can bias learning and propose a two-stage method to extract and regulate these groups to ensure more equitable representation. From a metric design perspective, [60] introduces Topological Concentration (TC) to quantify local neighborhood informativeness for link prediction. Their analysis shows how low-TC nodes consistently underperform due to topological constraints, and propose re-weighting schemes to correct this. Meanwhile, [7] unifies topology and class imbalance under the lens of AUC optimization. Their TOPOAUC framework jointly optimizes class-wise margins and topological influence via a TAIL (Topology-Aware Importance Learning) module, offering a practical and effective solution. Finally, [9] proposes neously constructs the graph structure and trains the semi-supervised model, thereby overcoming biases introduced by pre-constructed graphs and addressing topological imbalance in large-scale settings. [58] proposes a Topological Sample Selection (TSS) scheme that promotes informative node selection by leveraging graph position, especially around class boundaries—crucial under noisy labels. [64] designs CE-GSL, a community-entropy based GSL framework that reconstructs the graph by connecting uncertain nodes and enhancing supervision using both class-level and node-level entropy measures. Together, these works establish topology-imbalance as a fundamental limitation in GNN-based learning and offer diverse methodologies—from augmentation and margin-based regularization to influence modeling and graph construction—for addressing it across different learning paradigms.

# C  Proof of Theorem 1

*Proof of Theorem 1 (Message Passing Perspective).* We aim to prove that the average entropy $\hat{H}$ of pseudo-label predictions for unlabeled nodes is positively correlated with intra-class compactness $D_{\text{intra}}$, and inversely correlated with inter-class separation $D_{\text{inter}}$, i.e.,

$$\hat{H} \propto \frac{D_{\text{intra}}}{D_{\text{inter}}}.$$

### Step 1: Message Passing–Induced Representation Shift

Consider a GNN where each node representation is updated via:

$$h_i^{(l+1)} = \sigma \left( W^{(l)} \cdot \left( \sum_{j \in \mathcal{N}(i)} \alpha_{ij} h_j^{(l)} + \alpha_{ii} h_i^{(l)} \right) \right).$$

Assume initial representations $h_i^{(0)} \in \mathbb{R}^d$ are normalized to lie on the unit hypersphere, i.e., $\|h_i^{(0)}\| = 1$. We ignore nonlinearity $\sigma$ and weight matrix $W^{(l)}$ for clarity, so:

$$h_i^{(1)} = \sum_{j \in \mathcal{N}(i)} \alpha_{ij} h_j^{(0)}.$$

Let node $i$ belong to class $c$. The updated representation after one message passing step can be decomposed as:

$$h_i^{(1)} = (1 - \beta_i) \cdot h_i^{(c)} + \beta_i \cdot h_i^{(\bar{c})},$$

where:

- $h_i^{(c)}$ is the contribution from same-class neighbors.

- $h_i^{(\bar{c})}$ is the contribution from different-class neighbors.

- $\beta_i \in [0, 1]$ quantifies the influence of different-class neighbors.

This update shifts $h_i^{(1)}$ away from the class center $\mu_c$, especially when $\beta_i$ is large or when minority class nodes are sparsely connected.

### Step 2: Definition of Geometric Terms

$$D_{\text{intra}} = \sum_{u^j \in V_{\text{minor}} \cap \mathcal{D}^{\text{unlabel}}} \left\| \tilde{h}_{u^j}^{(l)} - \tilde{\mu}_{y^{u^j}} \right\|^2,$$

$$D_{\text{inter}} = \sum_{c_1 \neq c_2} \left\| \tilde{\mu}_{c_1} - \tilde{\mu}_{c_2} \right\|^2.$$

Minority node representations deviate from their true class centers due to influence from majority neighbors, increasing $D_{\text{intra}}$ and potentially decreasing $D_{\text{inter}}$.

### Step 3: Entropy under von Mises–Fisher Posterior

The posterior is:

$$p_i^{u^j} = \frac{\exp\left( \kappa \cdot \cos(\tilde{h}_{u^j}, \tilde{\mu}_i) \right)}{\sum_{k=1}^{C} \exp\left( \kappa \cdot \cos(\tilde{h}_{u^j}, \tilde{\mu}_k) \right)}.$$

Using $\cos(x, y) \approx 1 - \frac{1}{2}\|x - y\|^2$ for unit vectors:

$$p_i^{u^j} \approx \frac{\exp\left( -\frac{\kappa}{2} \|\tilde{h}_{u^j} - \tilde{\mu}_i\|^2 \right)}{Z}, \quad Z = \sum_{k=1}^{C} \exp\left( -\frac{\kappa}{2} \|\tilde{h}_{u^j} - \tilde{\mu}_k\|^2 \right).$$

The entropy is then:

$$H(u^j) = \log Z + \frac{\kappa}{2Z} \sum_{i=1}^{C} \|\tilde{h}_{u^j} - \tilde{\mu}_i\|^2 \cdot e^{-\frac{\kappa}{2}\|\tilde{h}_{u^j} - \tilde{\mu}_i\|^2}.$$

## Step 4: Bounding the Average Entropy

Upper bound: Suppose for most classes $i \neq y^{u^j}$, the distance $\|\tilde{h}_{u^j} - \tilde{\mu}_i\|^2 \approx D_{\text{inter}}/C$, and for the correct class $y^{u^j}$, it is $D_{\text{intra}}$. Then:

$$Z = \sum_{i=1}^{C} \exp\left(-\frac{\kappa}{2} m_i\right)$$

$$= \exp\left(-\frac{\kappa}{2} D_{\text{intra}}\right) + (C-1) \cdot \exp\left(-\frac{\kappa}{2} \cdot \frac{D_{\text{inter}}}{C}\right),$$

Using $\log(1+x) \leq x$ and Taylor expanding:

$$H(u^j) = \log Z + \frac{\kappa}{2Z} \sum_{i=1}^{C} m_i e^{-\frac{\kappa}{2} m_i}$$

$$\leq \log C - \frac{\kappa}{2C^2}(D_{\text{inter}} - D_{\text{intra}})/D_{\text{inter}} + o(1),$$

Averaging gives:

$$\hat{H} \leq \log C - \frac{\kappa}{2C^2} \cdot \left(\frac{D_{\text{inter}} - D_{\text{intra}}}{D_{\text{inter}}}\right).$$

Lower bound: Assume the posterior is highly confident: $p_{y^{u^j}}^{u^j} \approx 1 - \varepsilon$ and $p_{i \neq y^{u^j}}^{u^j} \approx \varepsilon/(C-1)$ with $\varepsilon \ll 1$ due to $\tilde{h}_{u^j}$ being close to $\tilde{\mu}_{y^{u^j}}$.

Using entropy expansion:

$$H(u^j) \approx -(1-\varepsilon)\log(1-\varepsilon) - \varepsilon \log\left(\frac{\varepsilon}{C-1}\right)$$

$$\geq \varepsilon \log(C-1) \geq \frac{\kappa}{2} \cdot \frac{D_{\text{intra}}}{D_{\text{inter}}},$$

where the last inequality holds since $\varepsilon \propto e^{-\kappa(D_{\text{inter}} - D_{\text{intra}})}$ decays with class separation.

So we conclude:

$$\hat{H} \geq \log C - \frac{\kappa}{2} \cdot \left(\frac{D_{\text{intra}}}{D_{\text{inter}}}\right).$$

So, we obtain the sandwich inequality:

$$\log C - \frac{\kappa}{2C^2} \cdot \left(\frac{D_{\text{inter}} - D_{\text{intra}}}{D_{\text{inter}}}\right) \geq \hat{H} \geq \log C - \frac{\kappa}{2} \cdot \left(\frac{D_{\text{intra}}}{D_{\text{inter}}}\right),$$

which implies:

$$\hat{H} \propto \frac{D_{\text{intra}}}{D_{\text{inter}}}.$$

$\square$

# D  Proof of Theorem 2

*Proof of Theorem 2.* Let $\rho = \frac{n_{\text{major}}}{n_{\text{minor}}}$ be the imbalance ratio between majority and minority class nodes. Fix the feature extractor and class centers $\{\tilde{\mu}_c\}_{c=1}^C$.

Recall the geometric imbalance score:

$$G(u^j) := \frac{\left\| \tilde{h}_{u^j} - \tilde{\mu}_{y^{u^j}} \right\|^2}{\sum_{c_1 \neq c_2} \| \tilde{\mu}_{c_1} - \tilde{\mu}_{c_2} \|^2}, \quad \bar{G}_{\text{minor}} := \frac{1}{|V_{\text{minor}} \cap \mathcal{D}^{\text{unlabel}}|} \sum_{u^j} G(u^j).$$

We aim to show that $\bar{G}_{\text{minor}} \propto \rho$. To do this, we analyze the effect of imbalance on the representation shift of minority nodes.

From the message-passing formulation (see Theorem 1), we have:

$$h_i^{(1)} = (1 - \beta_i) \cdot h_i^{(c)} + \beta_i \cdot h_i^{(\bar{c})},$$

where $\beta_i$ is the proportion of neighbors from other classes. Under class imbalance, minority nodes are more likely to have majority neighbors, hence:

$$\mathbb{E}[\beta_i \mid y_i \in \mathcal{C}_{\text{minor}}] \uparrow \text{ with } \rho.$$

Since $h_i^{(1)}$ is a convex combination of class-consistent and inconsistent information, increasing $\beta_i$ pushes $h_i^{(1)}$ away from its own class center $\tilde{\mu}_c$. Specifically, the deviation magnitude satisfies:

$$\|h_i^{(1)} - \tilde{\mu}_c\|^2 \geq \beta_i^2 \cdot \|h_i^{(\bar{c})} - \tilde{\mu}_c\|^2.$$

Therefore, when $\rho$ increases, so does $\beta_i$, and consequently:

$$G(u^j) \propto \|\tilde{h}_{u^j} - \tilde{\mu}_{y^{u^j}}\|^2 \propto \beta_i^2 \propto \rho^2.$$

Averaging over all $u^j \in V_{\text{minor}} \cap \mathcal{D}^{\text{unlabel}}$, and observing that $\sum_{c_1 \neq c_2} \| \tilde{\mu}_{c_1} - \tilde{\mu}_{c_2} \|^2$ is constant under fixed class centers, we obtain:

$$\bar{G}_{\text{minor}} \propto \mathbb{E}[\beta_i^2] \propto \rho^2.$$

Thus, $\bar{G}_{\text{minor}}$ increases monotonically with $\rho$, and we conclude:

$$\bar{G}_{\text{minor}} \propto \rho.$$

$\square$

# E  Visualization and Analysis of Geometric Imbalance

## E.1  Experimental Setup

To demonstrate the phenomenon of geometric imbalance in semi-supervised imbalanced node classification (SINC), we design an illustrative experiment using the Cora dataset. We construct a controlled scenario that isolates the effects of class imbalance on geometric properties of embeddings by manipulating both model architecture and training settings.

- **Initial Pretraining Phase:**
  We begin with a balanced labeled training set where each class contains exactly 50 labeled nodes. On this balanced data, we pretrain two types of encoders:
  - a **Graph Neural Network (GNN)** encoder,
  - a **Multilayer Perceptron (MLP)** encoder.
    These pretrained models serve as feature extractors, capturing representations in a fair, geometry-preserving way.

- **Fine-tuning on Imbalanced Data:**
  We then fine-tune the pretrained models on an imbalanced variant of Cora, where class-wise labeled node counts are set to $[100, 30, 100, 100, 100, 30, 30]$, inducing an imbalance ratio $\rho \approx 3.3$. We consider two fine-tuning settings:
  1. **Frozen (Pretrained):** the encoder parameters are fixed and not updated during fine-tuning.
  2. **Unfrozen (Fine-tuned):** the encoder is fully trainable.

- **Visualization Protocol:**
  To inspect geometric structure, we extract latent node embeddings (from the final hidden layer), project them using t-SNE, and visualize labeled/unlabeled nodes across majority/minority classes.

## E.2  Results and Analysis

Figure 5 and Figure 6 illustrate the change in geometric structure before and after fine-tuning, across both GNN and MLP backbones.

**Case 1 (GNN-Pretrained):**

- The initial GNN encoder produces geometrically well-separated clusters with clear class boundaries.
- Minority class nodes, both labeled and unlabeled, exhibit compact intra-class distributions and are reasonably separated from majority classes.

**Case 2 (GNN-Finetune):**

- After fine-tuning on the imbalanced set, the embeddings show signs of *geometric degradation*:
  - $D_{\text{inter}}$ (distance between class centroids) shrinks significantly.
  - $D_{\text{intra}}$ (dispersion within minority clusters) increases.
- This leads to reduced separability and greater geometric confusion for minority nodes—typical of *geometric imbalance*.

**Case 3 (MLP-Pretrained):**

- The MLP, lacking structural bias from graph topology, preserves geometric structure during pretraining.
- Clusters are reasonably separated and compact, although slightly less so than GNN due to lack of message passing.

**Case 4 (MLP-Finetune):**

- Interestingly, the MLP does *not* suffer from significant degradation after fine-tuning.
  - $D_{\text{inter}}$ remains stable.
  - $D_{\text{intra}}$ does not increase notably.
- This suggests that geometric imbalance is *amplified by message passing in GNNs*, especially when label imbalance exists.

**Conclusion of Analysis:** The results empirically confirm our theoretical claim (Theorem 1 and 2) that:

- Class imbalance leads to greater prediction entropy due to increased $D_{\text{intra}}$ and reduced $D_{\text{inter}}$,
- And that GNNs—due to their structural bias and message propagation—are more vulnerable to this effect than MLPs.

This observation motivates the need to explicitly account for geometric imbalance during self-training and pseudo-label propagation, as we do in the proposed UNREAL framework.

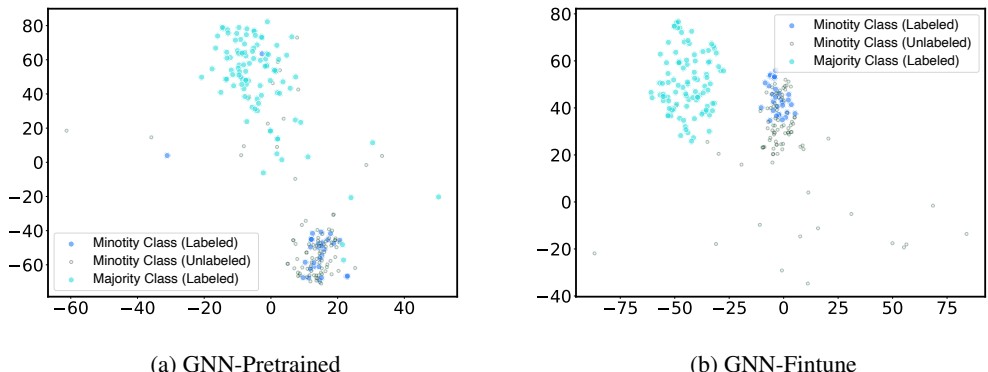

(a) GNN-Pretrained            (b) GNN-Fintune

Figure 5: Illustration of geometric imbalance across different GNN encoder cases.

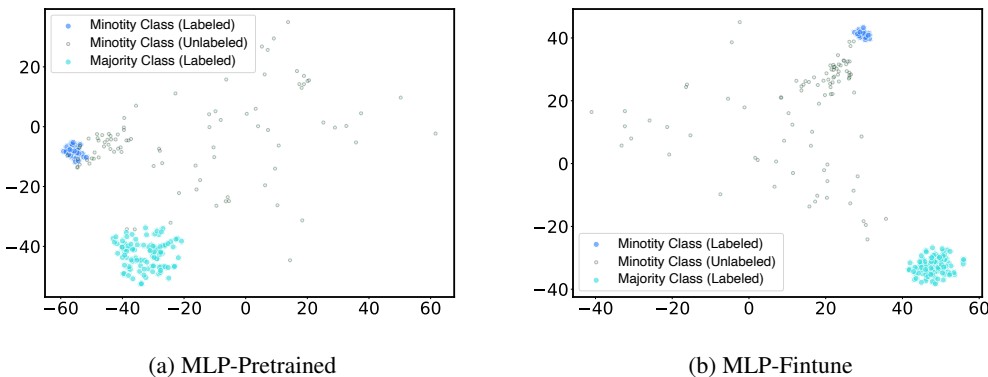

(a) MLP-Pretrained            (b) MLP-Fintune

Figure 6: Illustration of geometric imbalance across different GNN encoder cases.

# F    Analysis of Geometric Imbalance under Varying Class Imbalance

## F.1    Experimental Setup

To comprehensively analyze the effect of class imbalance on geometric properties of node embeddings, we conduct controlled experiments on the **Cora** dataset using three mainstream GNN architectures: **GCN**, **GAT**, and **GraphSAGE**.

For each architecture, we simulate **50 different imbalance scenarios** by systematically varying the class distributions—ranging from balanced to highly imbalanced cases. For every setting, we train the model and extract node embeddings, based on which we compute two key metrics:

- **Geometric Imbalance (GI)**: quantifies the structural disparity of embeddings between minority and majority classes, considering both intra-class compactness and inter-class separation.
- **Average Entropy**: evaluates the uncertainty of the classifier's predictions across all unlabeled nodes.

Additionally, we record the **Imbalance Rate**, defined as the logarithmic ratio between the sample sizes of majority and minority classes.

## F.2    Quantitative Analysis

Figure 7 illustrates the relationship between **Imbalance Rate** and **Geometric Imbalance** across the three architectures. We observe a *monotonic increasing trend*, confirming that higher label imbalance leads to greater geometric distortion in the learned representations. This effect is consistent across GCN, GAT, and GraphSAGE, although the absolute GI values vary slightly depending on the architecture.

Figure 8 explores how **Geometric Imbalance correlates with Average Entropy**. Again, we find a *strong positive correlation*, indicating that as geometric imbalance worsens, model confidence on unlabeled nodes degrades, leading to more uncertain predictions. Notably:

- GCN and GAT exhibit smooth, concave-upward trends, suggesting gradual degradation in certainty.
- GraphSAGE shows a peak in entropy followed by a plateau, possibly due to over-smoothing or reduced model sensitivity at extreme imbalance levels.

These results empirically support the theoretical intuition that geometric imbalance serves as an intermediate variable linking class imbalance to model uncertainty, and underscore the necessity of designing imbalance-aware algorithms to preserve embedding quality and classification reliability.

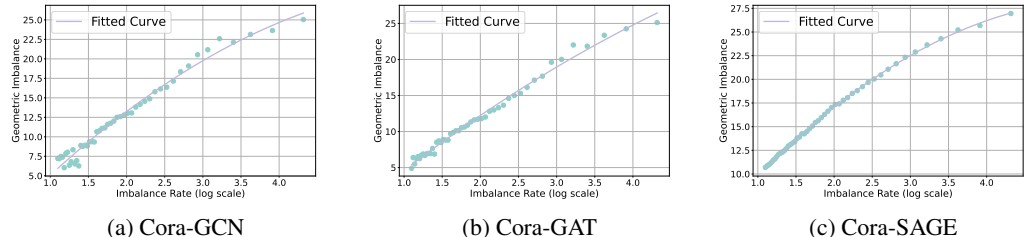

|  (a) Cora-GCN  |  (b) Cora-GAT  |  (c) Cora-SAGE  |

Figure 7: Relationship between Imbalance Rate and Geometric Imbalance across different GNN architectures.

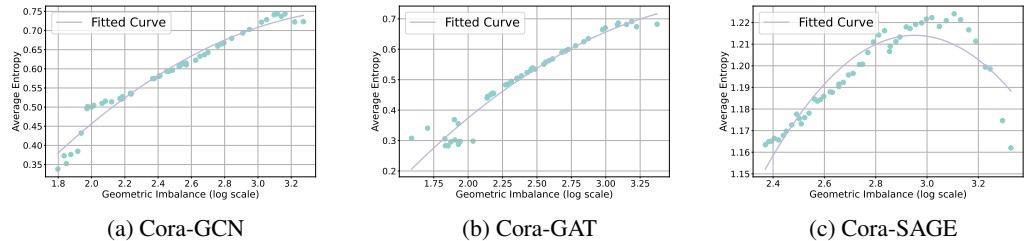

(a) Cora-GCN      (b) Cora-GAT      (c) Cora-SAGE

Figure 8: Relationship between Geometric Imbalance and Average Entropy across different GNN architectures.

# G   Distinction Between Geometric Imbalance and Topology Imbalance

*Geometric imbalance* and *topology imbalance* [6] are two fundamentally distinct forms of imbalance that challenge semi-supervised node classification on graphs. Understanding their differences is crucial for designing effective algorithms for imbalanced graph learning.

## G.1   Geometric Imbalance

Geometric imbalance refers to the ambiguity in node representations within the embedding space induced by class imbalance and GNN message passing mechanisms. Specifically, it characterizes how minority-class nodes, due to limited and biased label propagation, are more likely to have embeddings that are close to the boundaries between classes or even equidistant from multiple class centers on the hypersphere. This phenomenon is rigorously analyzed by modeling embeddings using the von Mises-Fisher (vMF) distribution and quantifying ambiguity via geometric imbalance scores, which measure the relative position of a node's embedding to the centers of all classes. Higher geometric imbalance is directly linked to greater prediction uncertainty and error in pseudo-labeling, particularly for unlabeled minority-class nodes. The severity of geometric imbalance is shown to increase monotonically with the class imbalance ratio, and it is especially pronounced in GNNs due to their structural aggregation process, while being marginal in non-graph models like MLPs [41].

## G.2   Topology Imbalance

In contrast, topology imbalance is defined on the original graph structure and focuses on the asymmetric topological distribution of labeled nodes across classes. Unlike quantity imbalance, which concerns the number of labeled nodes per class, topology imbalance arises when labeled nodes in the same or different classes occupy distinct structural roles—such as being closer or farther from the class boundary or center. This leads to issues like influence conflict (labeled nodes near boundaries causing ambiguous label propagation) and influence insufficiency (labeled nodes far from target regions failing to effectively propagate information). Topology imbalance is quantified via metrics like Totoro, which captures the degree of conflicting influence among labeled nodes across the graph. Its perniciousness lies in shifting the effective decision boundary away from the true class boundary, thereby degrading GNN performance even when the class labeling ratio is balanced [6, 37, 47].

## G.3   Summary of Differences

- **Domain:** Geometric imbalance concerns embedding space positions of (especially un-labeled) nodes, while topology imbalance concerns the graph structure and the relative positions of labeled nodes.

- **Target:** Geometric imbalance primarily affects unlabeled minority nodes by making their embeddings ambiguous, leading to unreliable pseudo-labels. Topology imbalance focuses on the distribution and influence of labeled nodes, particularly those near class boundaries.

- **Measurement:** Geometric imbalance is measured via geometric distances (e.g., on the hypersphere) between node embeddings and class centers. Topology imbalance is measured by influence metrics like Totoro, which consider the network-wide propagation of label information.

- **Mechanism:** Geometric imbalance is amplified by message passing and normalization in GNNs, while topology imbalance arises from the structural arrangement of labeled nodes and their impact on message propagation.

- **Orthogonality:** The two forms of imbalance are orthogonal: one can exist without the other. For example, a graph may have balanced class sizes (no quantity imbalance) but still suffer from severe topology imbalance if labeled nodes are poorly distributed structurally.

In a word, while both geometric and topology imbalance deteriorate node classification performance in semi-supervised learning, they originate from different sources and demand distinct mitigation strategies. Geometric imbalance calls for techniques addressing ambiguity in embedding space, especially for unlabeled nodes, whereas topology imbalance requires careful selection or re-weighting of labeled nodes according to their topological positions in the graph.

# H Proof of Theorem 3

*Proof of Theorem 3.* Recall the construction of the proposed ranking:

$$\mathcal{N}_m^{\text{New}} = \max\{r_m, 1 - r_m\} \cdot \mathcal{S}_m + \min\{r_m, 1 - r_m\} \cdot \mathcal{T}_m, \tag{1}$$

where $r_m = \text{RBO}(\mathcal{S}_m, \mathcal{T}_m)$ measures the similarity between the geometric ranking $\mathcal{S}_m$ and the confidence ranking $\mathcal{T}_m$.

We consider the two extremal regimes of disagreement:

**Case 1:** $r_m \to 0$ **(maximum disagreement).** In this case,

$$\max\{r_m, 1 - r_m\} = 1 - r_m, \quad \min\{r_m, 1 - r_m\} = r_m.$$

Hence,

$$\lim_{r_m \to 0} \mathcal{N}_m^{\text{New}} = \lim_{r_m \to 0} \left( (1 - r_m) \cdot \mathcal{S}_m + r_m \cdot \mathcal{T}_m \right)$$
$$= \mathcal{S}_m.$$

**Case 2:** $r_m \to 1$ **(maximum agreement).** In this case,

$$\max\{r_m, 1 - r_m\} = r_m, \quad \min\{r_m, 1 - r_m\} = 1 - r_m.$$

Thus,

$$\lim_{r_m \to 1} \mathcal{N}_m^{\text{New}} = \lim_{r_m \to 1} \left( r_m \cdot \mathcal{S}_m + (1 - r_m) \cdot \mathcal{T}_m \right)$$
$$= \mathcal{S}_m.$$

In both limiting cases, the ranking $\mathcal{N}_m^{\text{New}}$ converges to the geometric ranking $\mathcal{S}_m$. Therefore, $\mathcal{S}_m$ dominates the selection process particularly in early stages when $r_m$ is small, i.e., when disagreement between geometry and confidence is high. This confirms that the design reduces the effect of geometric imbalance by emphasizing $\mathcal{S}_m$ until $\mathcal{T}_m$ becomes sufficiently aligned. $\square$

# I Comprehensive Experimental Results

## I.1 Comprehensive Results Under Varying Levels of Class Imbalance

Due to space limitations, we present the complete experimental results here. In Table 20, Table 21, Table 22, and Table 23, we report results under different imbalance scenarios ($\rho = 10, 20, 50, 100$). Several consistent trends can be observed across varying imbalance levels.

In Table 20, when $\rho = 10$, our method outperforms all baselines by a clear margin on most datasets. Notably, on *Cora* and *CiteSeer*, our method achieves substantial gains in both balanced accuracy and F1-score. On *PubMed* and *Amazon-Computers*, although the improvements are relatively smaller, our model still ranks among the top performers. We attribute this to two factors: (1) *PubMed* has only three classes, resulting in a less severe imbalance scenario. Most methods can still handle this setting reasonably well. (2) The *Amazon-Computers* dataset has been shown to be less sensitive to label sparsity, which narrows the performance differences among competing methods.

As the imbalance ratio increases (Tables 21 to 23), three key observations emerge:

(1) The performance gap between our method and the baselines consistently increases. For example, at $\rho = 100$, our method outperforms the second-best method by up to **+10.60%** in balanced accuracy and **+7.85%** in F1-score on Cora. This demonstrates that while baseline methods degrade significantly under severe imbalance, our method maintains stable and superior performance.

(2) Oversampling-based strategies such as GraphENS [37] or GraphSMOTE [73], although effective in mild imbalance settings, show limited scalability under higher $\rho$. We hypothesize that generating synthetic nodes and edges may introduce noisy or redundant samples that interfere with representation learning, especially when minority groups are extremely sparse.

(3) BalancedSoftmax [40] performs competitively across all $\rho$ levels by correcting the bias in the classifier head. However, it lacks representation-level adaptability, limiting its potential compared to our method, which additionally improves node representations through geometric ranking and uncertainty-aware selection.

Overall, these results validate the robustness of our method in handling various degrees of class imbalance, outperforming both classical re-weighting techniques and recent augmentation-based approaches.

## I.2 Comprehensive Results on Natural Imbalanced Datasets

The experimental results on the naturally imbalanced datasets *CS-Random* and *Computers-Random* (Table 24, Table 25) further demonstrate the consistent superiority of our method UNREAL. On *CS-Random*, our method achieves the highest balanced accuracy and F1-score across all three GNN architectures, with particularly notable gains on GAT. While methods such as BalancedSoftmax and Renode perform competitively in some cases, they suffer from performance instability across different backbones and fail to generalize under higher imbalance.

On the more challenging *Computers-Random* dataset, Our model delivers even more substantial improvements, especially with GCN and GAT, achieving gains up to **+2.47%** in balanced accuracy and **+6.71** in F1-score. Although the improvements on SAGE are relatively modest—partly due to the dataset's structure enabling easier classification—our method still ranks among the top performers.

These findings confirm that our model is not only effective in synthetic long-tail settings but also robust and generalizable in real-world class-imbalanced scenarios, regardless of architecture choice.

## I.3 Comprehensive Results On Large-Scale Natural Imbalanced Datasets

We further evaluate our method on large-scale real-world datasets with naturally occurring class imbalance: *Flickr* and *Ogbn-Arxiv* (Table 26, Table 27). These datasets present distinct challenges: *Flickr* features moderate imbalance with noisy labels, while *Ogbn-Arxiv* has an extreme imbalance ratio exceeding 700.

On *Flickr*, our method achieves consistent improvements over baselines across all three GNN architectures. Specifically, our method outperforms prior methods by up to **+4.79** in F1-score and **+3.45%** in balanced accuracy. While several augmentation-based methods (e.g., GraphENS, ReNode)

fail due to memory or scalability issues (OOM), our method remains robust and computationally feasible. Notably, on GCN and SAGE, our performance gains are both statistically significant and practically meaningful.

For *Ogbn-Arxiv*, the extreme long-tail label distribution poses severe challenges. Most oversampling or memory-intensive methods cannot run on this scale. In contrast, our method achieves the best overall performance, surpassing all baselines by a small but consistent margin. Despite the relative difficulty of obtaining large improvements under such extreme imbalance, our method shows gains up to **+0.07** in F1-score and **+0.34%** in balanced accuracy, demonstrating its scalability and reliability.

These results confirm the applicability of our method to large-scale, naturally imbalanced benchmarks where many existing methods fail to scale or generalize.

To further validate the scalability and effectiveness of our proposed method, we conduct experiments on the large-scale natural imbalanced dataset Ogbn-Products. Following the same experimental settings as BIM [71], we evaluate our method using three widely adopted graph neural network backbones: GraphSAGE, GAT, and GCN. We report the performance in terms of *F1 score, Accuracy (ACC)*, and *AUC-ROC*. All experiments are repeated five times with different random seeds, and the mean and standard deviation are reported. Tables 9 summarize the results on the ogbn-products dataset. Across all three architectures, our method consistently outperforms the baselines, demonstrating superior robustness and generalization under large-scale natural class imbalance. Compared to BIM, which is specifically designed for imbalanced graph learning, our approach achieves an average improvement of around 2–3% in both F1 score and AUC-ROC across all backbones. These results verify the scalability of our method and its strong empirical effectiveness on large-scale graphs.

Table 9: Experimental results of our method and other baselines on the large-scale natural imbalanced dataset Ogbn-Products. We report averaged Accuracy (ACC, %), F1-score (%), and AUC-ROC (%) with standard deviations over 5 repetitions on three representative GNN architectures.

| Dataset (ogbn-products) | GCN | | | GAT | | | GraphSAGE | | |
|---|---|---|---|---|---|---|---|---|---|
| | F1 | ACC | AUC-ROC | F1 | ACC | AUC-ROC | F1 | ACC | AUC-ROC |
| Vanilla | 57.41±1.18 | 60.12±0.85 | 93.96±0.15 | 59.27±1.08 | 62.10±0.73 | 94.25±0.14 | 58.18±1.21 | 61.03±0.82 | 94.12±0.12 |
| (RN) ReNode | 66.82±0.32 | 67.80±0.44 | 94.75±0.13 | 68.55±0.28 | 69.27±0.36 | 95.01±0.10 | 67.89±0.35 | 68.19±0.41 | 94.86±0.14 |
| BIM | 68.05±0.21 | 68.47±0.23 | 95.41±0.13 | 69.88±0.16 | 70.05±0.21 | 95.63±0.11 | 69.36±0.18 | 69.55±0.19 | 95.51±0.12 |
| **Ours** | **70.71±0.13** | **70.33±0.10** | **96.45±0.12** | **72.04±0.09** | **71.95±0.13** | **96.67±0.10** | **71.85±0.10** | **71.62±0.11** | **96.59±0.14** |

## I.4 Experimental Results of Our Method with Less Training Rounds

We also validated our method on the *Cora* ($\rho = 10$) dataset with fewer iteration rounds (node selections). The experimental results are presented in Table 10, demonstrating that our method outperforms state-of-the-art methods even with a reduced number of rounds.

Table 10: Experimental Results of Our Method under Less Rounds.

| Model (F1) | Cora-GCN | Cora-GAT | Cora-SAGE |
|---|---|---|---|
| GraphENS (*w* TAM) | 72.14 | 70.00 | 70.40 |
| Our method | 76.44 | 75.99 | 73.63 |
| Our method (2 rounds) | 70.62 | 69.32 | 69.83 |
| Our method (4 rounds) | 71.83 | 72.93 | 70.21 |
| Our method (6 rounds) | 73.41 | 73.55 | 71.09 |
| Our method (8 rounds) | 75.63 | 74.54 | 72.90 |

## I.5 Adapting Our Framework to Imbalanced Image Classification

Although our theoretical analysis and methodology primarily address the issue of geometric imbalance in node classification, we have also validated the versatility of our method. For instance, by applying our method within an image classification framework, we conducted experiments on CIFAR100LT using the our method framework in conjunction with ResNet-32. The results is presented in Table 11, and the results demonstrate our method's commendable performance.

Table 11: Recognition results (%) of different training data. All configurations are evaluated on the testing set of normal CIFAR100.

| Model | Accuracy | Precison | Recall |
|---|---|---|---|
| ResNet-32 (CIFAR100LT) | 37.74 | 42.12 | 37.54 |
| **+ Our method** | **39.31** | **44.32** | **40.43** |

Table 12: Refined running time analysis of vanilla GNN models and our method. (a) Training over 1000 epochs, and (b) selecting 100 nodes into the training set. Results are averaged over five runs.

| Model/Time (1000 epochs) | Cora | CiteSeer | PubMed | Model/Time (Selecting 100 Nodes) | Cora | CiteSeer | PubMed |
|---|---|---|---|---|---|---|---|
| Vanilla (GCN) | 6.221(s) | 7.212(s) | 10.771(s) | Vanilla Self-Training (GCN) | 2.244(s) | 2.234(s) | 2.675(s) |
| **Ours (GCN)** | **6.208(s)** | **7.191(s)** | **10.858(s)** | **Ours (GCN)** | **2.367(s)** | **2.305(s)** | **2.790(s)** |
| Vanilla (GAT) | 8.201(s) | 9.574(s) | 14.443(s) | Vanilla Self-Training (GAT) | 2.624(s) | 2.520(s) | 2.431(s) |
| **Ours (GAT)** | **8.218(s)** | **9.517(s)** | **15.088(s)** | **Ours (GAT)** | **2.697(s)** | **2.670(s)** | **2.411(s)** |
| Vanilla (SAGE) | 6.522(s) | 11.170(s) | 15.475(s) | Vanilla Self-Training (SAGE) | 2.424(s) | 2.780(s) | 2.032(s) |
| **Ours (SAGE)** | **6.452(s)** | **11.256(s)** | **14.658(s)** | **Ours (SAGE)** | **2.461(s)** | **2.827(s)** | **2.115(s)** |

## I.6 Analysis of Computational Complexity

**Time Complexity Analysis.** Our framework, as described in Algorithm 1 (see Appendix L), consists of three main modules per round: Dual-Path Pseudo-label Alignment (DPAM), Node-Reordering (NR), and Discarding Geometrically Imbalanced Nodes (DGIS). (1) The dominant computational cost arises from the DPAM step, which clusters the embeddings of unlabeled nodes using k-means (complexity $\mathcal{O}(n_u k' d T_{iter})$, with $n_u$ unlabeled nodes, $k'$ clusters, $d$-dimensional embeddings, and $T_{iter}$ k-means iterations). Computing class centroids and intersecting with classifier-predicted labels are both linear-time operations. (2) NR involves ranking nodes by geometric distance and classifier confidence (each $\mathcal{O}(n_u \log n_u)$), with ranking fusion being negligible. (3) DGIS computes distances from candidate nodes to their closest and second-closest centroids ($\mathcal{O}(n_u C d)$) per round, where $C$ is the number of classes. (4) Overall, the per-round complexity is primarily determined by k-means, but since $k' \ll n_u$ and k-means can be efficiently implemented (e.g., on GPU), so, the total complexity remains comparable to standard self-training pipelines. Memory usage is linear in the number of nodes, as we only store embeddings and label predictions. Importantly, our method is significantly more efficient than prior over-sampling or augmentation approaches such as GraphENS and GraphSMOTE, which may have $\mathcal{O}(n^2)$ or worse complexity on large graphs. In summary, the only notable overhead beyond vanilla GNN training comes from clustering and ranking, both of which are scalable and can be accelerated by modern hardware.

**Empirical Scalability and Runtime.** To address practical efficiency, we conducted comprehensive scalability analyses on both medium- and large-scale datasets, comparing our method ("Ours") to vanilla GNN backbones and representative baselines. (1) On Cora, CiteSeer, and PubMed, the training time per epoch for our method is almost identical to the backbone GNNs (see Table 12), confirming the negligible overhead of our extra modules (clustering and ranking). (2) On large-scale graphs (e.g., Flickr, ogbn-arxiv), our method remains tractable and runs smoothly on a single RTX 3090 GPU (Tables 26 and Table 27). By contrast, prior data augmentation and oversampling baselines (e.g., GraphSMOTE, GraphENS) either become prohibitively slow or run out-of-memory, as synthesizing large numbers of new nodes/edges is expensive.

## I.7 Comparison of UNREAL and Recent Soft Pseudo-Labeling Methods

**Experimental Setup.** Here, we compare UNREAL with soft pseudo-labeling strategies such as SCR [70] and ConsisGAD [8]. We have conducted comprehensive experiments comparing our method with these approaches under class-imbalanced settings. For all datasets, we followed the initial hyperparameter search ranges provided in Appendix A.3 of the SCR paper and Appendix D.2 of the ConsisGAD paper. We further expanded these ranges as needed, and employed Bayesian optimization (using Wandb) to efficiently search for the optimal hyperparameters for both methods

Table 13: Results with $\rho = 10$ (GCN) on three class-imbalanced node classification benchmark datasets for comparison of UNREAL and Recent Soft Pseudo-Labeling Methods.

| Method | Cora bAcc. | Cora F1 | CiteSeer bAcc. | CiteSeer F1 | PubMed bAcc. | PubMed F1 |
|---|---|---|---|---|---|---|
| GSR | 70.85±0.44 | 71.37±0.63 | 59.28±0.72 | 55.96±0.95 | 73.61±1.25 | 71.88±1.33 |
| BIM | 72.19±0.42 | 72.67±0.48 | 58.54±0.61 | 56.81±0.98 | 74.62±1.15 | 72.93±1.21 |
| **Ours** | **78.33±1.04** | **76.44±1.06** | **65.63±1.38** | **64.94±1.38** | **75.35±1.41** | **73.65±1.43** |
| SCR | 66.28±0.57 | 65.43±0.91 | 44.92±1.10 | 39.12±1.59 | 67.88±0.67 | 62.70±1.89 |
| ConsisGAD | 66.50±0.59 | 65.61±0.90 | 44.79±1.09 | 39.33±1.61 | 68.05±0.62 | 62.75±1.90 |

Table 14: Results with $\rho = 100$ (SAGE) on three class-imbalanced node classification benchmark datasets for comparison of UNREAL and Recent Soft Pseudo-Labeling Methods.

| Method | Cora bAcc. | Cora F1 | CiteSeer bAcc. | CiteSeer F1 | PubMed bAcc. | PubMed F1 |
|---|---|---|---|---|---|---|
| GSR | 66.45±2.10 | 64.42±1.83 | 53.52±1.47 | 53.01±1.75 | 74.09±2.12 | 72.97±1.90 |
| BIM | 67.75±2.13 | 64.68±1.95 | 53.83±1.62 | 53.29±1.80 | 74.38±2.08 | 73.24±1.85 |
| **Ours** | **73.47±2.31** | **68.30±2.11** | **59.77±2.98** | **58.92±3.07** | **77.11±0.59** | **74.03±0.81** |
| SCR | 59.31±2.91 | 55.34±4.32 | 36.51±1.29 | 27.19±2.84 | 63.41±0.76 | 54.97±1.64 |
| ConsisGAD | 59.63±2.89 | 55.18±4.43 | 36.39±1.28 | 27.28±2.87 | 63.27±0.74 | 55.33±1.60 |

on the various imbalanced datasets. This ensured that both SCR and ConsisGAD were evaluated under their best possible configurations in our experiments.

**Results and Analysis.** Empirical results (see Table 13-17) show that both SCR and ConsisGAD experience a significant performance drop under severe class imbalance, particularly in terms of Macro-F1. In contrast, our method consistently achieves much stronger performance in these challenging scenarios. We believe this is due to the following reasons: (1) Soft pseudo-labeling methods, including SCR and ConsisGAD, propagate pseudo-label information uniformly across all nodes with global consistency constraints. In imbalanced graphs, the majority classes dominate the label distribution and, consequently, the consistency regularization process, causing minority class signals to be overwhelmed or ignored. This leads to suboptimal decision boundaries that do not adequately account for the geometric marginalization of minority nodes. (2) In contrast, our method is specifically motivated by the phenomenon of geometric imbalance on non-Euclidean manifolds, and is designed to selectively generate reliable pseudo-labels for ambiguous nodes. By doing so, we explicitly enhance the representation learning for minority classes and avoid overfitting to majority class signals.

### I.8 Experimental Results on Imbalanced Heterophilous Graphs.

To further evaluate the generalization capability of our method, we conduct experiments on heterophilous graph benchmarks, including *Chameleon*, *Squirrel*, and *Wisconsin*. Unlike homophilic graphs, where neighboring nodes tend to share similar labels, heterophilous graphs pose additional challenges due to the weak correlation between node features and their adjacency structure.

As shown in Table 18, our model consistently outperforms existing baselines on all three datasets. This demonstrates that our approach not only handles class imbalance effectively but also generalizes well to complex graph topologies where conventional message-passing schemes tend to fail. The improvement observed across both balanced accuracy and F1 score indicates that the proposed model achieves a better trade-off between precision and recall under highly non-homophilic conditions.

Furthermore, the performance gain highlights the robustness of our method's representation learning mechanism. By dynamically adjusting pseudo-label confidence and leveraging relational consistency, our approach mitigates the over-smoothing and noise propagation that commonly affect GNNs on heterophilous graphs. These results verify that our framework remains effective even when node similarity is low, suggesting its potential for broader application scenarios such as citation, social, and web networks characterized by heterophily.

Table 15: Experimental results on Computers-Random ($\rho \approx 17.7$) for comparison of UNREAL and Recent Soft Pseudo-Labeling Methods.

| Model | GCN bAcc. | GCN F1 | GAT bAcc. | GAT F1 | SAGE bAcc. | SAGE F1 |
|---|---|---|---|---|---|---|
| GSR | 83.82±0.74 | 77.78±0.42 | 76.79±0.68 | 73.61±0.63 | 77.63±0.32 | 72.56±0.51 |
| BIM | 84.03±0.73 | 77.96±0.45 | 77.01±0.70 | 73.82±0.60 | 77.76±0.65 | 72.09±0.37 |
| **Ours** | **85.32±0.22** | **80.43±0.56** | **82.52±0.35** | **78.90±0.38** | 75.81±1.86 | **71.86±1.86** |
| SCR | 80.61±0.47 | 75.18±0.57 | 71.90±0.82 | 70.71±0.53 | 66.41±1.45 | 66.05±1.44 |
| ConsisGAD | 80.56±0.46 | 75.15±0.56 | 71.97±0.79 | 70.62±0.54 | 66.54±1.49 | 66.13±1.48 |

Table 16: Experimental results on CS-Random ($\rho \approx 41.0$) for comparison of UNREAL and Recent Soft Pseudo-Labeling Methods.

| Model | GCN bAcc. | GCN F1 | GAT bAcc. | GAT F1 | SAGE bAcc. | SAGE F1 |
|---|---|---|---|---|---|---|
| GSR | 86.73±0.22 | 85.91±0.21 | 85.34±0.13 | 86.56±0.29 | 85.44±0.27 | 86.46±0.23 |
| BIM | 86.89±0.23 | 85.99±0.21 | 85.63±1.87 | 86.65±0.35 | 85.65±0.28 | 86.73±0.22 |
| **Ours** | **88.94±0.09** | **89.87±0.06** | **87.65±0.12** | **87.65±0.11** | **88.03±0.21** | **88.65±0.07** |
| SCR | 87.41±0.18 | 88.73±0.11 | 83.61±0.41 | 84.68±0.33 | 85.80±0.23 | 87.37±0.18 |
| ConsisGAD | 87.44±0.16 | 88.68±0.12 | 83.58±0.38 | 84.75±0.34 | 85.73±0.25 | 87.30±0.15 |

## I.9 Comparison to General Self-Training Methods

We also compared our method with other self-training frameworks on the *CS-Random* dataset, as shown in Table 19. By explicitly addressing hard samples in the representation space, our method achieves superior performance compared to existing frameworks.

Table 17: Experimental results on Ogban-arXiv ($\rho \approx 775.4$) for comparison of UNREAL and Recent Soft Pseudo-Labeling Methods.

| Model | GCN bAcc. | GCN F1 | GAT bAcc. | GAT F1 | SAGE bAcc. | SAGE F1 |
|---|---|---|---|---|---|---|
| GSR | 50.31±0.24 | 49.70±0.17 | 51.31±0.41 | 49.33±0.26 | 50.86±0.30 | 49.53±0.20 |
| BIM | 50.33±0.42 | 49.71±0.20 | 49.36±0.28 | 50.87±0.18 | 49.56±0.23 | 50.21±0.26 |
| **Ours** | **51.21±0.36** | **50.32±0.32** | **51.84±0.27** | **51.78±0.24** | **51.34±0.32** | **51.36±0.27** |
| SCR | 50.23±0.41 | 49.69±0.13 | 51.10±0.21 | 49.67±0.26 | 50.83±0.20 | 49.80±0.23 |
| ConsisGAD | 50.25±0.39 | 49.72±0.14 | 51.13±0.37 | 49.63±0.27 | 50.80±0.19 | 49.77±0.22 |

Table 18: Results on heterophilous graphs. We report balanced accuracy (bAcc.) and F1 scores on three benchmark datasets: Chameleon, Squirrel, and Wisconsin.

| Method | Chameleon bAcc. | Chameleon F1 | Squirrel bAcc. | Squirrel F1 | Wisconsin bAcc. | Wisconsin F1 |
|---|---|---|---|---|---|---|
| GSR(GraphSR) | 38.45±0.75 | 37.67±0.72 | 27.94±0.44 | 27.01±0.32 | 31.54±2.30 | 28.91±2.05 |
| BIM | 38.26±0.68 | 37.48±0.66 | 27.83±0.46 | 27.16±0.37 | 31.59±2.25 | 28.79±2.01 |
| **Ours** | **43.59±0.60** | **42.24±0.63** | **30.04±0.48** | **29.12±0.46** | **46.22±3.51** | **41.52±4.05** |

Table 19: Comparison with General Self-Training Frameworks.

| Model (F1) | CS-GCN | CS-GAT | CS-SAGE |
|---|---|---|---|
| Self-Training [26] | 87.54 | 85.32 | 86.54 |
| Co-Training [26] | 87.56 | 85.32 | 87.21 |
| M3S [48] | 88.12 | 86.54 | 87.43 |
| **Our method** | 89.87 | 87.65 | 88.65 |

Table 20: Experimental results of our method and other baselines on four class-imbalanced node classification benchmark datasets with $\rho = 10$. We report averaged balanced accuracy (bAcc.,%) and F1-score (%) with the standard errors over 5 repetitions on three representative GNN architectures.

| Dataset | Cora | | CiteSeer | | PubMed | | Amazon-Computers | |
|---|---|---|---|---|---|---|---|---|
| Imbalance Ratio ($\rho = 10$) | bAcc. | F1 | bAcc. | F1 | bAcc. | F1 | bAcc. | F1 |
| **GCN** | | | | | | | | |
| Vanilla | 62.82 ± 1.43 | 61.67 ± 1.59 | 38.72 ± 1.88 | 28.74 ± 3.21 | 65.64 ± 1.72 | 56.97 ± 3.17 | 80.01 ± 0.71 | 71.56 ± 0.81 |
| Re-Weight | 65.36 ± 1.15 | 64.97 ± 1.39 | 44.69 ± 1.78 | 38.61 ± 2.37 | 69.06 ± 1.84 | 64.08 ± 2.97 | 80.93 ± 1.30 | 73.99 ± 2.20 |
| PC Softmax | 68.04 ± 0.82 | 67.84 ± 0.81 | 50.18 ± 0.55 | 46.14 ± 0.14 | 72.46 ± 0.80 | 70.27 ± 0.94 | 81.54 ± 0.76 | 73.30 ± 0.51 |
| GraphSMOTE | 66.39 ± 0.56 | 65.49 ± 0.93 | 44.87 ± 1.12 | 39.20 ± 1.62 | 67.91 ± 0.64 | 62.68 ± 1.92 | 79.48 ± 0.47 | 72.63 ± 0.76 |
| BalancedSoftmax | 69.98 ± 0.58 | 68.68 ± 0.55 | 55.52 ± 0.97 | 53.74 ± 1.42 | 73.73 ± 0.89 | 71.53 ± 1.06 | 81.46 ± 0.74 | 74.31 ± 0.51 |
| BalancedSoftmax (*w* TAM) | 69.94 ± 0.45 | 69.54 ± 0.47 | 56.73 ± 0.71 | 56.15 ± 0.78 | 74.62 ± 0.97 | 72.25 ± 1.30 | 82.36 ± 0.67 | 72.94 ± 1.43 |
| Renode | 67.03 ± 1.41 | 67.16 ± 1.67 | 43.47 ± 2.22 | 37.52 ± 3.10 | 71.40 ± 1.42 | 67.27 ± 2.96 | 81.89 ± 0.77 | 73.13 ± 1.60 |
| Renode (*w* TAM) | 68.26 ± 1.84 | 68.11 ± 1.97 | 46.20 ± 1.17 | 39.96 ± 2.76 | 72.63 ± 2.03 | 68.28 ± 3.30 | 80.36 ± 1.19 | 72.51 ± 0.68 |
| GraphENS | 70.89 ± 0.71 | 70.90 ± 0.81 | 56.57 ± 0.98 | 55.29 ± 1.33 | 72.13 ± 1.04 | 70.72 ± 1.07 | 82.40 ± 0.39 | 74.26 ± 1.05 |
| GraphENS (*w* TAM) | 71.69 ± 0.36 | 72.14 ± 0.53 | 58.01 ± 0.68 | 56.32 ± 1.03 | 74.14 ± 1.42 | 72.42 ± 1.39 | 81.02 ± 0.99 | 70.78 ± 1.72 |
| GraphSR | 70.85 ± 0.44 | 71.37 ± 0.63 | 59.28 ± 0.72 | 55.96 ± 0.95 | 73.61 ± 1.25 | 71.88 ± 1.33 | 83.09 ± 0.29 | 72.03 ± 0.98 |
| BIM | 72.19 ± 0.42 | 72.67 ± 0.48 | 58.54 ± 0.61 | 56.81 ± 0.98 | 74.62 ± 1.15 | 72.93 ± 1.21 | 82.34 ± 0.21 | 72.32 ± 0.32 |
| **Ours** | **78.33 ± 1.04** | **76.44 ± 1.06** | **65.63 ± 1.38** | **64.94 ± 1.38** | **75.35 ± 1.41** | **73.65 ± 1.43** | **85.08 ± 0.38** | **75.27 ± 0.23** |
| **Δ** | **+6.14** | **+3.77** | **+6.35** | **+8.13** | **+0.73** | **+0.72** | **+1.99** | **+0.96** |
| **GAT** | | | | | | | | |
| Vanilla | 62.33 ± 1.56 | 61.82 ± 1.84 | 38.84 ± 1.13 | 31.25 ± 1.64 | 64.60 ± 1.64 | 55.24 ± 2.80 | 79.04 ± 1.60 | 70.00 ± 2.50 |
| Re-Weight | 66.87 ± 0.97 | 66.62 ± 1.13 | 45.47 ± 2.35 | 40.60 ± 2.98 | 68.10 ± 2.85 | 63.76 ± 3.54 | 80.38 ± 0.66 | 69.99 ± 0.76 |
| PC Softmax | 66.69 ± 0.79 | 66.04 ± 1.21 | 50.78 ± 1.66 | 48.56 ± 2.08 | 72.88 ± 0.83 | 71.09 ± 0.89 | 79.43 ± 0.94 | 71.33 ± 0.86 |
| GraphSMOTE | 66.71 ± 0.32 | 65.01 ± 1.21 | 45.68 ± 0.93 | 38.96 ± 0.97 | 67.43 ± 1.23 | 61.97 ± 2.54 | 79.38 ± 1.97 | 69.76 ± 2.31 |
| BalancedSoftmax | 67.89 ± 0.36 | 67.96 ± 0.41 | 54.78 ± 1.25 | 51.83 ± 2.11 | 72.30 ± 1.20 | 69.30 ± 1.79 | 82.02 ± 1.19 | 72.94 ± 1.54 |
| BalancedSoftmax (*w* TAM) | 69.16 ± 0.27 | 69.39 ± 0.37 | 56.30 ± 1.25 | 53.87 ± 1.14 | 73.50 ± 1.24 | 71.36 ± 1.99 | 75.54 ± 2.09 | 66.69 ± 1.44 |
| Renode | 67.33 ± 0.79 | 68.08 ± 1.16 | 44.48 ± 2.06 | 37.93 ± 2.87 | 69.93 ± 2.10 | 65.27 ± 2.90 | 76.01 ± 1.08 | 66.72 ± 1.42 |
| Renode (*w* TAM) | 67.50 ± 0.67 | 68.06 ± 0.96 | 45.12 ± 1.41 | 39.29 ± 1.79 | 70.66 ± 2.13 | 66.94 ± 3.54 | 74.30 ± 1.13 | 66.13 ± 1.75 |
| GraphENS | 70.45 ± 1.25 | 69.87 ± 1.32 | 51.45 ± 1.28 | 47.98 ± 2.08 | 73.15 ± 1.24 | 71.90 ± 1.03 | 81.23 ± 0.74 | 71.23 ± 0.42 |
| GraphENS (*w* TAM) | 70.15 ± 0.18 | 70.00 ± 0.40 | 56.15 ± 1.13 | 54.31 ± 1.68 | 73.45 ± 1.07 | 72.10 ± 0.36 | 81.07 ± 1.03 | 71.27 ± 1.98 |
| GraphSR | 70.86 ± 0.22 | 70.61 ± 0.38 | 56.85 ± 1.09 | 55.02 ± 1.55 | 74.18 ± 1.01 | 72.65 ± 0.33 | 81.72 ± 1.00 | 71.91 ± 1.87 |
| BIM | 71.53 ± 0.20 | 71.34 ± 0.36 | 57.54 ± 1.02 | 55.76 ± 1.48 | 73.91 ± 0.97 | 72.54 ± 0.35 | 82.48 ± 0.96 | 72.58 ± 1.81 |
| **Ours** | **78.91 ± 0.59** | **75.99 ± 0.47** | **64.10 ± 1.49** | **63.44 ± 1.47** | **74.68 ± 1.43** | **72.78 ± 0.89** | **85.62 ± 0.44** | **75.34 ± 0.99** |
| **Δ** | **+7.38** | **+4.65** | **+6.56** | **+7.68** | **+0.50** | **+0.13** | **+3.14** | **+2.40** |
| **SAGE** | | | | | | | | |
| Vanilla | 61.82 ± 0.97 | 60.97 ± 1.07 | 43.18 ± 0.52 | 36.66 ± 1.25 | 68.68 ± 1.51 | 64.16 ± 2.38 | 72.36 ± 2.39 | 64.32 ± 2.21 |
| Re-Weight | 63.94 ± 1.07 | 63.82 ± 1.30 | 46.17 ± 1.32 | 40.13 ± 1.68 | 69.89 ± 1.60 | 65.71 ± 2.31 | 76.08 ± 1.14 | 65.76 ± 1.40 |
| PC Softmax | 65.79 ± 0.70 | 66.04 ± 0.92 | 50.66 ± 0.99 | 47.48 ± 1.66 | 71.49 ± 0.94 | 70.23 ± 0.67 | 74.63 ± 3.01 | 66.44 ± 4.04 |
| GraphSMOTE | 61.65 ± 0.34 | 60.97 ± 0.98 | 42.73 ± 2.87 | 35.18 ± 1.75 | 66.63 ± 0.65 | 61.97 ± 2.54 | 71.85 ± 0.98 | 68.92 ± 0.73 |
| BalancedSoftmax | 67.43 ± 0.61 | 67.66 ± 0.69 | 51.74 ± 2.32 | 49.01 ± 3.16 | 71.36 ± 1.37 | 69.66 ± 1.81 | 73.67 ± 1.11 | 65.23 ± 2.44 |
| BalancedSoftmax (*w* TAM) | 69.03 ± 0.92 | 69.03 ± 0.97 | 51.93 ± 2.19 | 48.67 ± 3.25 | 72.28 ± 1.47 | 71.02 ± 1.31 | 77.00 ± 2.93 | 70.85 ± 2.28 |
| Renode | 66.84 ± 1.78 | 67.08 ± 1.75 | 48.65 ± 1.37 | 44.25 ± 2.20 | 71.37 ± 1.33 | 67.78 ± 1.38 | 77.37 ± 0.74 | 68.42 ± 1.81 |
| Renode (*w* TAM) | 67.28 ± 1.11 | 67.15 ± 1.11 | 48.39 ± 1.76 | 43.56 ± 2.31 | 71.25 ± 1.07 | 68.69 ± 0.98 | 74.87 ± 2.25 | 66.87 ± 2.52 |
| GraphENS | 68.74 ± 0.46 | 68.34 ± 0.33 | 53.51 ± 0.78 | 51.42 ± 1.19 | 70.97 ± 0.78 | 70.00 ± 1.22 | 82.57 ± 0.50 | 71.95 ± 0.51 |
| GraphENS (*w* TAM) | 70.45 ± 0.74 | 70.40 ± 0.75 | 54.69 ± 1.12 | 53.56 ± 1.86 | 73.61 ± 1.35 | 72.50 ± 1.58 | 82.17 ± 0.93 | 72.46 ± 1.00 |
| GraphSR | 69.24 ± 0.42 | 68.82 ± 0.36 | 53.98 ± 0.74 | 51.92 ± 1.10 | 71.43 ± 0.75 | 70.46 ± 1.15 | 82.97 ± 0.48 | 72.34 ± 0.55 |
| BIM | 70.59 ± 0.71 | 70.55 ± 0.72 | 54.83 ± 1.08 | 53.71 ± 1.78 | 73.75 ± 1.30 | 72.66 ± 1.52 | 82.31 ± 0.91 | 72.61 ± 0.98 |
| **Ours** | **75.99 ± 0.98** | **73.63 ± 1.23** | **66.45 ± 0.39** | **65.83 ± 0.30** | **74.78 ± 1.30** | **72.80 ± 0.54** | **83.21 ± 1.50** | **70.81 ± 1.70** |
| **Δ** | **+5.40** | **+3.08** | **+11.62** | **+12.12** | **+1.03** | **+0.14** | **+0.24** | **-1.65** |

Table 21: Experimental results of our method and other baselines on four class-imbalanced node classification benchmark datasets with $\rho = 20$. We report averaged balanced accuracy (bAcc.,%) and F1-score (%) with the standard errors over 5 repetitions on three representative GNN architectures.

| | Dataset | Cora | | CiteSeer | | PubMed | | Amazon-Computers | |
|---|---|---|---|---|---|---|---|---|---|
| | Imbalance Ratio ($\rho = 20$) | bAcc. | F1 | bAcc. | F1 | bAcc. | F1 | bAcc. | F1 |
| **GCN** | Vanilla | $53.20 \pm 0.88$ | $47.81 \pm 1.23$ | $35.32 \pm 0.15$ | $21.81 \pm 0.12$ | $61.13 \pm 0.35$ | $46.85 \pm 0.76$ | $72.34 \pm 2.92$ | $65.42 \pm 3.00$ |
| | Re-Weight | $57.51 \pm 1.05$ | $54.63 \pm 1.08$ | $36.99 \pm 1.79$ | $27.33 \pm 2.32$ | $66.52 \pm 2.42$ | $58.22 \pm 3.65$ | $72.45 \pm 2.06$ | $65.85 \pm 1.46$ |
| | PC Softmax | $61.74 \pm 1.50$ | $60.55 \pm 1.97$ | $42.53 \pm 1.53$ | $36.54 \pm 1.13$ | $68.26 \pm 1.99$ | $66.54 \pm 1.87$ | $73.84 \pm 2.64$ | $66.32 \pm 2.97$ |
| | BalancedSoftmax | $64.06 \pm 0.74$ | $62.88 \pm 0.86$ | $47.29 \pm 1.29$ | $44.08 \pm 1.71$ | $69.71 \pm 1.74$ | $68.31 \pm 1.71$ | $76.92 \pm 2.01$ | $69.86 \pm 1.99$ |
| | BalancedSoftmax (*w* TAM) | $64.75 \pm 0.54$ | $63.46 \pm 0.72$ | $48.52 \pm 1.62$ | $46.38 \pm 1.79$ | $69.95 \pm 2.09$ | $68.90 \pm 1.86$ | $77.09 \pm 2.02$ | $69.86 \pm 1.76$ |
| | Renode | $59.40 \pm 1.00$ | $56.88 \pm 1.52$ | $38.25 \pm 1.60$ | $27.61 \pm 2.25$ | $67.45 \pm 3.34$ | $60.40 \pm 5.74$ | $74.15 \pm 1.72$ | $67.27 \pm 0.92$ |
| | Renode (*w* TAM) | $59.88 \pm 1.16$ | $58.05 \pm 1.66$ | $41.11 \pm 2.45$ | $31.58 \pm 2.62$ | $68.53 \pm 3.53$ | $64.82 \pm 4.32$ | $73.46 \pm 1.77$ | $67.50 \pm 1.18$ |
| | GraphENS | $67.30 \pm 1.45$ | $66.82 \pm 1.40$ | $46.39 \pm 3.48$ | $42.38 \pm 4.14$ | $71.37 \pm 1.77$ | $69.37 \pm 1.69$ | $75.41 \pm 1.75$ | $69.32 \pm 1.58$ |
| | GraphENS (*w* TAM) | $66.94 \pm 1.38$ | $66.67 \pm 1.42$ | $48.80 \pm 2.98$ | $45.06 \pm 4.16$ | $71.92 \pm 1.58$ | $69.35 \pm 1.88$ | $75.78 \pm 1.57$ | $68.58 \pm 1.78$ |
| | GraphSR | $67.98 \pm 1.42$ | $67.53 \pm 1.36$ | $47.03 \pm 3.40$ | $43.06 \pm 4.06$ | $72.05 \pm 1.72$ | $70.01 \pm 1.64$ | $75.97 \pm 1.70$ | $69.96 \pm 1.54$ |
| | BIM | $67.94 \pm 1.32$ | $67.51 \pm 1.26$ | $46.98 \pm 3.26$ | $42.91 \pm 3.95$ | $72.05 \pm 1.68$ | $69.98 \pm 1.52$ | $76.04 \pm 1.61$ | $69.91 \pm 1.44$ |
| | **Ours** | $\mathbf{77.02 \pm 0.75}$ | $\mathbf{74.15 \pm 0.87}$ | $\mathbf{55.81 \pm 6.11}$ | $\mathbf{55.19 \pm 6.23}$ | $\mathbf{73.06 \pm 1.87}$ | $\mathbf{70.77 \pm 1.96}$ | $\mathbf{85.69 \pm 0.11}$ | $\mathbf{74.81 \pm 0.68}$ |
| | **Δ** | **+9.04** | **+6.62** | **+7.01** | **+8.81** | **+1.01** | **+0.76** | **+8.60** | **+4.85** |
| **GAT** | Vanilla | $51.51 \pm 0.53$ | $46.59 \pm 0.61$ | $34.74 \pm 0.16$ | $22.00 \pm 0.15$ | $60.22 \pm 0.47$ | $46.03 \pm 0.70$ | $68.09 \pm 2.96$ | $60.08 \pm 2.76$ |
| | Re-Weight | $58.68 \pm 3.44$ | $55.98 \pm 3.97$ | $36.78 \pm 0.94$ | $26.63 \pm 1.61$ | $63.47 \pm 1.73$ | $54.63 \pm 3.25$ | $71.44 \pm 2.42$ | $62.86 \pm 1.94$ |
| | PC Softmax | $59.62 \pm 1.41$ | $58.77 \pm 1.95$ | $43.38 \pm 2.01$ | $37.76 \pm 2.12$ | $70.81 \pm 1.41$ | $70.25 \pm 1.30$ | $71.16 \pm 1.15$ | $62.26 \pm 0.87$ |
| | BalancedSoftmax | $62.05 \pm 1.62$ | $61.14 \pm 1.71$ | $47.89 \pm 1.25$ | $44.84 \pm 1.35$ | $69.91 \pm 1.68$ | $67.43 \pm 1.73$ | $72.91 \pm 1.93$ | $62.79 \pm 0.98$ |
| | BalancedSoftmax (*w* TAM) | $63.30 \pm 0.99$ | $62.81 \pm 1.18$ | $49.34 \pm 1.29$ | $46.92 \pm 1.39$ | $71.17 \pm 2.09$ | $68.85 \pm 2.90$ | $65.59 \pm 2.86$ | $58.12 \pm 1.22$ |
| | Renode | $59.52 \pm 2.28$ | $57.16 \pm 2.47$ | $37.21 \pm 2.01$ | $27.09 \pm 3.17$ | $64.56 \pm 1.65$ | $55.87 \pm 2.83$ | $69.34 \pm 2.35$ | $59.02 \pm 1.67$ |
| | Renode (*w* TAM) | $61.32 \pm 2.18$ | $59.19 \pm 2.64$ | $39.85 \pm 2.20$ | $30.63 \pm 2.63$ | $66.28 \pm 3.24$ | $58.99 \pm 3.04$ | $65.81 \pm 2.57$ | $56.73 \pm 1.62$ |
| | GraphENS | $64.52 \pm 2.05$ | $62.52 \pm 1.84$ | $43.74 \pm 3.81$ | $37.47 \pm 4.21$ | $69.00 \pm 2.67$ | $65.54 \pm 3.54$ | $71.78 \pm 2.30$ | $61.83 \pm 1.75$ |
| | GraphENS (*w* TAM) | $65.78 \pm 1.62$ | $63.80 \pm 1.79$ | $44.81 \pm 2.66$ | $39.47 \pm 3.54$ | $70.33 \pm 2.33$ | $67.00 \pm 3.25$ | $73.55 \pm 2.04$ | $64.03 \pm 1.32$ |
| | GraphSR | $64.76 \pm 2.01$ | $62.75 \pm 1.79$ | $43.96 \pm 3.70$ | $37.73 \pm 4.10$ | $69.21 \pm 2.61$ | $65.76 \pm 3.48$ | $72.03 \pm 2.25$ | $62.04 \pm 1.72$ |
| | BIM | $64.72 \pm 2.03$ | $62.81 \pm 1.88$ | $43.91 \pm 3.79$ | $37.72 \pm 4.18$ | $69.21 \pm 2.65$ | $65.77 \pm 3.52$ | $72.01 \pm 2.33$ | $62.06 \pm 1.76$ |
| | **Ours** | $\mathbf{79.10 \pm 0.71}$ | $\mathbf{76.21 \pm 0.58}$ | $\mathbf{55.11 \pm 5.00}$ | $\mathbf{53.67 \pm 5.51}$ | $\mathbf{72.54 \pm 1.52}$ | $\mathbf{70.54 \pm 1.91}$ | $\mathbf{83.19 \pm 0.66}$ | $\mathbf{74.39 \pm 0.89}$ |
| | **Δ** | **+13.22** | **+12.41** | **+6.75** | **+8.81** | **+1.37** | **+1.69** | **+9.64** | **+10.36** |
| **SAGE** | Vanilla | $54.61 \pm 1.21$ | $50.95 \pm 1.90$ | $37.36 \pm 1.03$ | $27.49 \pm 1.41$ | $62.04 \pm 1.34$ | $54.18 \pm 1.73$ | $62.70 \pm 2.87$ | $55.39 \pm 2.69$ |
| | Re-Weight | $57.37 \pm 0.61$ | $55.30 \pm 0.72$ | $37.69 \pm 1.20$ | $27.92 \pm 2.01$ | $65.01 \pm 2.69$ | $58.34 \pm 2.19$ | $68.31 \pm 2.06$ | $60.45 \pm 2.40$ |
| | PC Softmax | $59.25 \pm 0.74$ | $58.55 \pm 0.81$ | $42.77 \pm 1.82$ | $40.08 \pm 1.82$ | $70.55 \pm 1.19$ | $67.60 \pm 1.59$ | $70.57 \pm 2.86$ | $62.73 \pm 2.69$ |
| | BalancedSoftmax | $61.93 \pm 1.26$ | $60.89 \pm 1.36$ | $43.64 \pm 1.33$ | $38.31 \pm 1.13$ | $69.89 \pm 1.40$ | $68.12 \pm 0.78$ | $68.45 \pm 2.92$ | $62.12 \pm 3.10$ |
| | BalancedSoftmax (*w* TAM) | $64.16 \pm 0.94$ | $63.63 \pm 1.10$ | $44.32 \pm 2.36$ | $40.17 \pm 2.06$ | $70.06 \pm 1.46$ | $69.54 \pm 1.35$ | $66.10 \pm 2.37$ | $59.22 \pm 2.48$ |
| | Renode | $58.48 \pm 0.97$ | $55.39 \pm 0.94$ | $40.65 \pm 2.36$ | $31.78 \pm 3.24$ | $66.50 \pm 2.63$ | $58.72 \pm 4.16$ | $68.36 \pm 1.54$ | $61.60 \pm 2.00$ |
| | Renode (*w* TAM) | $59.77 \pm 2.20$ | $57.98 \pm 2.79$ | $42.50 \pm 0.93$ | $35.11 \pm 1.84$ | $67.31 \pm 2.73$ | $60.63 \pm 3.49$ | $66.42 \pm 2.32$ | $58.62 \pm 1.95$ |
| | GraphENS | $63.54 \pm 0.91$ | $62.20 \pm 0.87$ | $44.89 \pm 2.51$ | $40.48 \pm 2.94$ | $71.37 \pm 1.77$ | $69.37 \pm 1.69$ | $75.47 \pm 2.20$ | $67.49 \pm 1.65$ |
| | GraphENS (*w* TAM) | $63.39 \pm 1.36$ | $61.66 \pm 1.53$ | $45.92 \pm 1.96$ | $41.97 \pm 2.50$ | $69.62 \pm 2.57$ | $66.85 \pm 3.00$ | $75.75 \pm 2.30$ | $68.86 \pm 1.29$ |
| | GraphSR | $63.75 \pm 0.92$ | $62.42 \pm 0.89$ | $45.06 \pm 2.48$ | $40.71 \pm 2.91$ | $71.59 \pm 1.76$ | $69.61 \pm 1.67$ | $75.71 \pm 2.18$ | $67.74 \pm 1.66$ |
| | BIM | $63.98 \pm 0.93$ | $62.68 \pm 0.88$ | $45.29 \pm 2.50$ | $40.93 \pm 2.90$ | $71.84 \pm 1.78$ | $69.86 \pm 1.66$ | $75.95 \pm 2.21$ | $67.97 \pm 1.64$ |
| | **Ours** | $\mathbf{73.10 \pm 1.60}$ | $\mathbf{69.92 \pm 1.43}$ | $\mathbf{58.35 \pm 4.58}$ | $\mathbf{57.51 \pm 4.92}$ | $\mathbf{73.67 \pm 0.58}$ | $\mathbf{71.15 \pm 0.67}$ | $\mathbf{78.88 \pm 2.16}$ | $\mathbf{69.00 \pm 1.42}$ |
| | **Δ** | **+8.94** | **+5.69** | **+12.43** | **+15.54** | **+1.83** | **+1.29** | **+2.93** | **+0.14** |

Table 22: Experimental results of our method and other baselines on four class-imbalanced node classification benchmark datasets with $\rho = 50$. We report averaged balanced accuracy (bAcc.,%) and F1-score (%) with the standard errors over 5 repetitions on three representative GNN architectures.

| | Dataset | Cora | | CiteSeer | | PubMed | | Amazon-Computers | |
|---|---|---|---|---|---|---|---|---|---|
| | Imbalance Ratio ($\rho = 50$) | bAcc. | F1 | bAcc. | F1 | bAcc. | F1 | bAcc. | F1 |
| **GCN** | Vanilla | 51.81 ± 0.62 | 43.98 ± 1.00 | 37.59 ± 0.17 | 23.54 ± 0.13 | 61.65 ± 0.34 | 47.95 ± 0.58 | 77.36 ± 3.41 | 69.68 ± 3.12 |
| | Re-Weight | 58.54 ± 2.39 | 54.13 ± 3.20 | 38.19 ± 1.28 | 27.43 ± 2.34 | 65.70 ± 1.59 | 56.35 ± 4.26 | 79.10 ± 2.44 | 71.40 ± 2.86 |
| | PC Softmax | 64.87 ± 2.23 | 62.01 ± 3.14 | 42.42 ± 2.19 | 38.83 ± 2.70 | 69.21 ± 0.59 | 69.40 ± 0.87 | 81.90 ± 1.63 | 74.34 ± 2.13 |
| | BalancedSoftmax | 65.94 ± 1.55 | 64.00 ± 2.05 | 47.62 ± 1.11 | 46.55 ± 1.46 | 70.40 ± 1.00 | 69.04 ± 0.66 | 82.97 ± 0.83 | 73.74 ± 1.27 |
| | BalancedSoftmax (*w* TAM) | 68.57 ± 1.58 | 67.25 ± 1.27 | 53.43 ± 2.42 | 51.74 ± 2.80 | 77.20 ± 1.45 | 74.86 ± 0.99 | 81.74 ± 2.30 | 73.85 ± 2.68 |
| | Renode | 62.22 ± 1.76 | 61.18 ± 2.24 | 41.23 ± 1.66 | 33.66 ± 2.69 | 68.67 ± 1.21 | 63.05 ± 1.47 | 81.71 ± 0.99 | 72.55 ± 1.61 |
| | Renode (*w* TAM) | 63.93 ± 1.96 | 61.64 ± 2.71 | 48.17 ± 1.58 | 41.07 ± 2.34 | 69.63 ± 2.55 | 64.30 ± 3.51 | 80.55 ± 1.75 | 72.33 ± 1.63 |
| | GraphENS | 63.47 ± 0.98 | 62.21 ± 1.65 | 48.17 ± 1.58 | 41.07 ± 2.34 | 69.63 ± 2.55 | 64.30 ± 3.51 | 81.63 ± 2.35 | 72.57 ± 2.33 |
| | GraphENS (*w* TAM) | 65.05 ± 1.11 | 62.11 ± 1.98 | 45.03 ± 1.34 | 42.65 ± 1.94 | 69.74 ± 0.78 | 70.82 ± 0.63 | 81.69 ± 2.22 | 72.09 ± 1.75 |
| | GraphSR | 64.12 ± 0.94 | 62.89 ± 1.58 | 48.84 ± 1.52 | 41.76 ± 2.26 | 70.31 ± 2.48 | 64.98 ± 3.40 | 82.28 ± 2.30 | 73.21 ± 2.28 |
| | BIM | 65.72 ± 1.07 | 62.80 ± 1.90 | 45.68 ± 1.29 | 43.33 ± 1.88 | 70.42 ± 0.74 | 71.46 ± 0.66 | 82.34 ± 2.17 | 72.76 ± 1.71 |
| | **Ours** | **75.62 ± 2.02** | **72.59 ± 2.13** | **59.97 ± 4.59** | **58.66 ± 5.20** | **78.55 ± 0.84** | **75.91 ± 0.81** | **85.54 ± 0.26** | **75.76 ± 0.13** |
| | **Δ** | **+7.05** | **+5.34** | **+6.54** | **+6.92** | **+1.35** | **+1.06** | **+2.57** | **+1.91** |
| **GAT** | Vanilla | 53.90 ± 0.63 | 45.53 ± 0.89 | 36.48 ± 0.08 | 23.68 ± 0.16 | 60.16 ± 0.47 | 46.99 ± 0.58 | 72.42 ± 2.17 | 64.41 ± 2.68 |
| | Re-Weight | 59.78 ± 1.92 | 56.69 ± 2.21 | 38.70 ± 2.23 | 29.38 ± 3.06 | 66.27 ± 0.68 | 57.34 ± 1.41 | 73.46 ± 3.07 | 67.00 ± 2.60 |
| | PC Softmax | 59.44 ± 2.62 | 58.06 ± 2.69 | 43.13 ± 1.56 | 37.04 ± 2.07 | 70.86 ± 0.44 | 70.96 ± 0.54 | 77.21 ± 2.90 | 69.17 ± 2.89 |
| | BalancedSoftmax | 64.71 ± 2.28 | 62.55 ± 2.61 | 51.89 ± 1.15 | 49.36 ± 1.52 | 70.94 ± 1.09 | 70.33 ± 0.99 | 77.49 ± 1.58 | 70.44 ± 2.33 |
| | BalancedSoftmax (*w* TAM) | 68.05 ± 1.03 | 66.07 ± 1.14 | 54.28 ± 0.79 | 52.77 ± 0.97 | 75.65 ± 1.11 | 74.02 ± 1.44 | 78.86 ± 1.53 | 70.71 ± 2.04 |
| | Renode | 63.81 ± 1.72 | 60.63 ± 2.26 | 41.60 ± 2.30 | 33.94 ± 4.60 | 70.35 ± 1.26 | 67.43 ± 0.01 | 72.39 ± 2.75 | 65.23 ± 3.35 |
| | Renode (*w* TAM) | 64.40 ± 1.83 | 63.48 ± 2.83 | 43.54 ± 1.54 | 35.80 ± 2.43 | 71.23 ± 2.04 | 66.61 ± 4.31 | 76.07 ± 2.70 | 68.43 ± 2.68 |
| | GraphENS | 64.52 ± 2.51 | 61.41 ± 3.15 | 45.23 ± 2.97 | 41.12 ± 4.23 | 69.66 ± 1.01 | 66.83 ± 0.94 | 78.36 ± 2.74 | 70.44 ± 2.51 |
| | GraphENS (*w* TAM) | 65.33 ± 2.67 | 65.34 ± 2.53 | 48.00 ± 1.46 | 48.14 ± 1.43 | 71.50 ± 1.26 | 72.58 ± 1.07 | 80.02 ± 2.32 | 72.38 ± 2.47 |
| | GraphSR | 65.17 ± 2.44 | 62.11 ± 3.08 | 45.89 ± 2.89 | 41.79 ± 4.10 | 70.31 ± 0.98 | 67.49 ± 0.91 | 79.05 ± 2.66 | 71.12 ± 2.46 |
| | BIM | 65.98 ± 2.60 | 66.03 ± 2.47 | 48.63 ± 1.42 | 48.87 ± 1.38 | 72.19 ± 1.22 | 73.28 ± 1.03 | 80.65 ± 2.27 | 73.03 ± 2.42 |
| | **Ours** | **77.07 ± 0.83** | **73.44 ± 1.05** | **57.70 ± 4.35** | **56.81 ± 4.67** | **79.41 ± 0.29** | **77.38 ± 0.39** | **86.06 ± 0.45** | **77.55 ± 0.71** |
| | **Δ** | **+9.02** | **+7.37** | **+3.42** | **+4.04** | **+3.76** | **+3.36** | **+5.41** | **+4.52** |
| **SAGE** | Vanilla | 53.02 ± 0.83 | 45.58 ± 1.30 | 38.81 ± 0.89 | 25.28 ± 0.51 | 61.41 ± 1.01 | 50.46 ± 2.47 | 56.53 ± 2.12 | 48.52 ± 2.75 |
| | Re-Weight | 58.03 ± 0.81 | 54.32 ± 0.99 | 38.49 ± 1.34 | 30.41 ± 1.82 | 64.25 ± 0.90 | 51.37 ± 2.62 | 70.36 ± 2.21 | 61.52 ± 2.73 |
| | PC Softmax | 62.33 ± 1.62 | 59.97 ± 1.98 | 41.79 ± 1.19 | 36.90 ± 0.84 | 69.58 ± 1.09 | 67.13 ± 0.95 | 73.53 ± 2.02 | 66.12 ± 3.19 |
| | BalancedSoftmax | 64.57 ± 0.77 | 62.22 ± 0.82 | 41.84 ± 1.72 | 40.09 ± 1.04 | 70.43 ± 0.38 | 68.99 ± 0.99 | 73.27 ± 2.30 | 68.30 ± 1.97 |
| | BalancedSoftmax (*w* TAM) | 65.97 ± 0.71 | 65.53 ± 0.88 | 52.89 ± 1.65 | 49.92 ± 1.83 | 71.11 ± 0.75 | 71.73 ± 0.79 | 73.12 ± 1.41 | 66.45 ± 1.04 |
| | Renode | 61.35 ± 1.86 | 58.88 ± 2.53 | 40.37 ± 2.33 | 32.57 ± 3.62 | 67.54 ± 3.05 | 59.77 ± 5.30 | 70.46 ± 3.45 | 62.30 ± 4.40 |
| | Renode (*w* TAM) | 62.79 ± 0.47 | 61.05 ± 0.82 | 43.04 ± 1.30 | 36.97 ± 1.92 | 71.79 ± 1.33 | 67.80 ± 2.45 | 74.55 ± 2.95 | 66.06 ± 2.16 |
| | GraphENS | 63.95 ± 0.96 | 62.63 ± 2.12 | 41.99 ± 1.54 | 37.44 ± 2.43 | 66.07 ± 1.12 | 61.63 ± 1.82 | 76.21 ± 2.84 | 68.10 ± 2.56 |
| | GraphENS (*w* TAM) | 65.98 ± 1.37 | 64.84 ± 1.13 | 49.54 ± 1.79 | 49.48 ± 1.70 | 73.24 ± 1.32 | 73.73 ± 1.14 | 80.75 ± 1.22 | 72.31 ± 0.95 |
| | GraphSR | 64.58 ± 0.91 | 63.32 ± 2.05 | 42.67 ± 1.49 | 38.13 ± 2.35 | 66.78 ± 1.08 | 62.31 ± 1.75 | 76.87 ± 2.78 | 68.74 ± 2.49 |
| | BIM | 65.60 ± 0.91 | 64.32 ± 2.06 | 43.70 ± 1.50 | 39.13 ± 2.35 | 67.84 ± 1.07 | 63.37 ± 1.76 | 77.92 ± 2.79 | 69.78 ± 2.49 |
| | **Ours** | **76.04 ± 1.30** | **72.99 ± 1.25** | **58.70 ± 4.10** | **57.53 ± 4.59** | **75.27 ± 1.26** | **72.16 ± 1.50** | **82.03 ± 0.77** | **72.98 ± 0.52** |
| | **Δ** | **+10.06** | **+7.46** | **+5.81** | **+7.61** | **+2.03** | **-1.57** | **+1.28** | **+0.67** |

Table 23: Experimental results of our method and other baselines on four class-imbalanced node classification benchmark datasets with $\rho = 100$. We report averaged balanced accuracy (bAcc.,%) and F1-score (%) with the standard errors over 5 repetitions on three representative GNN architectures.

| | Dataset | Cora | | CiteSeer | | PubMed | | Amazon-Computers | |
|---|---|---|---|---|---|---|---|---|---|
| | Imbalance Ratio ($\rho = 100$) | bAcc. | F1 | bAcc. | F1 | bAcc. | F1 | bAcc. | F1 |
| **GCN** | Vanilla | $51.62 \pm 0.20$ | $43.91 \pm 0.25$ | $38.83 \pm 0.26$ | $24.71 \pm 0.25$ | $61.28 \pm 0.12$ | $47.55 \pm 0.16$ | $76.09 \pm 3.79$ | $69.32 \pm 3.49$ |
| | Re-Weight | $59.11 \pm 1.06$ | $54.04 \pm 1.36$ | $42.67 \pm 2.06$ | $33.17 \pm 3.40$ | $67.14 \pm 2.71$ | $55.24 \pm 5.36$ | $81.53 \pm 2.20$ | $71.45 \pm 2.05$ |
| | PC Softmax | $63.75 \pm 1.02$ | $61.19 \pm 1.43$ | $38.34 \pm 0.71$ | $33.65 \pm 1.42$ | $70.85 \pm 0.44$ | $70.26 \pm 0.63$ | $82.22 \pm 1.99$ | $72.38 \pm 2.52$ |
| | BalancedSoftmax | $63.03 \pm 1.57$ | $61.28 \pm 1.77$ | $48.49 \pm 1.20$ | $46.59 \pm 1.34$ | $70.77 \pm 1.88$ | $68.88 \pm 1.74$ | $83.33 \pm 3.35$ | $74.34 \pm 2.74$ |
| | BalancedSoftmax (*w* TAM) | $69.44 \pm 0.59$ | $67.10 \pm 0.88$ | $52.60 \pm 0.69$ | $51.21 \pm 0.84$ | $73.73 \pm 1.10$ | $73.72 \pm 0.83$ | $83.70 \pm 2.17$ | $75.39 \pm 1.43$ |
| | Renode | $60.76 \pm 2.53$ | $58.09 \pm 3.00$ | $43.41 \pm 2.07$ | $33.69 \pm 2.76$ | $67.63 \pm 2.77$ | $61.70 \pm 4.84$ | $82.13 \pm 1.73$ | $71.79 \pm 1.85$ |
| | Renode (*w* TAM) | $64.19 \pm 1.46$ | $60.90 \pm 1.56$ | $44.78 \pm 1.51$ | $35.90 \pm 2.61$ | $70.53 \pm 0.75$ | $64.35 \pm 1.79$ | $82.32 \pm 2.19$ | $73.09 \pm 1.75$ |
| | GraphENS | $63.00 \pm 1.30$ | $62.33 \pm 1.67$ | $45.99 \pm 2.06$ | $37.23 \pm 3.40$ | $68.65 \pm 1.00$ | $62.17 \pm 1.60$ | $83.37 \pm 2.17$ | $73.96 \pm 1.98$ |
| | GraphENS (*w* TAM) | $60.40 \pm 4.42$ | $57.77 \pm 4.02$ | $42.72 \pm 2.54$ | $39.40 \pm 2.57$ | $70.73 \pm 1.96$ | $72.50 \pm 1.87$ | $81.29 \pm 1.52$ | $71.66 \pm 1.75$ |
| | GraphSR | $64.64 \pm 1.25$ | $64.04 \pm 1.62$ | $47.66 \pm 1.98$ | $38.96 \pm 3.28$ | $70.29 \pm 0.95$ | $63.85 \pm 1.52$ | $83.02 \pm 2.12$ | $73.60 \pm 1.90$ |
| | BIM | $64.38 \pm 1.26$ | $63.69 \pm 1.61$ | $47.31 \pm 2.00$ | $38.61 \pm 3.28$ | $70.03 \pm 0.96$ | $63.51 \pm 1.54$ | $82.77 \pm 2.10$ | $73.24 \pm 1.91$ |
| | **Ours** | $\mathbf{72.82 \pm 3.55}$ | $\mathbf{69.12 \pm 3.45}$ | $\mathbf{57.66 \pm 1.96}$ | $\mathbf{56.50 \pm 1.12}$ | $\mathbf{78.73 \pm 0.88}$ | $\mathbf{76.03 \pm 1.08}$ | $\mathbf{84.30 \pm 0.30}$ | $\mathbf{76.06 \pm 0.32}$ |
| | **Δ** | **+3.38** | **+2.02** | **+5.06** | **+5.29** | **+5.00** | **+2.31** | **+0.60** | **+0.67** |
| **GAT** | Vanilla | $51.58 \pm 0.32$ | $43.37 \pm 0.21$ | $37.91 \pm 0.28$ | $23.49 \pm 0.21$ | $62.07 \pm 0.17$ | $47.39 \pm 0.20$ | $72.66 \pm 2.97$ | $64.87 \pm 3.46$ |
| | Re-Weight | $58.28 \pm 1.88$ | $54.47 \pm 2.35$ | $38.13 \pm 1.55$ | $29.60 \pm 3.02$ | $67.41 \pm 2.69$ | $58.06 \pm 5.07$ | $77.10 \pm 3.26$ | $68.35 \pm 2.71$ |
| | PC Softmax | $63.74 \pm 2.01$ | $59.76 \pm 2.19$ | $45.07 \pm 1.13$ | $39.21 \pm 2.29$ | $69.68 \pm 1.29$ | $69.44 \pm 1.29$ | $79.72 \pm 1.52$ | $70.78 \pm 1.45$ |
| | BalancedSoftmax | $63.19 \pm 1.35$ | $61.03 \pm 1.46$ | $46.03 \pm 2.11$ | $43.38 \pm 2.24$ | $71.45 \pm 1.23$ | $69.10 \pm 1.20$ | $79.15 \pm 2.08$ | $69.68 \pm 2.13$ |
| | BalancedSoftmax (*w* TAM) | $64.96 \pm 3.23$ | $62.91 \pm 3.96$ | $52.75 \pm 1.29$ | $50.69 \pm 1.83$ | $73.38 \pm 0.77$ | $72.45 \pm 0.88$ | $80.86 \pm 2.52$ | $72.93 \pm 2.95$ |
| | Renode | $60.04 \pm 2.21$ | $58.04 \pm 2.66$ | $42.40 \pm 2.97$ | $34.09 \pm 0.04$ | $68.54 \pm 2.11$ | $65.63 \pm 3.15$ | $75.34 \pm 1.65$ | $69.99 \pm 1.60$ |
| | Renode (*w* TAM) | $63.45 \pm 1.41$ | $61.51 \pm 1.95$ | $41.55 \pm 1.39$ | $36.13 \pm 2.87$ | $71.53 \pm 2.35$ | $68.11 \pm 4.28$ | $78.60 \pm 1.90$ | $70.35 \pm 2.80$ |
| | GraphENS | $63.93 \pm 2.70$ | $61.77 \pm 3.38$ | $44.43 \pm 1.90$ | $39.26 \pm 2.55$ | $68.50 \pm 1.81$ | $64.14 \pm 3.28$ | $81.63 \pm 2.08$ | $71.20 \pm 2.75$ |
| | GraphENS (*w* TAM) | $62.52 \pm 0.95$ | $61.65 \pm 1.19$ | $45.79 \pm 1.31$ | $44.80 \pm 1.14$ | $69.09 \pm 1.11$ | $70.64 \pm 1.10$ | $83.33 \pm 0.83$ | $72.81 \pm 1.22$ |
| | GraphSR | $64.89 \pm 2.62$ | $62.74 \pm 3.30$ | $45.39 \pm 1.86$ | $40.18 \pm 2.48$ | $69.47 \pm 1.75$ | $65.08 \pm 3.21$ | $82.52 \pm 2.02$ | $72.13 \pm 2.69$ |
| | BIM | $65.84 \pm 2.55$ | $63.72 \pm 3.22$ | $46.26 \pm 1.82$ | $41.10 \pm 2.42$ | $70.45 \pm 1.70$ | $66.00 \pm 3.15$ | $83.39 \pm 1.97$ | $73.04 \pm 2.63$ |
| | **Ours** | $\mathbf{75.42 \pm 0.91}$ | $\mathbf{71.50 \pm 0.89}$ | $\mathbf{60.35 \pm 1.87}$ | $\mathbf{59.63 \pm 1.86}$ | $\mathbf{77.88 \pm 1.31}$ | $\mathbf{74.98 \pm 1.35}$ | $\mathbf{85.33 \pm 0.19}$ | $\mathbf{75.83 \pm 0.74}$ |
| | **Δ** | **+9.58** | **+7.78** | **+7.60** | **+8.94** | **+4.50** | **+2.53** | **+1.94** | **+2.79** |
| **SAGE** | Vanilla | $52.65 \pm 0.24$ | $43.79 \pm 0.47$ | $36.63 \pm 0.09$ | $24.12 \pm 0.09$ | $62.29 \pm 0.25$ | $47.02 \pm 0.38$ | $55.94 \pm 2.37$ | $47.21 \pm 2.73$ |
| | Re-Weight | $59.42 \pm 2.88$ | $55.26 \pm 4.40$ | $36.24 \pm 1.30$ | $27.07 \pm 2.88$ | $63.33 \pm 0.75$ | $55.11 \pm 1.62$ | $70.76 \pm 3.35$ | $62.09 \pm 3.30$ |
| | PC Softmax | $64.01 \pm 1.15$ | $60.74 \pm 1.68$ | $44.74 \pm 1.41$ | $37.61 \pm 1.69$ | $72.62 \pm 1.42$ | $70.95 \pm 1.70$ | $75.96 \pm 2.44$ | $69.12 \pm 2.90$ |
| | BalancedSoftmax | $63.43 \pm 2.12$ | $62.30 \pm 2.27$ | $49.33 \pm 1.12$ | $44.58 \pm 1.64$ | $70.68 \pm 0.92$ | $69.15 \pm 0.84$ | $74.66 \pm 0.86$ | $66.28 \pm 1.92$ |
| | BalancedSoftmax (*w* TAM) | $66.58 \pm 1.53$ | $64.56 \pm 2.49$ | $53.33 \pm 1.06$ | $50.15 \pm 1.45$ | $72.59 \pm 2.06$ | $72.22 \pm 2.08$ | $78.01 \pm 1.06$ | $71.02 \pm 1.08$ |
| | Renode | $62.42 \pm 0.90$ | $60.08 \pm 1.19$ | $39.61 \pm 2.66$ | $30.13 \pm 3.86$ | $67.11 \pm 1.12$ | $61.09 \pm 3.50$ | $73.73 \pm 2.26$ | $64.47 \pm 2.39$ |
| | Renode (*w* TAM) | $62.06 \pm 2.08$ | $60.72 \pm 3.32$ | $42.08 \pm 1.88$ | $33.19 \pm 3.45$ | $69.95 \pm 1.01$ | $65.99 \pm 2.28$ | $74.81 \pm 3.29$ | $67.48 \pm 3.32$ |
| | GraphENS | $63.09 \pm 0.97$ | $61.20 \pm 1.74$ | $42.03 \pm 1.88$ | $36.71 \pm 2.99$ | $69.71 \pm 1.87$ | $63.47 \pm 3.87$ | $81.33 \pm 1.66$ | $72.83 \pm 1.76$ |
| | GraphENS (*w* TAM) | $65.95 \pm 2.25$ | $63.88 \pm 1.78$ | $51.03 \pm 1.51$ | $50.49 \pm 1.88$ | $73.58 \pm 2.01$ | $72.44 \pm 1.77$ | $81.72 \pm 1.08$ | $72.31 \pm 1.98$ |
| | GraphSR | $66.45 \pm 2.10$ | $64.42 \pm 1.83$ | $53.52 \pm 1.47$ | $53.01 \pm 1.75$ | $74.09 \pm 2.12$ | $72.97 \pm 1.90$ | $81.45 \pm 0.87$ | $72.65 \pm 1.54$ |
| | BIM | $67.75 \pm 2.13$ | $64.68 \pm 1.95$ | $53.83 \pm 1.62$ | $53.29 \pm 1.80$ | $74.38 \pm 2.04$ | $73.24 \pm 1.85$ | $82.01 \pm 0.43$ | $72.32 \pm 1.01$ |
| | **Ours** | $\mathbf{73.47 \pm 2.31}$ | $\mathbf{68.30 \pm 2.11}$ | $\mathbf{59.77 \pm 2.98}$ | $\mathbf{58.92 \pm 3.07}$ | $\mathbf{77.11 \pm 0.59}$ | $\mathbf{74.03 \pm 0.81}$ | $\mathbf{82.92 \pm 2.94}$ | $\mathbf{73.11 \pm 2.57}$ |
| | **Δ** | **+5.72** | **+3.62** | **+6.04** | **+5.63** | **+2.73** | **+0.79** | **+0.91** | **+0.79** |

Table 24: Experimental results of our method and other baselines on Computers-Random. We report averaged balanced accuracy (bAcc.,%) and F1-score (%) with the standard errors over 5 repetitions on three representative GNN architectures.

| Dataset (Computers-Random) | GCN | | GAT | | SAGE | |
|---|---|---|---|---|---|---|
| Imbalance Ratio ($\rho = 25.50$) | bAcc. | F1 | bAcc. | F1 | bAcc. | F1 |
| Vanilla | $78.43 \pm 0.41$ | $77.14 \pm 0.39$ | $71.35 \pm 1.18$ | $69.60 \pm 1.11$ | $65.30 \pm 1.07$ | $64.77 \pm 1.19$ |
| Re-Weight | $80.49 \pm 0.44$ | $75.07 \pm 0.58$ | $71.95 \pm 0.80$ | $70.67 \pm 0.51$ | $66.50 \pm 1.47$ | $66.10 \pm 1.46$ |
| PC Softmax | $81.34 \pm 0.55$ | $75.17 \pm 0.57$ | $70.56 \pm 1.46$ | $67.26 \pm 1.48$ | $69.73 \pm 0.53$ | $67.03 \pm 0.6$ |
| BalancedSoftmax | $81.39 \pm 0.25$ | $74.54 \pm 0.64$ | $72.09 \pm 0.31$ | $68.38 \pm 0.69$ | $73.80 \pm 1.06$ | $69.74 \pm 0.60$ |
| GraphSMOTE | $80.50 \pm 1.11$ | $73.79 \pm 0.14$ | $71.98 \pm 0.21$ | $67.98 \pm 0.31$ | $72.69 \pm 0.82$ | $68.73 \pm 1.01$ |
| Renode | $81.64 \pm 0.34$ | $76.87 \pm 0.32$ | $72.80 \pm 0.94$ | $71.40 \pm 0.97$ | $70.94 \pm 1.50$ | $70.04 \pm 1.16$ |
| GraphENS | $82.66 \pm 0.61$ | $76.55 \pm 0.17$ | $75.25 \pm 0.85$ | $71.49 \pm 0.54$ | $77.64 \pm 0.52$ | $72.65 \pm 0.53$ |
| BalancedSoftmax+TAM | $81.64 \pm 0.48$ | $75.59 \pm 0.83$ | $74.00 \pm 0.77$ | $70.72 \pm 0.50$ | $73.77 \pm 1.26$ | $71.03 \pm 0.69$ |
| Renode+TAM | $80.50 \pm 1.11$ | $75.79 \pm 0.14$ | $71.98 \pm 0.21$ | $70.98 \pm 0.31$ | $72.69 \pm 0.82$ | $70.73 \pm 1.01$ |
| GraphENS+TAM | $82.83 \pm 0.68$ | $76.76 \pm 0.39$ | $75.81 \pm 0.72$ | $72.62 \pm 0.57$ | $78.98 \pm 0.60$ | $73.59 \pm 0.55$ |
| GraphSR | $83.82 \pm 0.74$ | $77.78 \pm 0.42$ | $76.79 \pm 0.68$ | $73.61 \pm 0.63$ | $77.63 \pm 0.32$ | $72.56 \pm 0.51$ |
| BIM | $84.03 \pm 0.73$ | $77.96 \pm 0.45$ | $77.01 \pm 0.70$ | $73.82 \pm 0.60$ | $77.76 \pm 0.65$ | $72.09 \pm 0.37$ |
| **Ours** | $\mathbf{85.32 \pm 0.22}$ | $\mathbf{80.43 \pm 0.56}$ | $\mathbf{82.52 \pm 0.35}$ | $\mathbf{78.90 \pm 0.38}$ | $75.81 \pm 1.86$ | $\underline{71.86 \pm 1.86}$ |
| **Δ** | **+1.29** | **+2.47** | **+5.51** | **+5.08** | **-3.17** | **-1.73** |

Table 25: Experimental results of our method and other baselines on CS-Random. We report averaged balanced accuracy (bAcc.,%) and F1-score (%) with the standard errors over 5 repetitions on three representative GNN architectures.

| Dataset (CS-Random) | GCN | | GAT | | SAGE | |
|---|---|---|---|---|---|---|
| Imbalance Ratio ($\rho = 41.00$) | bAcc. | F1 | bAcc. | F1 | bAcc. | F1 |
| Vanilla | $84.85 \pm 0.16$ | $87.12 \pm 0.14$ | $82.47 \pm 0.36$ | $84.21 \pm 0.31$ | $83.76 \pm 0.27$ | $86.22 \pm 0.19$ |
| Re-Weight | $87.42 \pm 0.17$ | $88.70 \pm 0.10$ | $83.55 \pm 0.39$ | $84.73 \pm 0.32$ | $85.76 \pm 0.24$ | $87.32 \pm 0.16$ |
| PC Softmax | $88.36 \pm 0.12$ | $88.94 \pm 0.04$ | $85.22 \pm 0.31$ | $85.54 \pm 0.33$ | $87.18 \pm 0.14$ | $88.00 \pm 0.19$ |
| GraphSMOTE | $85.76 \pm 1.73$ | $87.31 \pm 1.32$ | $84.65 \pm 1.32$ | $85.63 \pm 1.01$ | $85.76 \pm 1.98$ | $87.34 \pm 0.98$ |
| BalancedSoftmax | $87.72 \pm 0.07$ | $88.67 \pm 0.07$ | $84.38 \pm 0.20$ | $84.53 \pm 0.41$ | $86.78 \pm 0.10$ | $88.05 \pm 0.09$ |
| BalancedSoftmax (*w* TAM) | $88.22 \pm 0.11$ | $89.22 \pm 0.08$ | $85.48 \pm 0.24$ | $85.77 \pm 0.50$ | $87.83 \pm 0.13$ | $88.77 \pm 0.07$ |
| Renode | $87.53 \pm 0.11$ | $88.91 \pm 0.06$ | $85.98 \pm 0.19$ | $86.97 \pm 0.09$ | $86.13 \pm 0.10$ | $87.89 \pm 0.09$ |
| Renode (*w* TAM) | $87.55 \pm 0.06$ | $89.03 \pm 0.05$ | $86.61 \pm 0.30$ | $87.42 \pm 0.24$ | $85.21 \pm 0.33$ | $87.01 \pm 0.31$ |
| GraphENS | $85.97 \pm 0.29$ | $86.68 \pm 0.20$ | $85.86 \pm 0.19$ | $86.51 \pm 0.32$ | $85.39 \pm 0.26$ | $86.41 \pm 0.24$ |
| GraphENS (*w* TAM) | $86.34 \pm 0.12$ | $87.36 \pm 0.08$ | $86.29 \pm 0.20$ | $87.28 \pm 0.13$ | $85.99 \pm 0.13$ | $87.25 \pm 0.07$ |
| GraphSR | $86.73 \pm 0.22$ | $85.91 \pm 0.21$ | $85.34 \pm 0.13$ | $86.56 \pm 0.29$ | $85.44 \pm 0.27$ | $86.46 \pm 0.23$ |
| BIM | $86.89 \pm 0.23$ | $85.99 \pm 0.21$ | $85.63 \pm 1.87$ | $86.65 \pm 0.35$ | $85.65 \pm 0.28$ | $86.73 \pm 0.22$ |
| **Ours** | $\mathbf{88.94 \pm 0.09}$ | $\mathbf{89.87 \pm 0.06}$ | $\mathbf{87.65 \pm 0.12}$ | $\mathbf{87.65 \pm 0.11}$ | $\mathbf{88.03 \pm 0.21}$ | $\underline{88.65 \pm 0.07}$ |
| **Δ** | **+ 0.58** | **+ 0.65** | **+ 1.04** | **+ 0.23** | **+ 0.20** | **- 0.12** |

Table 26: Experimental results of our method and other baselines on Flickr. We report averaged balanced accuracy (bAcc.,%) and F1-score (%) with the standard errors over 5 repetitions on three representative GNN architectures.

| Model | GCN | | GAT | | SAGE | |
|---|---|---|---|---|---|---|
| Imbalance Ratio ($\rho = 10.80$) | bAcc. | F1 | bAcc. | F1 | bAcc. | F1 |
| Vanilla | 24.62 ± 0.07 | 24.53 ± 0.11 | 25.87 ± 0.30 | 25.32 ± 0.44 | 25.29 ± 0.18 | 24.16 ± 0.27 |
| Re-weight | 28.31 ± 1.64 | 24.06 ± 1.16 | **30.66 ± 0.76** | 27.12 ± 0.34 | 27.39 ± 1.84 | 22.62 ± 1.04 |
| PC Softmax | 29.21 ± 2.16 | 25.81 ± 1.75 | 30.20 ± 0.46 | 27.24 ± 0.37 | 25.40 ± 2.49 | 21.08 ± 1.73 |
| GraphSMOTE | OOM | OOM | OOM | OOM | OOM | OOM |
| BalancedSoftmax | 27.61 ± 0.61 | 23.70 ± 0.77 | 26.01 ± 2.81 | 23.50 ± 3.07 | 28.24 ± 2.10 | 24.98 ± 1.59 |
| BalancedSoftmax (*w* TAM) | 27.06 ± 1.03 | 23.97 ± 0.60 | 28.24 ± 0.99 | 25.52 ± 0.89 | 29.79 ± 0.37 | 27.56 ± 0.25 |
| Renode | OOM | OOM | OOM | OOM | OOM | OOM |
| Renode (*w* TAM) | OOM | OOM | OOM | OOM | OOM | OOM |
| GraphENS | OOM | OOM | OOM | OOM | OOM | OOM |
| GraphENS (*w* TAM) | OOM | OOM | OOM | OOM | OOM | OOM |
| GraphSR | 27.63 ± 0.59 | 23.73 ± 0.81 | 26.03 ± 2.75 | 23.53 ± 3.15 | 28.26 ± 2.18 | 25.01 ± 1.62 |
| BIM | 27.87 ± 0.65 | 23.75 ± 0.73 | 26.15 ± 2.70 | 23.74 ± 3.10 | 28.34 ± 2.00 | 25.03 ± 1.66 |
| **Ours** | **30.76 ± 0.27** | **30.60 ± 0.29** | 29.45 ± 0.72 | **28.21 ± 0.76** | **30.68 ± 0.63** | **31.01 ± 1.34** |
| Δ | +1.55 | +4.79 | -1.21 | +0.97 | +0.89 | +3.45 |

Table 27: Experimental results of our method and other baselines on Ogbn-Arxiv. We report averaged balanced accuracy (bAcc.,%) and F1-score (%) with the standard errors over 5 repetitions on three representative GNN architectures.

| Model | GCN | | GAT | | SAGE | |
|---|---|---|---|---|---|---|
| Imbalance Ratio ($\rho = 775.40$) | bAcc. | F1 | bAcc. | F1 | bAcc. | F1 |
| Vanilla | 50.21 ± 0.65 | 49.60 ± 0.14 | 51.21 ± 0.87 | 49.23 ± 0.33 | 50.76 ± 0.21 | 49.43 ± 0.29 |
| Re-weight | 50.24 ± 0.40 | 49.71 ± 0.12 | 51.12 ± 0.80 | 49.65 ± 0.25 | 50.81 ± 0.19 | 49.78 ± 0.22 |
| PC Softmax | 50.20 ± 0.58 | 49.64 ± 0.12 | 51.18 ± 0.77 | 49.16 ± 0.28 | 50.82 ± 0.19 | 49.65 ± 0.24 |
| GS | OOM | OOM | OOM | OOM | OOM | OOM |
| BalancedSoftmax | 50.34 ± 0.41 | 49.73 ± 0.13 | 51.35 ± 0.69 | 49.36 ± 0.22 | 50.89 ± 0.19 | 49.56 ± 0.18 |
| BalancedSoftmax (*w* TAM) | 50.34 ± 0.48 | 49.72 ± 0.10 | 51.36 ± 0.72 | 49.98 ± 0.26 | 50.94 ± 0.17 | 49.95 ± 0.22 |
| ReNode | OOM | OOM | OOM | OOM | OOM | OOM |
| REnode (*w* TAM) | OOM | OOM | OOM | OOM | OOM | OOM |
| GraphENS | OOM | OOM | OOM | OOM | OOM | OOM |
| GraphENS (*w* TAM) | OOM | OOM | OOM | OOM | OOM | OOM |
| GraphSR | 50.31 ± 0.24 | 49.70 ± 0.17 | 51.31 ± 0.41 | 49.33 ± 0.26 | 50.86 ± 0.30 | 49.53 ± 0.20 |
| BIM | 50.33 ± 0.42 | 49.71 ± 0.19 | 51.35 ± 0.60 | 49.36 ± 0.28 | 50.87 ± 0.18 | 49.56 ± 0.23 |
| **Ours** | $51.21_{\pm 0.32}$ | $50.65_{\pm 0.32}$ | $51.84_{\pm 0.87}$ | $51.28_{\pm 0.42}$ | $51.34_{\pm 0.32}$ | $51.36_{\pm 0.27}$ |
| Δ | +0.87 | +0.92 | +0.48 | +1.30 | +0.40 | +0.41 |

# J    Comprehensive Abaltion Study

## J.1    Analysis for Decoupling Representation and Classifier for Imbalance Node Classification.

We conduct more extensive experiments on the Cora and Amazon-Computers datasets using three different GNN architectures to analyze the effect of decoupling representation and classifier for imbalanced node classification. We hypothesize that even if the GNN encoder is trained on skewed data, the embeddings it learns are of high quality.

**Experimental Setup.** As explained in Section 4.1, we can obtain two pseudo-labels for all unlabeled nodes, one from unsupervised algorithms and the other from supervised classifiers. Experiments on more datasets are conducted to compare the accuracy of the two pseudo-labels for all unlabeled nodes. We chose the two benchmark datasets, Cora and Amazon-Computers, to build scenarios with varying degrees of imbalance ($\rho = 1, 5, 10, 20, 50, 100$). To be more specific, half of the classes are designated as minority classes and randomly selected labeled nodes are converted into unlabeled nodes until the training set's imbalance ratio reaches $\rho$. The GNN architecture is fixed as the 2-layer GNN (i.e. GCN [25], GAT [54], GraphSAGE [12]) having 128 hidden dimensions and train models for 2000 epochs. We set the K-Means algorithm's cluster size $k'$ to 200. Each experiment is repeated five times, and the average experiment results under different imbalance ratios are shown in Figure 9.

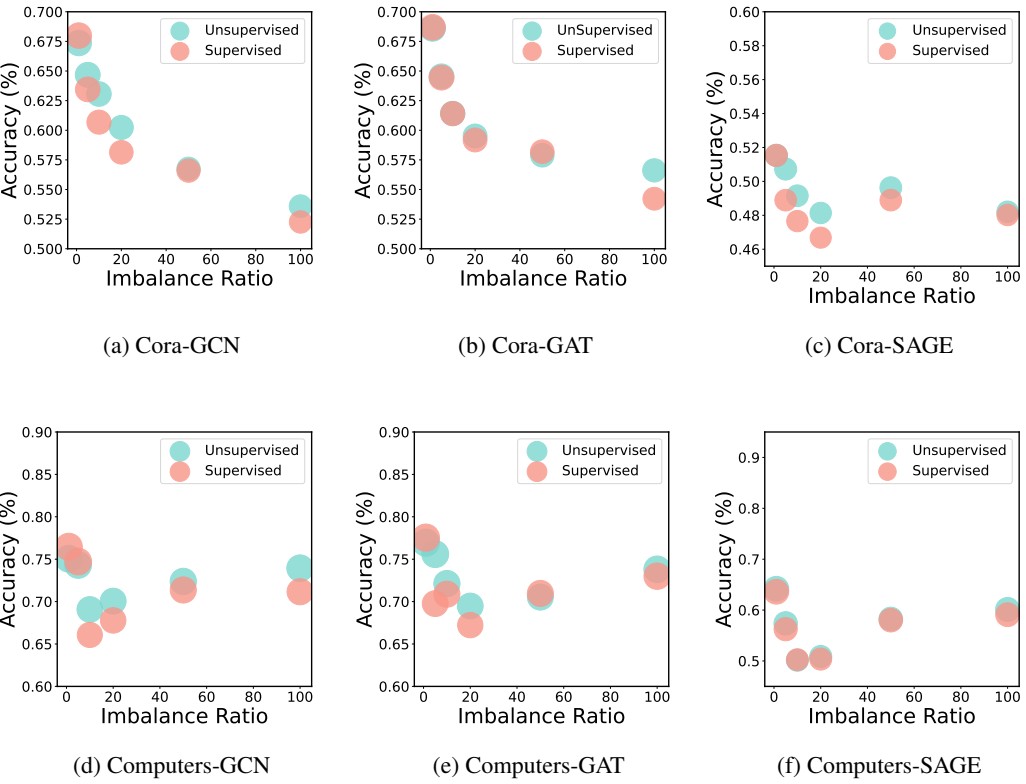

(a) Cora-GCN                   (b) Cora-GAT                   (c) Cora-SAGE

(d) Computers-GCN              (e) Computers-GAT              (f) Computers-SAGE

Figure 9: The experimental results on Cora and Amazon-Computers under different imbalance scenarios ($\rho = 1, 5, 10, 20, 50, 100$). We compare the accuracy of the two pseudo-labels (predictions) from unsupervised algorithms and supervised classifiers respectively for all unlabeled nodes.

**Analysis.** As depicted in Figure 9, the predictions generated by unsupervised algorithms maintain a high accuracy rate even in imbalanced scenarios. The final results unveil several intriguing insights: (1) In imbalanced scenarios, both supervised and unsupervised algorithms exhibit degraded performance, particularly in extreme cases ($\rho = 50, 100$). (2) The predictions derived from the embedding space outperform the biased classifier, indicating that the classifier is the weaker component when trained on an imbalanced training set. (3) Extensive experimental results demonstrate the significance

of predictions from unsupervised algorithms and classifiers, suggesting that relying on a single component does not lead to optimal performance.

## J.2 Detailed Analysis for DPAM

DPAM utilizes an unsupervised algorithm to derive pseudo-labels for each unlabeled node in the embedding space. Only unlabeled nodes with aligned pseudo-labels and classifier predictions are included in the candidate pool. This approach effectively mitigates the bias issue of the classifier, preventing the inclusion of low-quality nodes in the training set based on skewed confidence rankings. To gain a deeper understanding of DPAM's underlying mechanism, we conduct a set of novel experiments outlined below.

**Experimental Setup.** We use DPAM to filter the unlabeled nodes of the whole graph, and test the accuracy of pseudo-labels (prediction of the classifier) of the aligned node set $\mathcal{U}_{in}$ and the discarded node set $\mathcal{U}_{out}$ respectively. DPAM based on different GNN structures are trained on two node classification benchmark datasets, Cora, and Amazon-Computers. We process the two datasets with a traditional imbalanced distribution following [73, 37, 47]. The imbalance ratio $\rho$ between the numbers of the most frequent class and the least frequent class is set as 1, 5, 10, 20, 50, and 100. We fix architecture as the 2-layer GNN (i.e. GCN [25], GAT [54], GraphSAGE [12]) having 128 hidden dimensions and train models for 2000 epochs. We select the model by the validation accuracy. We observe the accuracy of pseudo labels for unlabeled nodes which are filtered out and absorbed into by DPAM respectively. We repeat each experiment five times and present the average experiment results in Table 28 and Table 29.

**Analysis.** DPAM partitions the unlabeled nodes of the entire graph into two subsets, namely, $\mathcal{U}_{in}$ and $\mathcal{U}_{out}$. The accuracy of pseudo-labels for these two subsets is examined to evaluate the effectiveness of DPAM. It is evident that the accuracy of pseudo-labels differs significantly between $\mathcal{U}_{in}$ and $\mathcal{U}_{out}$ in various imbalanced scenarios. Generally, the pseudo-label accuracy for $\mathcal{U}_{in}$ is high, while it is comparatively lower for $\mathcal{U}_{out}$, thereby validating the efficacy of DPAM. Moreover, as the imbalance ratio ($\rho$) increases, the accuracy of both subsets decreases, which reflects the model bias resulting from the imbalanced label distribution.

Table 28: Experimental results of DPAM effectiveness on Cora with $\rho = 1, 5, 10, 20, 50, 100$. We observe the accuracy (%) of the pseudo-label (prediction of the classifier) of the aligned node set $\mathcal{U}_{in}$ and the discarded node set $\mathcal{U}_{out}$ respectively. We report averaged results with the standard errors over 5 repetitions on three representative GNN architectures. All, Labeled, Unlabeled represent the size of whole nodes, labeled nodes, and unlabeled nodes on the graph. Align, Out, Align-True, Out-Ture represent the size of $\mathcal{U}_{in}, \mathcal{U}_{out}$, nodes with accurate pseudo-labels of $\mathcal{U}_{in}, \mathcal{U}_{out}$ respectively.

| | Dataset | All | Labled | Unlabeled | Align | Align-True | Accuracy(%) | Out | Out-True | Accuracy(%) |
|---|---|---|---|---|---|---|---|---|---|---|
| GCN | $\rho = 1$ | 2708 | 140 | 2568 | 2072.00 ± 10.29 | 1391.00 ± 22.56 | **67.11 ± 1.17** | 496.00 ± 10.29 | 233.80 ± 16.66 | **47.17 ± 3.74** |
| | $\rho = 5$ | 2708 | 92 | 2616 | 2122.80 ± 18.93 | 1392.00 ± 34.21 | **65.58 ± 1.57** | 493.20 ± 18.73 | 186.80 ± 13.08 | **37.86 ± 1.75** |
| | $\rho = 10$ | 2708 | 86 | 2622 | 2134.60 ± 23.42 | 1326.40 ± 24.23 | **62.14 ± 1.67** | 487.40 ± 23.43 | 181.60 ± 18.24 | **37.32 ± 3.13** |
| | $\rho = 20$ | 2708 | 83 | 2625 | 2149.60 ± 17.67 | 1310.20 ± 86.72 | **60.97 ± 3.50** | 475.40 ± 17.67 | 169.80 ± 21.47 | **35.64 ± 3.44** |
| | $\rho = 50$ | 2708 | 203 | 2505 | 1860.80 ± 31.15 | 1059.40 ± 58.77 | **56.90 ± 2.62** | 644.20 ± 31.14 | 225.80 ± 10.70 | **35.05 ± 3.79** |
| | $\rho = 100$ | 2708 | 403 | 2305 | 1820.40 ± 12.42 | 1001.60 ± 21.60 | **55.02 ± 3.99** | 484.60 ± 23.99 | 151.40 ± 20.74 | **31.78 ± 2.37** |
| GAT | $\rho = 1$ | 2708 | 140 | 2568 | 2072.00 ± 37.18 | 1412.40 ± 37.31 | **68.16 ± 1.41** | 496.00 ± 20.89 | 239.40 ± 11.37 | **48.29 ± 2.15** |
| | $\rho = 5$ | 2708 | 92 | 2616 | 2141.40 ± 26.36 | 1433.00 ± 59.82 | **66.90 ± 2.09** | 474.60 ± 26.36 | 195.20 ± 24.68 | **41.02 ± 3.27** |
| | $\rho = 10$ | 2708 | 86 | 2622 | 2132.60 ± 29.94 | 1377.40 ± 49.61 | **64.58 ± 1.60** | 489.40 ± 29.95 | 185.80 ± 12.28 | **37.97 ± 1.13** |
| | $\rho = 20$ | 2708 | 83 | 2625 | 2150.60 ± 37.35 | 1344.60 ± 54.17 | **62.16 ± 1.64** | 462.40 ± 33.28 | 178.00 ± 5.05 | **38.60 ± 2.12** |
| | $\rho = 50$ | 2708 | 140 | 2568 | 1892.40 ± 37.18 | 1080.80 ± 31.86 | **57.52 ± 1.52** | 612.60 ± 37.17 | 271.20 ± 6.30 | **44.35 ± 1.86** |
| | $\rho = 100$ | 2708 | 403 | 2305 | 1934.60 ± 19.65 | 1038.20 ± 21.08 | **53.66 ± 0.83** | 370.40 ± 37.17 | 147.53 ± 3.20 | **39.83 ± 1.36** |
| SAGE | $\rho = 1$ | 2708 | 140 | 2568 | 1944.00 ± 25.77 | 973.40 ± 32.26 | **51.27 ± 3.36** | 624.00 ± 25.77 | 237.00 ± 13.28 | **36.11 ± 4.07** |
| | $\rho = 5$ | 2708 | 92 | 2616 | 2004.40 ± 35.50 | 1038.20 ± 22.53 | **51.80 ± 3.73** | 611.60 ± 35.50 | 203.80 ± 7.15 | **33.40 ± 1.85** |
| | $\rho = 10$ | 2708 | 86 | 2622 | 2041.60 ± 32.48 | 1039.00 ± 41.32 | **50.89 ± 1.88** | 580.40 ± 32.48 | 189.20 ± 2.35 | **32.56 ± 4.25** |
| | $\rho = 20$ | 2708 | 83 | 2625 | 2040.20 ± 30.94 | 1002.20 ± 66.97 | **48.95 ± 2.66** | 578.80 ± 30.95 | 186.60 ± 18.00 | **32.18 ± 1.57** |
| | $\rho = 50$ | 2708 | 203 | 2505 | 1789.40 ± 30.56 | 870.20 ± 24.33 | **48.63 ± 1.03** | 715.60 ± 30.56 | 242.40 ± 16.77 | **33.87 ± 1.18** |
| | $\rho = 100$ | 2708 | 403 | 2305 | 1859.00 ± 192.42 | 914.41 ± 23.65 | **49.26 ± 2.59** | 446.00 ± 21.24 | 138.87 ± 6.32 | **31.15 ± 2.43** |

Table 29: Experimental results of DPAM effectiveness on Amazon-Computers with $\rho = 1, 5, 10, 20, 50, 100$. We observe the accuracy (%) of the pseudo-label (prediction of the classifier) of the aligned node set $\mathcal{U}_{in}$ and the discarded node set $\mathcal{U}_{out}$ respectively. We report averaged results with the standard errors over 5 repetitions on three representative GNN architectures. All, Labeled, Unlabeled represent the size of whole nodes, labeled nodes, and unlabeled nodes on the graph. Align, Out, Align-True, Out-Ture represent the size of $\mathcal{U}_{in}, \mathcal{U}_{out}$, nodes with accurate pseudo-labels of $\mathcal{U}_{in}, \mathcal{U}_{out}$ respectively.

| | Dataset | All | Labled | Unlabled | Align | Align-True | Accuracy(%) | Out | Out-True | Accuracy(%) |
|---|---|---|---|---|---|---|---|---|---|---|
| GCN | $\rho = 1$ | 13752 | 200 | 13552 | 11977.60 ± 108.09 | 9603.80 ± 93.34 | **80.08 ± 3.07** | 1554.40 ± 08.23 | 676.60 ± 141.11 | **43.58 ± 2.83** |
| | $\rho = 5$ | 13752 | 120 | 13632 | 11593.60 ± 73.16 | 9172.80 ± 87.32 | **79.06 ± 1.17** | 2308.40 ± 173.54 | 544.40 ± 66.26 | **30.74 ± 9.09** |
| | $\rho = 10$ | 13752 | 110 | 13642 | 11822.40 ± 13.43 | 8786.60 ± 55.48 | **74.24 ± 0.83** | 1807.60 ± 109.34 | 495.00 ± 100.37 | **27.24 ± 4.30** |
| | $\rho = 20$ | 13752 | 105 | 13647 | 11866.60 ± 17.34 | 8698.20 ± 188.13 | **73.40 ± 1.39** | 1780.40 ± 67.36 | 521.00 ± 60.76 | **29.20 ± 2.41** |
| | $\rho = 50$ | 13752 | 255 | 13497 | 11843.20 ± 168.20 | 8994.40 ± 175.24 | **75.94 ± 0.75** | 1653.80 ± 138.11 | 474.20 ± 50.72 | **28.68 ± 2.16** |
| | $\rho = 100$ | 13752 | 505 | 13247 | 9159.00 ± 192.42 | 7352.90 ± 61.23 | **81.41 ± 4.59** | 4088.00 ± 93.99 | 1129.60 ± 75.74 | **28.67 ± 4.77** |
| GAT | $\rho = 1$ | 13752 | 200 | 13552 | 12008.00 ± 101.93 | 9984.20 ± 308.03 | **83.44 ± 4.13** | 1544.80 ± 101.94 | 580.40 ± 190.49 | **43.33 ± 1.32** |
| | $\rho = 5$ | 13752 | 120 | 13632 | 11570.80 ± 136.11 | 8715.00 ± 86.33 | **75.33 ± 0.54** | 2061.20 ± 136.13 | 477.00 ± 97.07 | **25.39 ± 1.33** |
| | $\rho = 10$ | 13752 | 110 | 13642 | 8947.60 ± 13.40 | 6680.40 ± 177.54 | **75.85 ± 6.07** | 4694.40 ± 134.74 | 591.80 ± 13.74 | **15.94 ± 2.97** |
| | $\rho = 20$ | 13752 | 105 | 13647 | 10245.80 ± 68.00 | 7300.80 ± 64.89 | **71.42 ± 1.80** | 3401.20 ± 69.76 | 370.60 ± 43.87 | **18.52 ± 0.09** |
| | $\rho = 50$ | 13752 | 255 | 13497 | 10133.60 ± 31.56 | 7772.00 ± 155.87 | **77.17 ± 2.85** | 3363.40 ± 10.42 | 457.20 ± 108.19 | **19.28 ± 1.43** |
| | $\rho = 100$ | 13752 | 505 | 13247 | 11377.00 ± 63.32 | 9122.20 ± 96.70 | **80.46 ± 1.01** | 1910.00 ± 63.32 | 458.20 ± 41.04 | **24.78 ± 2.04** |
| SAGE | $\rho = 1$ | 13752 | 200 | 13552 | 10815.20 ± 86.50 | 7131.40 ± 72.83 | **65.94 ± 0.28** | 2736.80 ± 86.50 | 965.40 ± 56.42 | **35.26 ± 1.31** |
| | $\rho = 5$ | 13752 | 120 | 13632 | 10627.80 ± 78.33 | 6728.00 ± 53.24 | **63.25 ± 0.36** | 3004.20 ± 78.03 | 978.20 ± 59.93 | **32.55 ± 1.49** |
| | $\rho = 10$ | 13752 | 110 | 13642 | 10475.00 ± 118.41 | 6015.00 ± 41.14 | **57.43 ± 4.01** | 3167.00 ± 18.41 | 1064.40 ± 52.71 | **33.59 ± 6.23** |
| | $\rho = 20$ | 13752 | 105 | 13647 | 10653.20 ± 87.35 | 5998.40 ± 69.35 | **56.30 ± 4.01** | 2993.80 ± 87.35 | 886.20 ± 73.25 | **29.57 ± 1.77** |
| | $\rho = 50$ | 13752 | 255 | 13497 | 11044.80 ± 129.14 | 6760.80 ± 50.26 | **61.22 ± 3.42** | 2442.20 ± 28.48 | 879.00 ± 91.45 | **35.71 ± 1.78** |
| | $\rho = 100$ | 13752 | 505 | 13247 | 9175.20 ± 32.53 | 6475.60 ± 80.88 | **72.07 ± 1.96** | 4071.80 ± 32.63 | 1218.60 ± 14.70 | **34.43 ± 1.08** |

### J.3 Detailed Results and Analysis about Fluctuation of RBO Values in Node-reordering

In Section 4.2, we argue that the classifier's confidence becomes increasingly valuable as the iteration progresses, gradually balancing the training set, whereas the geometric rankings are determined in the embedding space and remain unaffected by the classifier. Consequently, we can trust that the similarities between the Confidence Rankings and the Geometric Rankings will gradually increase as the confidence gains credibility throughout the iterative process. It is worth noting that the unsupervised algorithm performs inferiorly compared to supervised methods, particularly when dealing with a balanced training set. Therefore, by leveraging the combined features of both rankings, we can significantly enhance the performance of our algorithm. To validate the aforementioned hypothesis, experiments are conducted.

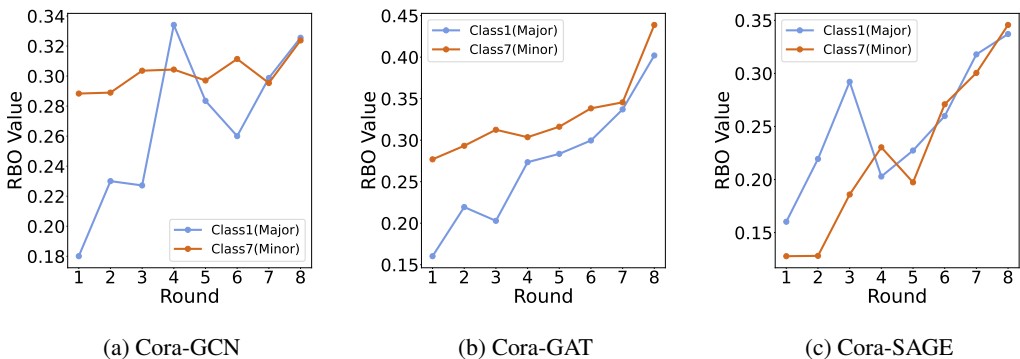

(a) Cora-GCN          (b) Cora-GAT          (c) Cora-SAGE

Figure 10: Fluctuation of RBO values ($\rho = 10$) of two rankings as iterations progress.

**Experimental Setup.** We conduct more experiments on Cora ($\rho = 10$) to observe the similarities between the Geometric Rankings and Confidence Rankings. The architecture is fixed as the 2-layer GNN (i.e. GCN [25], GAT [54], GraphSAGE [12]) having 128 hidden dimensions and train models

for 2000 epochs. The UNREAL model's hyperparameter settings can be found in Appendix K.4. We choose a majority and a minority class at random to compare the similarities of their respective two rankings (our setting is the first class and the last class of Cora), and we limit the number of iterations to eight. Each experiment is repeated five times, and the average experiment results are reported in Figure 10.

**Analysis.** As depicted in Figure 10, it is evident that the similarities between the Confidence Rankings and the Geometric Rankings exhibit a gradual increase during the initial stages of iteration. This observation substantiates our hypothesis. It is noteworthy that, as the training set becomes gradually balanced, the similarity between the two rankings of the minority class surpasses that of the majority class. This finding further emphasizes the compensatory advantage of UNREAL for the minority class.

### J.4 Detailed Analysis for Node-Reordering and DGIS

**Experimental Setup.** We conduct experiments to test the accuracy of pseudo labels for unlabeled nodes on class-imbalanced graphs. All model combinations based on different GNN structures are trained on two node classification benchmark datasets, Cora, and Amaon-Computers. We process the two datasets with a traditional imbalanced distribution following [73, 37, 47]. The imbalance ratio $\rho$ between the numbers of the most frequent class and the least frequent class is set as 1, 5, 10, 20, 50, and 100. We fix architecture as the 2-layer GNN (i.e. GCN [25], GAT [54], GraphSAGE [12]) having 128 hidden dimensions and train models for 2000 epochs. We select the model by the validation accuracy. We observe the accuracy of pseudo labels for unlabeled nodes which are newly added to the minority class of the training set. We repeat each experiment five times and present the average experiment results in Table 30 and Table 31.

**Analysis.** As shown in Table 30 and Table 31, we validate the efficacy of each component of our framework by assessing the accuracy of the selected pseudo-labels for nodes using various model combinations, namely DPAM+Confidence ranking (with or without DGIN), DPAM+Geometric ranking (with or without DGIS), and DPAM + Node-Reordering (with or without DGIS). Notably, across different imbalanced scenarios, both components of our framework (Node-reordering and DGIS) demonstrate significant importance, resulting in superior performance compared to other model combinations.

### J.5 Detailed Ablation Analysis

Considering the space limitations of the main paper, we present the detailed ablation analysis herein. In this section, we conduct ablation studies to analyze the individual contributions of each component in our proposed method. The results from Appendix J.1 have already confirmed the necessity of incorporating unsupervised learning in the embedding space. Therefore, in this section, DPAM is applied in all comparative methods. We evaluate the performance of three different ranking techniques: confidence ranking, geometric ranking, and Node-reordering (which combines the former two rankings using information retrieval techniques). Additionally, we examine the impact of DGIS, which aims to mitigate the presence of geometrically imbalanced nodes. As illustrated in Table 32, each component of our method demonstrates performance improvements. Notably, in three out of the four settings presented in the table, Node-Reordering + DGIS achieves the highest F1 scores. Furthermore, across all cases, geometric ranking consistently outperforms confidence ranking, supporting our hypothesis that confidence scores may be biased and less reliable.

### J.6 In-Depth Comparison between Self-Training and Our Method

#### J.6.1 The Comparison of Accuracy of Pseudo Labels for Self-Training and Our Methods

We provide complete evaluation results on more benchmark datasets, where more basic models are included in addition to the reported results in the main paper.

**Experimental Setup.** Since true labels for all benchmark nodes are provided, we first conduct experiments to test the accuracy of pseudo labels for unlabeled nodes on class-imbalanced graphs inventively. We select top 100 unlabeled nodes newly added to the training set through ST & Ours , and evaluate the performance of ST & Ours by testing the accuracy (%) with the standard errors of these nodes' pseudo labels. We test unlabeled nodes that are selected into the minority classes and

Table 30: Analyzed experimental results of Node-Reordering and DGIS on *Cora* with $\rho = 1, 5, 10, 20, 50, 100$. We select 100 unlabeled nodes newly added to the minority class of training set through different method combinations, and evaluate the validity of Node-Reordering & DGIS by testing the accuracy (%) with the standard errors of the pseudo labels for these nodes. We report averaged results over 5 repetitions on three representative GNN architectures.

| | Dataset | Cora | | | | | |
|---|---|---|---|---|---|---|---|
| | Imbalance Ratio ($\rho$) | $\rho = 1$ | $\rho = 5$ | $\rho = 10$ | $\rho = 20$ | $\rho = 50$ | $\rho = 100$ |
| GCN | DPAM+Confidence Ranking | $61.40 \pm 2.73$ | $62.40 \pm 2.59$ | $60.20 \pm 1.02$ | $58.40 \pm 1.05$ | $57.60 \pm 1.86$ | $58.40 \pm 2.15$ |
| | DPAM+Geometric Ranking | $64.00 \pm 3.67$ | $61.20 \pm 2.89$ | $61.20 \pm 2.54$ | $63.60 \pm 1.31$ | $55.60 \pm 2.31$ | $47.80 \pm 2.87$ |
| | DPAM+Node-Reordering | $89.65 \pm 3.23$ | $86.98 \pm 0.21$ | $88.32 \pm 0.83$ | $85.32 \pm 2.98$ | $90.87 \pm 2.31$ | $71.60 \pm 2.91$ |
| | DPAM+Confidence Ranking + DGIS | $71.00 \pm 5.47$ | $75.40 \pm 2.15$ | $68.20 \pm 1.25$ | $69.40 \pm 1.28$ | $67.80 \pm 2.75$ | $66.60 \pm 0.16$ |
| | DPAM+Geometric Ranking + DGIS | $69.60 \pm 3.78$ | $73.80 \pm 0.45$ | $64.80 \pm 1.26$ | $64.20 \pm 1.91$ | $57.00 \pm 1.57$ | $69.00 \pm 1.71$ |
| | **DPAM+Node-Reordering + DGIS (Ours)** | $\mathbf{92.80 \pm 1.30}$ | $\mathbf{96.40 \pm 4.27}$ | $\mathbf{92.20 \pm 0.85}$ | $\mathbf{89.40 \pm 1.37}$ | $\mathbf{93.00 \pm 0.82}$ | $\mathbf{77.80 \pm 2.50}$ |
| GAT | DPAM+Confidence Ranking | $61.60 \pm 4.26$ | $64.00 \pm 2.07$ | $62.60 \pm 3.47$ | $57.80 \pm 1.65$ | $58.20 \pm 1.07$ | $60.60 \pm 0.79$ |
| | DPAM+Geometric Ranking | $64.00 \pm 2.78$ | $67.80 \pm 3.76$ | $65.00 \pm 4.30$ | $52.00 \pm 1.02$ | $65.20 \pm 2.58$ | $40.80 \pm 2.63$ |
| | DPAM+Node-Reordering | $91.79 \pm 0.23$ | $90.45 \pm 5.78$ | $84.32 \pm 3.45$ | $88.34 \pm 0.23$ | $90.32 \pm 0.43$ | $75.34 \pm 1.54$ |
| | DPAM+Confidence Ranking + DGIS | $69.80 \pm 2.77$ | $72.80 \pm 3.94$ | $72.40 \pm 1.13$ | $67.60 \pm 1.59$ | $71.60 \pm 9.12$ | $64.00 \pm 1.74$ |
| | DPAM+Geometric Ranking + DGIS | $73.60 \pm 4.82$ | $74.00 \pm 5.47$ | $68.40 \pm 1.62$ | $57.20 \pm 2.17$ | $68.00 \pm 1.17$ | $62.00 \pm 1.53$ |
| | **DPAM+Node-Reordering + DGIS (Ours)** | $\mathbf{93.80 \pm 1.92}$ | $\mathbf{91.20 \pm 4.60}$ | $\mathbf{90.40 \pm 1.69}$ | $\mathbf{90.00 \pm 9.92}$ | $\mathbf{94.60 \pm 4.92}$ | $\mathbf{78.20 \pm 2.47}$ |
| SAGE | DPAM+Confidence Ranking | $54.80 \pm 4.96$ | $53.00 \pm 2.46$ | $51.80 \pm 1.97$ | $43.60 \pm 2.57$ | $46.20 \pm 0.53$ | $41.60 \pm 1.14$ |
| | DPAM+Geometric Ranking | $53.60 \pm 2.78$ | $45.40 \pm 1.75$ | $40.60 \pm 0.26$ | $52.60 \pm 2.47$ | $47.40 \pm 4.27$ | $44.80 \pm 2.84$ |
| | DPAM+Node-Reordering | $90.69 \pm 0.21$ | $86.90 \pm 0.56$ | $86.45 \pm 3.21$ | $88.34 \pm 2.43$ | $75.34 \pm 4.20$ | $76.43 \pm 1.43$ |
| | DPAM+Confidence Ranking + DGIS | $66.20 \pm 5.78$ | $59.00 \pm 3.04$ | $63.80 \pm 1.52$ | $54.60 \pm 1.64$ | $60.60 \pm 1.37$ | $57.40 \pm 2.26$ |
| | DPAM+Geometric Ranking + DGIS | $61.60 \pm 3.71$ | $61.80 \pm 5.21$ | $54.00 \pm 7.31$ | $53.60 \pm 1.38$ | $63.00 \pm 1.23$ | $45.20 \pm 1.96$ |
| | **DPAM+Node-Reordering + DGIS (Ours)** | $\mathbf{97.80 \pm 1.78}$ | $\mathbf{92.20 \pm 1.32}$ | $\mathbf{90.80 \pm 1.82}$ | $\mathbf{89.20 \pm 1.39}$ | $\mathbf{94.20 \pm 8.04}$ | $\mathbf{85.40 \pm 1.02}$ |

Table 31: Analyzed experimental results of Node-Reordering and DGIS on Amazon-Computers with $\rho = 1, 5, 10, 20, 50, 100$. We select 100 unlabeled nodes newly added to the minority class of training set through different method combinations, and evaluate the validity of Node-Reordering & DGIS by testing the accuracy (%) with the standard errors of the pseudo labels for these nodes. We report averaged results over 5 repetitions on three representative GNN architectures.

| | Dataset | Amazon-Computers | | | | | |
|---|---|---|---|---|---|---|---|
| | Imbalance Ratio ($\rho$) | $\rho = 1$ | $\rho = 5$ | $\rho = 10$ | $\rho = 20$ | $\rho = 50$ | $\rho = 100$ |
| GCN | DPAM+Confidence Ranking | $75.40 \pm 2.50$ | $70.20 \pm 3.03$ | $74.88 \pm 3.11$ | $68.20 \pm 4.20$ | $63.60 \pm 2.30$ | $61.40 \pm 1.51$ |
| | DPAM+Geometric Ranking | $76.00 \pm 1.41$ | $74.80 \pm 4.71$ | $76.80 \pm 2.28$ | $65.80 \pm 3.27$ | $64.80 \pm 3.70$ | $65.60 \pm 3.98$ |
| | DPAM+Node-Reordering | $82.80 \pm 2.38$ | $79.60 \pm 3.64$ | $78.20 \pm 0.26$ | $74.00 \pm 3.28$ | $65.20 \pm 1.87$ | $66.00 \pm 2.82$ |
| | DPAM+Confidence Ranking + DGIS | $76.40 \pm 2.07$ | $67.20 \pm 4.32$ | $75.80 \pm 2.38$ | $66.20 \pm 3.70$ | $62.80 \pm 0.12$ | $59.20 \pm 1.30$ |
| | DPAM+Geometric Ranking + DGIS | $78.20 \pm 0.83$ | $80.00 \pm 1.22$ | $76.40 \pm 1.67$ | $66.00 \pm 2.44$ | $64.20 \pm 3.83$ | $66.20 \pm 2.38$ |
| | **DPAM+Node-Reordering + DGIS (Ours)** | $\mathbf{84.40 \pm 3.60}$ | $\mathbf{82.20 \pm 2.16}$ | $\mathbf{80.40 \pm 3.46}$ | $\mathbf{80.60 \pm 1.51}$ | $\mathbf{69.60 \pm 3.04}$ | $\mathbf{66.40 \pm 3.20}$ |
| GAT | DPAM+Confidence Ranking | $84.60 \pm 2.40$ | $79.20 \pm 1.78$ | $73.00 \pm 2.12$ | $74.80 \pm 2.16$ | $65.00 \pm 1.73$ | $68.60 \pm 1.40$ |
| | DPAM+Geometric Ranking | $86.00 \pm 3.80$ | $79.80 \pm 2.94$ | $74.80 \pm 3.42$ | $75.00 \pm 2.91$ | $70.80 \pm 2.16$ | $69.40 \pm 1.10$ |
| | DPAM+Node-Reordering | $87.40 \pm 2.30$ | $80.60 \pm 3.04$ | $80.40 \pm 2.19$ | $79.00 \pm 3.67$ | $75.00 \pm 1.22$ | $73.40 \pm 2.52$ |
| | DPAM+Confidence Ranking + DGIS | $84.20 \pm 1.64$ | $79.40 \pm 2.07$ | $76.40 \pm 6.50$ | $76.00 \pm 2.34$ | $66.00 \pm 0.12$ | $72.00 \pm 1.84$ |
| | DPAM+Geometric Ranking + DGIS | $83.80 \pm 1.09$ | $80.20 \pm 1.09$ | $76.20 \pm 2.28$ | $77.80 \pm 2.58$ | $71.60 \pm 0.89$ | $69.00 \pm 1.16$ |
| | **DPAM+Node-Reordering + DGIS (Ours)** | $\mathbf{89.00 \pm 2.54}$ | $\mathbf{86.60 \pm 2.50}$ | $\mathbf{85.60 \pm 4.44}$ | $\mathbf{83.40 \pm 3.31}$ | $\mathbf{78.00 \pm 3.39}$ | $\mathbf{79.80 \pm 3.03}$ |
| SAGE | DPAM+Confidence Ranking | $85.20 \pm 3.38$ | $80.20 \pm 6.26$ | $84.8 \pm 0.83$ | $77.60 \pm 0.89$ | $61.00 \pm 0.70$ | $65.40 \pm 2.65$ |
| | DPAM+Geometric Ranking | $86.00 \pm 0.70$ | $81.20 \pm 2.16$ | $83.40 \pm 1.14$ | $78.00 \pm 1.22$ | $61.40 \pm 0.54$ | $65.00 \pm 1.72$ |
| | DPAM+Node-Reordering | $86.00 \pm 1.58$ | $83.20 \pm 3.27$ | $84.60 \pm 0.54$ | $79.20 \pm 1.92$ | $61.80 \pm 0.44$ | $67.80 \pm 1.03$ |
| | DPAM+Confidence Ranking + DGIS | $86.40 \pm 2.07$ | $81.60 \pm 3.20$ | $83.40 \pm 1.14$ | $\mathbf{79.20 \pm 0.44}$ | $61.20 \pm 0.44$ | $70.40 \pm 3.59$ |
| | DPAM+Geometric Ranking + DGIS | $87.00 \pm 2.12$ | $80.80 \pm 2.48$ | $84.20 \pm 1.30$ | $78.20 \pm 1.48$ | $61.20 \pm 0.47$ | $68.20 \pm 1.72$ |
| | **DPAM+Node-Reordering + DGIS (Ours)** | $\mathbf{88.20 \pm 2.16}$ | $\mathbf{87.60 \pm 1.14}$ | $\mathbf{85.40 \pm 4.72}$ | $78.00 \pm 1.55$ | $\mathbf{66.20 \pm 2.86}$ | $\mathbf{72.20 \pm 0.83}$ |

Table 32: Ablation analysis on different components.

| Modules | Confidence ranking | Geometric ranking | Node-reordering | DGIS | F1 |
|---|---|---|---|---|---|
| Cora+GCN ($\rho = 10$) | ✗ | ✓ | ✓ | ✓ | 73.93 ± 0.95 |
| | ✗ | ✓ | ✓ | ✗ | 72.74 ± 0.63 |
| | ✓ | ✗ | ✓ | ✓ | 75.85 ± 0.82 |
| | ✓ | ✗ | ✓ | ✗ | 75.34 ± 0.63 |
| | ✓ | ✓ | ✗ | ✓ | 75.00 ± 0.97 |
| | ✓ | ✓ | ✗ | ✗ | **76.44 ± 1.06** |
| CiteSeer+SAGE ($\rho = 20$) | ✗ | ✓ | ✓ | ✓ | 46.09 ± 4.08 |
| | ✗ | ✓ | ✓ | ✗ | 47.76 ± 1.06 |
| | ✓ | ✗ | ✓ | ✓ | 50.32 ± 3.75 |
| | ✓ | ✗ | ✓ | ✗ | 53.32 ± 3.75 |
| | ✓ | ✓ | ✗ | ✓ | 55.43± 2.14 |
| | ✓ | ✓ | ✗ | ✗ | **57.51 ± 4.92** |
| PubMed+GAT ($\rho = 50$) | ✗ | ✓ | ✓ | ✓ | 76.34 ± 0.39 |
| | ✗ | ✓ | ✓ | ✗ | 75.42 ± 0.39 |
| | ✓ | ✗ | ✓ | ✓ | 77.32 ± 0.21 |
| | ✓ | ✗ | ✓ | ✗ | 76.89 ± 1.43 |
| | ✓ | ✓ | ✗ | ✓ | 76.12 ± 2.63 |
| | ✓ | ✓ | ✗ | ✗ | **77.38 ± 0.39** |
| Computers+GAT ($\rho = 100$) | ✗ | ✓ | ✓ | ✓ | 70.86 ± 1.73 |
| | ✗ | ✓ | ✓ | ✗ | 68.86 ± 1.42 |
| | ✓ | ✗ | ✓ | ✓ | 72.32 ± 2.43 |
| | ✓ | ✗ | ✓ | ✗ | 73.65 ± 0.67 |
| | ✓ | ✓ | ✗ | ✓ | 74.03 ± 2.53 |
| | ✓ | ✓ | ✗ | ✗ | **75.83 ± 0.74** |

unlabeled nodes that are selected into the majority classes separately. We evaluate the performance of each method on Cora, CiteSeer, PubMed, Amazon-Computers under different imbalance scenarios. We process the datasets with a traditional imbalanced distribution following [73, 37, 47]. The imbalance ratio $\rho$ between the numbers of the most frequent class and the least frequent class is set as 1 (balanced), 5, 10, 20, 50, 100. We fix architecture as the 2-layer GNN (i.e. GCN [25], GAT [54], GraphSAGE [12]) having 128 hidden dimensions and train models for 2000 epochs. The validation accuracy is used to select the model. Each experiment is repeated five times, and the average experiment results are reported in Figure 11.

**Analysis.** It is observed that as the parameter $\rho$ increases, the accuracy of pseudo labels selected into minority classes decreases, indicating influence of classifier bias is amplified. The end results demonstrate a number of intriguing aspects. (1) Highly imbalanced scenarios render the pseudo labels generated by the classifier unreliable. Moreover, the addition of unlabeled nodes with pseudo labels corresponding to minority classes in the training set introduces excessive noise during the ST process. (2) It is important to highlight that the evaluation of ST focuses solely on the top 100 nodes selected based on Confidence Rankings (Section 4.2). So, we believe that even if a node's pseudo label is correct, the classifier's confidence is skewed, resulting in the inclusion of low-quality unlabeled nodes in the training set while overlooking high-quality unlabeled nodes. This factor likely contributes to the underperformance of ST in imbalanced scenarios. (3) Irrespective of selecting majority class nodes or minority class nodes, our approach consistently outperforms ST. A thorough examination of Figure 11 clearly indicates the significant superiority of ours over ST, with the gap in F1 scores between the two methods widening as the imbalance ratio increases.

### J.6.2 F1 Score Performance for Self-Training and Our Method

**Experimental Setup.** We utilized three citation datasets, Cora, CiteSeer, and PubMed, to construct scenarios representing varying degrees of class imbalance. Specifically, we select half of the classes as minority classes and convert a randomly selected subset of labeled nodes into unlabeled nodes until the training set achieved the desired imbalance ratio ($\rho$). The architecture we employed consisted of a 2-layer graph neural network (GNN) with 128 hidden dimensions, using GCN [25], GAT [54], or GraphSAGE [12]. The models were trained for 2000 epochs. As for the Self-Training (ST) method, the size of added nodes for each class was treated as a hyperparameter, which we fine-tune based on the accuracy of the validation set. Each experiment was repeated five times, and the average results were reported for different imbalance ratios, as depicted in Figure 12.

**Analysis.** The findings presented in Figure 12 demonstrate the efficacy of ST in improving imbalanced learning across the Cora, CiteSeer, and PubMed datasets under three GNN architectures. The results consistently reveal that ST surpasses the performance of the vanilla model across different ratios and datasets, indicating the usefulness of unlabeled nodes. Nevertheless, the study uncovers a gradual decline in ST's performance in heavily imbalanced scenarios, particularly for Cora and CiteSeer. We posit that this diminished performance in highly imbalanced data stems from the biased and unreliable predictions of classifiers, leading to the inclusion of low-quality nodes in the training set at an early stage.

### J.6.3 Discussion on the Choice of Clustering Method in Our Approach

Although embeddings in our model are approximately distributed on a hypersphere, we employ K-Means clustering for its computational efficiency and robustness. While vMF mixture models or other manifold-aware approaches (e.g., spectral clustering or cosine-based K-Means) are theoretically well-suited for such data, they involve expensive iterative optimization steps (such as Expectation–Maximization with Bessel function evaluations) and are less scalable to large graphs. In practice, we observe that normalized embeddings already exhibit meaningful Euclidean structure, allowing K-Means to perform effectively without significant overhead. Our framework remains modular, and replacing K-Means with vMF or cosine-based variants can be considered as future work.

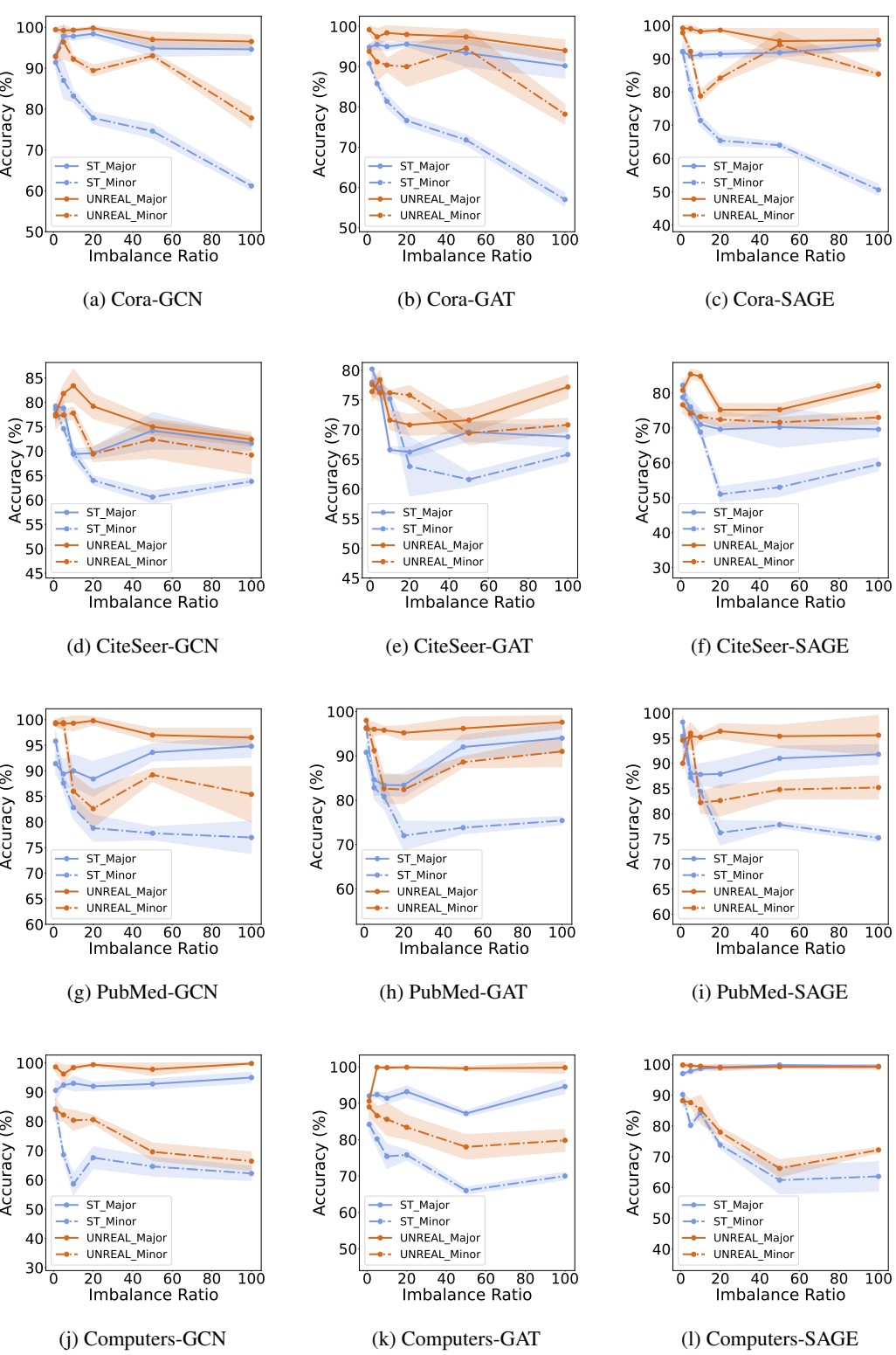

Figure 11: Here, we present the experimental results from four benchmark datasets under various imbalance scenarios. We select top 100 unlabeled nodes newly added to the training set via ST & Ours, and evaluate the performance of ST & Ours based on three GNN architectures by testing the accuracy with the standard errors of these nodes' pseudo labels. Minor means that we only test unlabeled nodes which are selected into the minority classes, and Major means that we only test unlabeled nodes which are selected into the majority classes.

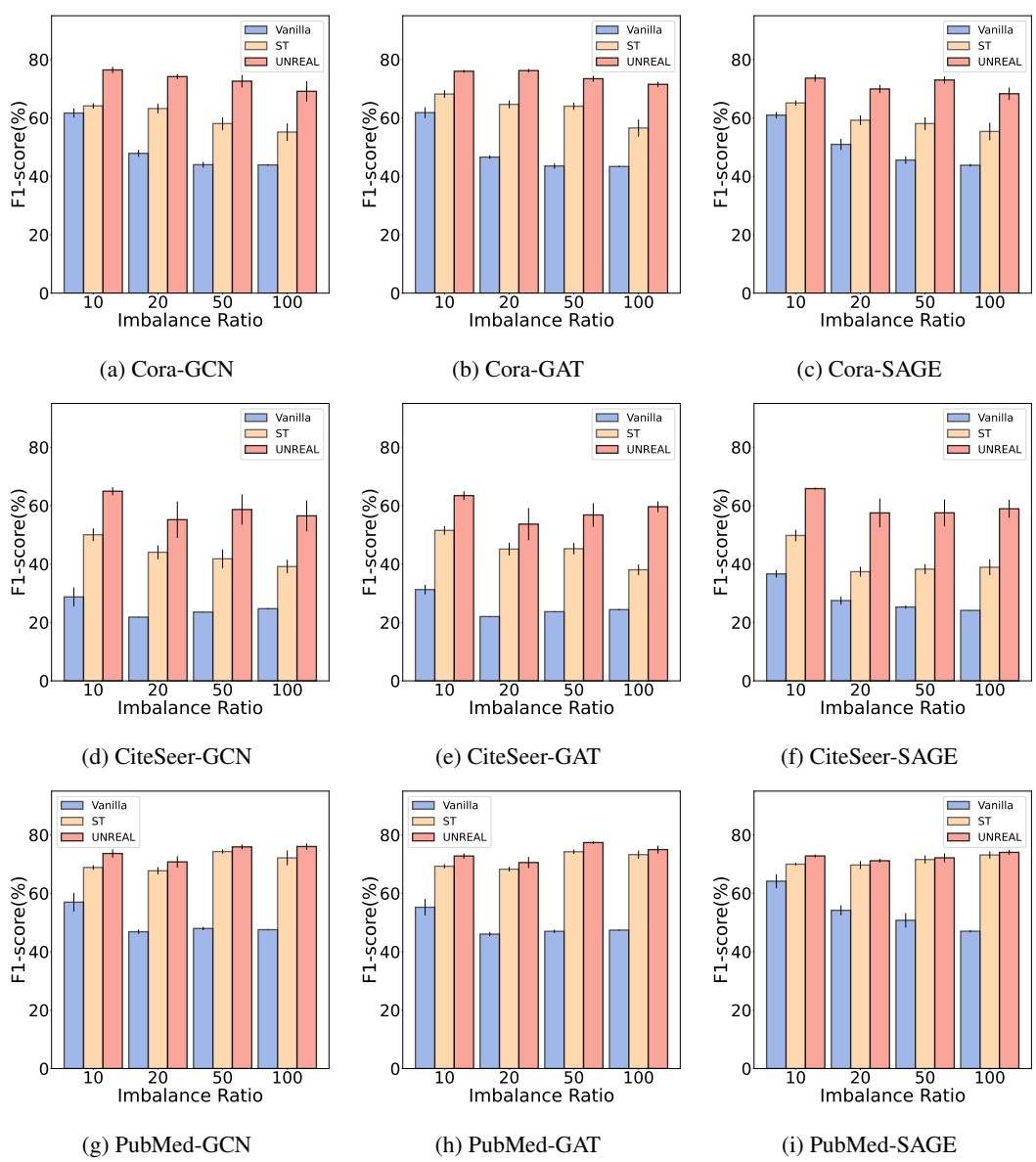

Figure 12: The experimental results on the three citation datasets under different imbalance scenarios ($\rho = 10, 20, 50, 100$). We report the F1-score (%) with the standard errors of Vanilla, ST, and Our Method.

# K Experimental Setup

In this section, we introduce the method of imbalanced datasets construction, evaluation protocol, and the details of our algorithm and baseline methods.

## K.1 Imbalanced Datasets Construction

Table 33: Summary of the datasets used in our experiments.

| Dataset | Nodes | Edges | Features | Classes |
|---|---|---|---|---|
| Cora | 2,708 | 5,429 | 1,433 | 7 |
| Citeseer | 3,327 | 4,732 | 3,703 | 6 |
| Pubmed | 19,717 | 44,338 | 500 | 3 |
| Amazon-Computers | 13,752 | 491,722 | 767 | 10 |
| Coauthor-CS | 18,333 | 163,788 | 6805 | 15 |
| Flickr | 89,250 | 899,756 | 500 | 7 |
| Obgn-Arxiv | 169343 | 1166243 | 128 | 40 |

Table 34: Label distributions on the whole graphs.

| Dataset | $\mathcal{C}_0$ | $\mathcal{C}_1$ | $\mathcal{C}_2$ | $\mathcal{C}_3$ | $\mathcal{C}_4$ | $\mathcal{C}_5$ | $\mathcal{C}_6$ | $\mathcal{C}_7$ | $\mathcal{C}_8$ | $\mathcal{C}_9$ | $\mathcal{C}_{10}$ | $\mathcal{C}_{11}$ | $\mathcal{C}_{12}$ | $\mathcal{C}_{13}$ | $\mathcal{C}_{14}$ |
|---|---|---|---|---|---|---|---|---|---|---|---|---|---|---|---|
| Cora ($\rho \approx 4.54$) | 351 | 217 | 418 | 818 | 426 | 298 | 180 | - | - | - | - | - | - | - | - |
| CiteSeer ($\rho \approx 2.66$) | 264 | 590 | 668 | 701 | 696 | 508 | - | - | - | - | - | - | - | - | - |
| PubMed ($\rho \approx 1.91$) | 4103 | 7739 | 7835 | - | - | - | - | - | - | - | - | - | - | - | - |
| Amazon-Computers ($\rho \approx 17.73$) | 436 | 2142 | 1414 | 542 | 5158 | 308 | 487 | 818 | 2156 | 291 | - | - | - | - | - |
| Coauthor-CS ($\rho \approx 35.05$) | 708 | 462 | 2050 | 429 | 1394 | 2193 | 371 | 924 | 775 | 118 | 1444 | 2033 | 420 | 4136 | 876 |
| Flickr ($\rho \approx 10.84$) | 5264 | 8506 | 6413 | 4903 | 22966 | 3479 | 37719 | - | - | - | - | - | - | - | - |

Table 35: Ogbn-arxiv label statistics: node counts per class in the full graph, training set, and test set.

| Class | Whole | Train | Test | Class | Whole | Train | Test | Class | Whole | Train | Test | Class | Whole | Train | Test |
|---|---|---|---|---|---|---|---|---|---|---|---|---|---|---|---|
| $\mathcal{C}_0$ | 565 | 437 | 54 | $\mathcal{C}_{10}$ | 7869 | 5182 | 1455 | $\mathcal{C}_{20}$ | 2076 | 1495 | 313 | $\mathcal{C}_{30}$ | 11814 | 4334 | 4631 |
| $\mathcal{C}_1$ | 687 | 382 | 187 | $\mathcal{C}_{11}$ | 750 | 391 | 239 | $\mathcal{C}_{21}$ | 393 | 304 | 51 | $\mathcal{C}_{31}$ | 2828 | 1350 | 892 |
| $\mathcal{C}_2$ | 4839 | 3604 | 733 | $\mathcal{C}_{12}$ | 79 | 21 | 5 | $\mathcal{C}_{22}$ | 1903 | 1268 | 386 | $\mathcal{C}_{32}$ | 411 | 270 | 83 |
| $\mathcal{C}_3$ | 2080 | 1014 | 654 | $\mathcal{C}_{13}$ | 2358 | 1290 | 628 | $\mathcal{C}_{23}$ | 2834 | 1539 | 808 | $\mathcal{C}_{33}$ | 1271 | 926 | 220 |
| $\mathcal{C}_4$ | 5832 | 2864 | 1869 | $\mathcal{C}_{14}$ | 597 | 433 | 71 | $\mathcal{C}_{24}$ | 22187 | 6989 | 10740 | $\mathcal{C}_{34}$ | 7867 | 5436 | 1414 |
| $\mathcal{C}_5$ | 4958 | 2933 | 1246 | $\mathcal{C}_{15}$ | 403 | 248 | 87 | $\mathcal{C}_{25}$ | 1257 | 457 | 475 | $\mathcal{C}_{35}$ | 127 | 25 | 36 |
| $\mathcal{C}_6$ | 1618 | 703 | 622 | $\mathcal{C}_{16}$ | 27321 | 9948 | 10471 | $\mathcal{C}_{26}$ | 4605 | 2834 | 1041 | $\mathcal{C}_{36}$ | 3524 | 2506 | 627 |
| $\mathcal{C}_7$ | 589 | 380 | 134 | $\mathcal{C}_{17}$ | 515 | 202 | 203 | $\mathcal{C}_{27}$ | 4801 | 1661 | 2066 | $\mathcal{C}_{37}$ | 2369 | 1615 | 481 |
| $\mathcal{C}_8$ | 6232 | 4056 | 1250 | $\mathcal{C}_{18}$ | 749 | 402 | 209 | $\mathcal{C}_{28}$ | 21406 | 16284 | 2849 | $\mathcal{C}_{38}$ | 1507 | 1100 | 214 |
| $\mathcal{C}_9$ | 2820 | 2245 | 345 | $\mathcal{C}_{19}$ | 2877 | 1873 | 419 | $\mathcal{C}_{29}$ | 416 | 239 | 120 | $\mathcal{C}_{39}$ | 2009 | 1551 | 269 |

The detailed descriptions of the datasets are shown in Table 33. For each citation dataset, for $\rho = 10, 20$, we follow the "public" split, and randomly convert minority class nodes to unlabeled nodes until the dataset reaches an imbalanced ratio $\rho$. For $\rho = 50, 100$, since there are not enough nodes per class in the public split training set, we choose randomly selected nodes as training samples, and for validation and test sets we still follow the public split. For the co-purchased networks Amazon-Computers, we randomly select nodes as training set in each replicated experiment, construct a random validation set with 30 nodes in each class and treat the remaining nodes as the testing set. For Flickr, we follow the dataset split from [69]. For Computers-Random, we build a training set of equal proportions based on the label distribution of the entire graph (Amazon-Computers). The label distribution in the training set for Computers-Random is summarized in Table 36. The details of label distribution in the training set of the five imbalanced benchmark datasets are in Table 36, and the label distribution of the full graph is provided in Table 34.

## K.2 Details of GNNs

We evaluate our method with three classic GNN architectures, namely GCN [25], GAT [54], and GraphSAGE [12]. GNN consists of $L = 1, 2, 3, 4$ layers, and each GNN layer is followed by a BatchNorm layer (momentum = 0.99) and a PRelu activation [15]. For GAT, we adopt multi-head attention with 8 heads. We search for the best model on the validation set. The choices of the hidden unit size for each layer are 64, 128, and 256.

### K.3 Evaluation Protocol

We adopt Adam [24] optimizer with an initial learning rate of 0.01 or 0.005. We follow [47] to devise a scheduler, which cuts the learning rate by half if there is no decrease in validation loss for 100 consecutive epochs. All learnable parameters in the model adopt weight decay with a rate of 0.0005. For the first training iteration, we train the model for 200 epochs using the original training set for Cora, CiteSeer, PubMed, or Amazon-Computers. For Flickr, we train for 2000 epochs in the first iteration. We train models for 2000 epochs in the rest of the iteration with the above optimizer and scheduler. The best models are selected based on validation accuracy. Early stopping is used with patience set to 300.

### K.4 Implementation Details

In UNREAL, we employ the vanilla K-means algorithm as the unsupervised clustering method. The number of clusters $K$ is chosen from $\{100, 200, 250, 300, 350, 400, 450, 500\}$ for Cora, CiteSeer, PubMed, Amaon-Computers, Computers-Random and CS-Random. For Flickr, $K$ is selected among $\{1000, 2000, 3000, 5000\}$. For Cora, CiteSeer, PubMed, and Amazon-Computers, the number of training round $T$ are tuned among $\{40, 60, 80, 100\}$. For Computers-Random and CS-Random, $T$ are selected from $\{4, 8, 12, 16, 20, 24\}$. For Flickr, $T$ is tuned among $\{40, 50, 60, 70\}$. We also introduce a hyperparameter $\alpha$, which is the upper bound on the number of nodes being added per class per round. The tuning range of $\alpha$ is $\{4, 6, 8, 10\}$ for Cora, CiteSeer, Amazon-Computers and $\{64, 72, 80\}$ for PubMed. For Computers-Random and CS-Random, the value of $\alpha$ is chosen among $\{2, 4, 6, 8, 10, 12, 14, 16\}$. For Flickr, the value of $\alpha$ is selected among $\{30, 40, 50, 60\}$. The weight parameters $p$ in RBO is selected among $\{0.5, 0.75, 0.98\}$, and the threshold in DGIN is tuned among $\{0.25, 0.5, 0.75, 1.00\}$. For Flickr, we only add minority nodes to the training set in all iterations, which means that we set $\alpha = 0$ for majority classes in Flickr.

### K.5 Baselines

For GraphSMOTE [73], we use the branched algorithms whose edge predictions are discrete-valued, which have achieved superior performance over other variants in most experiments. For the ReNode method [6], we search hyperparameters among lower bound of cosine annealing $w_{min} \in \{0.25, 0.5, 0.75\}$ and upper bound of the cosine annealing $w_{max} \in \{1.25, 1.5, 1.75\}$ following [6]. PageRank teleport probability is fixed as $\alpha = 0.15$, which is the default setting in the released codes. For TAM [47], we search the best hyperparameters among the coefficient of ACM term $\alpha \in \{1.25, 1.5, 1.75\}$, the coefficient of ADM term $\beta \in \{0.125, 0.25, 0.5\}$, and the minimum temperature of class-wise temperature $\phi \in \{0.8, 1.2\}$ following [47]. The sensitivity to imbalance ratio of class-wise temperature $\delta$ is fixed as 0.4 for all main experiments. Following [47], we adopt a warmup for 5 iterations since we utilize model prediction for unlabeled nodes. For BIM [71] and GraphSR [75], we follow the hyperparameter settings and network architectures provided in their official code repositories.

### K.6 Configuration

All the algorithms and models are implemented in Python and PyTorch Geometric. Experiments are conducted on a server with an NVIDIA 3090 GPU (24 GB memory) and an Intel(R) Xeon(R) Silver 4210R CPU @ 2.40GHz.

Table 36: Label distributions in the training sets.

| Dataset | $\mathcal{C}_0$ | $\mathcal{C}_1$ | $\mathcal{C}_2$ | $\mathcal{C}_3$ | $\mathcal{C}_4$ | $\mathcal{C}_5$ | $\mathcal{C}_6$ | $\mathcal{C}_7$ | $\mathcal{C}_8$ | $\mathcal{C}_9$ | $\mathcal{C}_{10}$ | $\mathcal{C}_{11}$ | $\mathcal{C}_{12}$ | $\mathcal{C}_{13}$ | $\mathcal{C}_{14}$ |
|---|---|---|---|---|---|---|---|---|---|---|---|---|---|---|---|
| Cora ($\rho = 10$) | 23.26% | 23.26% | 23.26% | 23.26% | 2.32% | 2.32% | 2.32% | - | - | - | - | - | - | - | - |
| Cora ($\rho = 20$) | 24.10% | 24.10% | 24.10% | 24.10% | 1.19% | 1.19% | 1.19% | - | - | - | - | - | - | - | - |
| Cora ($\rho = 50$) | 24.63% | 24.63% | 24.63% | 24.63% | 0.49% | 0.49% | 0.49% | - | - | - | - | - | - | - | - |
| Cora ($\rho = 100$) | 24.81% | 24.81% | 24.81% | 24.81% | 0.25% | 0.25% | 0.25% | - | - | - | - | - | - | - | - |
| CiteSeer ($\rho = 10$) | 30.30% | 30.30% | 30.30% | 3.03% | 3.03% | 3.03% | - | - | - | - | - | - | - | - | - |
| CiteSeer ($\rho = 20$) | 31.75% | 31.75% | 31.75% | 1.59% | 1.59% | 1.59% | - | - | - | - | - | - | - | - | - |
| CiteSeer ($\rho = 50$) | 32.68% | 32.68% | 32.68% | 0.65% | 0.65% | 0.65% | - | - | - | - | - | - | - | - | - |
| CiteSeer ($\rho = 100$) | 33.00% | 33.00% | 33.00% | 0.33% | 0.33% | 0.33% | - | - | - | - | - | - | - | - | - |
| PubMed ($\rho = 10$) | 47.62% | 47.62% | 4.76% | - | - | - | - | - | - | - | - | - | - | - | - |
| PubMed ($\rho = 20$) | 48.78% | 48.78% | 2.44% | - | - | - | - | - | - | - | - | - | - | - | - |
| PubMed ($\rho = 50$) | 49.50% | 49.50% | 0.99% | - | - | - | - | - | - | - | - | - | - | - | - |
| PubMed ($\rho = 100$) | 49.75% | 49.75% | 0.50% | - | - | - | - | - | - | - | - | - | - | - | - |
| Amazon-Computers ($\rho = 10$) | 18.18% | 18.18% | 18.18% | 18.18% | 18.18% | 1.82% | 1.82% | 1.82% | 1.82% | 1.82% | - | - | - | - | - |
| Amazon-Computers ($\rho = 20$) | 19.05% | 19.05% | 19.05% | 19.05% | 19.05% | 0.95% | 0.95% | 0.95% | 0.95% | 0.95% | - | - | - | - | - |
| Amazon-Computers ($\rho = 50$) | 19.61% | 19.61% | 19.61% | 19.61% | 19.61% | 0.39% | 0.39% | 0.39% | 0.39% | 0.39% | - | - | - | - | - |
| Amazon-Computers ($\rho = 100$) | 19.80% | 19.80% | 19.80% | 19.80% | 19.80% | 0.20% | 0.20% | 0.20% | 0.20% | 0.20% | - | - | - | - | - |
| Computers-Random ($\rho = 25.50$) | 3.01% | 15.79% | 10.53% | 3.76% | 38.35% | 2.26% | 3.01% | 6.02% | 15.79% | 1.50% | - | - | - | - | - |
| CS-Random ($\rho = 41.00$) | 3.98% | 2.27% | 11.36% | 2.27% | 7.39% | 11.93% | 1.70% | 5.11% | 3.98% | 0.57% | 7.95% | 11.36% | 2.27% | 23.30% | 4.55% |
| Flickr ($\rho \approx 10.80$) | 5.89% | 9.68% | 7.09% | 5.45% | 25.83% | 3.90% | 42.16% | - | - | - | - | - | - | - | - |

# L  Pseudocode of Our Algorithm

---

**Algorithm 1** Our Algorithm

---

**Input:** Graph $\mathcal{G} = (\mathcal{V}, \mathcal{E})$, labeled set $\mathcal{L}_0$, feature matrix $\mathcal{X}$, adjacency matrix $\mathcal{A}$, unlabeled set $\mathcal{U} = \mathcal{V} \setminus \mathcal{L}_0$, max rounds $T$, per-class selection threshold $\alpha$, RBO weight parameter $p$, GI threshold $\gamma$, cluster size $k'$, learning rate $\eta$, GNN model $f_\text{g}$, clustering function $f_\text{cluster}$, mean function $M(\cdot)$.

1: **for** $i = 1$ to $T$ **do**
2:    Train GNN $f_\text{g}$ on current labeled set $\mathcal{L}_{i-1}$.
3:    Obtain embeddings $H^L$ and $H^U$ for labeled and unlabeled nodes.
4:    Predict class logits $\hat{y}$ and confidence scores $r$ for $\mathcal{U}$.
5:    **// Step 1: Dual Pseudo-label Alignment (DPAM)**
6:    Cluster $H^U$ into $k'$ clusters: $f_\text{cluster}(H^U) \rightarrow \{\mathcal{K}_j, \mu_{\mathcal{K}_j}\}_{j=1}^{k'}$.
7:    Compute class centroids from labeled data: $\mu_{\mathcal{C}_m} = M(\{h_v^L \mid y_v = m\})$.
8:    Assign pseudo-labels to clusters: $\tilde{y}_j = \arg\min_m \text{distance}(\mu_{\mathcal{K}_j}, \mu_{\mathcal{C}_m})$.
9:    Construct cluster-based pseudo-label sets $\tilde{\mathcal{U}}_m$ and classifier-based sets $\mathcal{U}_m$.
10:    Obtain consistent node sets: $\mathcal{U}_m^{\text{final}} = \tilde{\mathcal{U}}_m \cap \mathcal{U}_m$.
11:    **// Step 2: Node-Reordering (NR)**
12:    **for** each class $m = 1$ to $C$ **do**
13:        Compute geometric distances: $\delta_u = \text{distance}(h_u, \mu_{\mathcal{C}_m})$ for $u \in \mathcal{U}_m^{\text{final}}$.
14:        Build geometric ranking $\mathcal{S}_m$ (ascending by $\delta_u$) and confidence ranking $\mathcal{T}_m$ (descending by $r_u$).
15:        Compute RBO score $r_m = \text{RBO}(\mathcal{S}_m, \mathcal{T}_m)$.
16:        Fuse rankings: $\mathcal{N}_m^{\text{New}} = \max\{r_m, 1 - r_m\} \cdot \mathcal{S}_m + \min\{r_m, 1 - r_m\} \cdot \mathcal{T}_m$.
17:        Select top-$\alpha$ nodes from $\mathcal{N}_m^{\text{New}}$ as candidates $\mathcal{C}_m^{\text{cand}}$.
18:    **end for**
19:    **// Step 3: Discarding Geometrically Imbalanced Nodes (DGIS)**
20:    **for** each selected node $u \in \bigcup_m \mathcal{C}_m^{\text{cand}}$ **do**
21:        Let $\delta_u$ be distance to closest centroid, $\beta_u$ be distance to second closest.
22:        Compute GI index: $\text{GI}_u = \frac{\beta_u - \delta_u}{\delta_u}$.
23:        **if** $\text{GI}_u < \gamma$ **then**
24:            Discard node $u$.
25:        **else**
26:            Add node $u$ to labeled set $\mathcal{L}_i$ with its pseudo-label.
27:        **end if**
28:    **end for**
29: **end for**

---

