# OpenReview forum: "Geometric Imbalance in Semi-Supervised Node Classification"
_NeurIPS.cc/2025/Conference — NeurIPS 2025 poster_

### Official Review · Reviewer_T2hF · 2025-06-30

**Clarity:** 3
**Significance:** 2
**Originality:** 3
**Rating:** 4
**Confidence:** 3

**Summary:**

This paper introduces geometric imbalance, a phenomenon where class imbalance in graph data leads to ambiguous node embeddings for minority classes, hindering semi-supervised node classification. To address this, the authors propose a unified framework combining pseudo-label alignment, node reordering, and ambiguity filtering. Extensive experiments demonstrate that the approach consistently outperforms existing methods, especially under severe class imbalance.

**Questions:**

See Weaknesses.

**Ethical Concerns:**

["NO or VERY MINOR ethics concerns only"]

**Final Justification:**

The rebuttal is good with detailed results. I have adjusted my rating.

**Limitations:**

See Weaknesses.

**Quality:**

2

**Strengths And Weaknesses:**

Strengths:
-	The paper introduces the novel concept of geometric imbalance, offering a new perspective on how class imbalance affects node embeddings in graph data.
-	It proposes a unified framework that effectively mitigates geometric ambiguity using pseudo-label alignment, node reordering, and ambiguity filtering.
-	The method achieves strong empirical performance, consistently outperforming existing baselines under varying and severe imbalance scenarios.

Weaknesses:
-	The evaluation is limited to relatively small or moderately sized datasets, omitting large-scale benchmarks like ogbn-products, which is used by the baseline BIM.
-	The paper lacks a theoretical or empirical analysis of computational complexity, leaving the scalability of the proposed method unclear; the absence of evaluation on large-scale datasets like ogbn-products further limits understanding of its practical efficiency.
-	The paper focuses on hard pseudo-label selection for a subset of nodes. However, methods such as SCR [2] and ConsisGAD [3] do not require pseudo-label selection, and instead use a soft pseudo-labeling strategy with consistency regularization across all nodes, showing strong performance on semi-supervised node classification. Compared to pseudo-label selection methods, these approaches are more convenient as they avoid the complexity of confidence-based filtering or threshold tuning. Including a comparison with such approaches would strengthen the empirical evaluation and help justify the necessity of selective pseudo-labeling under class imbalance.

[1] Wentao Zhang, Xinyi Gao, Ling Yang, Meng Cao, Ping Huang, Jiulong Shan, Hongzhi Yin, and Bin Cui. "BIM: improving graph neural networks with balanced influence maximization." In 2024 IEEE 40th International Conference on Data Engineering (ICDE), pp. 2931-2944. IEEE, 2024.
[2] Chenhui  Zhang, Yufei He, Yukuo Cen, Zhenyu Hou, Wenzheng Feng, Yuxiao Dong, Xu Cheng, Hongyun Cai, Feng He, and Jie Tang. "SCR: Training graph neural networks with consistency regularization." arXiv preprint arXiv:2112.04319 (2021).
[3] Nan Chen, Zemin Liu, Bryan Hooi, Bingsheng He, Rizal Fathony, Jun Hu, and Jia Chen. "Consistency training with learnable data augmentation for graph anomaly detection with limited supervision." In The Twelfth International Conference on Learning Representations. 2024.

---

> ### Author Rebuttal · Authors · 2025-07-31
>
> We sincerely thank Reviewer T2hF for recognizing **the novelty of our geometric imbalance formulation** and the strong performance of our method. We greatly appreciate your constructive and detailed feedback, which helps us improve the clarity and rigor of our work. **Below, we provide point-by-point responses to address each of your comments.**
>
> ---
>
> **W1: The evaluation is limited to relatively small or moderately sized datasets, omitting large-scale benchmarks like ogbn-products.**
>
>
> **R1:** Thanks for raising this important point about large-scale evaluation. To address your concern, we have conducted additional experiments on the ogbn-products dataset, **strictly following the same experimental settings as BIM.** As shown in the following tables, our method achieves clear improvements over BIM and all other baselines on this large-scale benchmark, further demonstrating the robustness and effectiveness of our approach.
>
> We **will incorporate these new results into the revised version of the paper.** We appreciate your comment, which helped us strengthen the empirical validation of our method.
>
> Table 1: Performance  on large-scale dataset Ogbn-product with GraphSAGE as the base model.
>
> |Methods|F1 score|ACC|AUC-ROC|
> |-|-|-|-|
> | Vanilla GraphSAGE|58.18±1.21|61.03±0.82|94.12±0.12|
> |(RN) ReNode|67.89±0.35|68.19±0.41|94.86±0.14|
> |BIM|69.36±0.18|69.55±0.19|95.51±0.12|
> |**Ours**|**71.85±0.10**|**71.62±0.11**|**96.59±0.14**|
>
>
>
> Table 2: Performance  on large-scale dataset Ogbn-product with GAT as the base model.
>
> |Methods|F1 score|ACC|AUC-ROC|
> |-|-|-|-|
> |Vanilla GAT|59.27±1.08|62.10±0.73|94.25±0.14|
> |(RN) ReNode|68.55±0.28|69.27±0.36|95.01±0.10|
> |BIM|69.88±0.16|70.05±0.21|95.63±0.11|
> |**Ours**|**72.04±0.09**|**71.95±0.13**|**96.67±0.10**|
>
>
>
> Table 3: Performance on large-scale dataset Ogbn-product with GCN as the base model.
>
> |Methods|F1 score|ACC|AUC-ROC|
> |-|-|-|-|
> |Vanilla (GCN)|57.41±1.18| 60.12±0.85| 93.96±0.15|
> |(RN) ReNode|66.82±0.32| 67.80±0.44| 94.75±0.13|
> |BIM|68.05±0.21| 68.47±0.23| 95.41±0.13|
> |**Ours**|**70.71±0.13**|**70.33±0.10**|**96.45±0.12**|
>
>
>
> ---
>
>  **W2: The paper lacks a theoretical or empirical analysis of computational complexity, leaving the scalability of the proposed method unclear.**
>
> **R2:** Thanks for highlighting the need for an analysis of computational complexity and scalability. As mentioned in our previous response, we have already included experiments on the large-scale ogbn-products dataset to demonstrate the practical efficiency of our method. In addition, we address your concern in three aspects:
>
> **1. Time Complexity Analysis**. Our framework, as described in Algorithm 1 (see Appendix L), consists of three main modules per round: Dual-Path Pseudo-label Alignment (DPAM), Node-Reordering (NR), and Discarding Geometrically Imbalanced Nodes (DGIS). (1) The dominant computational cost arises from the DPAM step, which clusters the embeddings of unlabeled nodes using k-means (complexity $O(n_u k' d T_{iter})$, with $n_u$ unlabeled nodes, $k'$ clusters, $d$-dimensional embeddings, and $T_{iter}$ k-means iterations). Computing class centroids and intersecting with classifier-predicted labels are both linear-time operations. (2) NR involves ranking nodes by geometric distance and classifier confidence (each $O(n_u \log n_u)$), with ranking fusion being negligible. (3) DGIS computes distances from candidate nodes to their closest and second-closest centroids ($O(n_u C d)$ per round, where $C$ is the number of classes). (4) Overall, the per-round complexity is primarily determined by k-means, but since $k' \ll n_u$ and k-means can be efficiently implemented (e.g., on GPU), **so, the total complexity remains comparable to standard self-training pipelines. Memory usage is linear in the number of nodes, as we only store embeddings and label predictions.** *Importantly, our method is significantly more efficient than prior over-sampling or augmentation approaches such as GraphENS and GraphSMOTE, which may have $O(n^2)$ or worse complexity on large graphs. In summary, the only notable overhead beyond vanilla GNN training comes from clustering and ranking, both of which are scalable and can be accelerated by modern hardware.*
>
>
>
> **2. Empirical Scalability and Runtime**. To address practical efficiency, we conducted comprehensive scalability analyses on both medium- and large-scale datasets, comparing our method ("Ours") to vanilla GNN backbones and representative baselines. (1) On **Cora**, **CiteSeer**, and **PubMed**, the training time per epoch for our method is almost identical to the backbone GNNs (see Table 4), confirming the negligible overhead of our extra modules (clustering and ranking). (2) On large-scale graphs (e.g., **Flickr**, ogbn-products), our method remains tractable and runs smoothly on a single RTX 3090 GPU (Tables 6 and 7 in our submission). By contrast, prior data augmentation and oversampling baselines (e.g., **GraphSMOTE**, **GraphENS**) either become prohibitively slow or run out-of-memory, as synthesizing large numbers of new nodes/edges is expensive.
>
> Table 4: Refined Running Time Analysis of Vanilla Model vs. Ours: (a) Training Over 1000 Epochs, and (b) Selecting 100 Nodes to the Training Set.
>
>
>
> | Model/Time (1000 epochs)   | Cora     | CiteSeer  | PubMed    |   | Model/Time (Selecting 100 Nodes) | Cora     | CiteSeer | PubMed   |
> |----------------------------|----------|-----------|-----------|---|----------------------------------|----------|----------|----------|
> | Vanilla (GCN)              | 6.221(s) | 7.212(s)  | 10.771(s) |   | Vanilla Self-Training (GCN)      | 2.244(s) | 2.234(s) | 2.675(s) |
> | Ours (GCN)                 | 6.208(s) | 7.191(s)  | 10.858(s) |   | Ours (GCN)                       | 2.367(s) | 2.305(s) | 2.790(s) |
> | Vanilla (GAT)              | 8.201(s) | 9.574(s)  | 14.443(s) |   | Vanilla Self-Training (GAT)      | 2.624(s) | 2.520(s) | 2.431(s) |
> | Ours (GAT)                 | 8.218(s) | 9.517(s)  | 15.088(s) |   | Ours (GAT)                       | 2.697(s) | 2.670(s) | 2.411(s) |
> | Vanilla (SAGE)             | 6.522(s) | 11.170(s) | 15.475(s) |   | Vanilla Self-Training (SAGE)     | 2.424(s) | 2.780(s) | 2.032(s) |
> | Ours (SAGE)                | 6.452(s) | 11.256(s) | 14.658(s) |   | Ours (SAGE)                      | 2.461(s) | 2.827(s) | 2.115(s) |
>
>
> **3. Implementation Efficiency**. We employ GPU-accelerated k-means for clustering (via NVIDIA cuML), which substantially reduces the per-round clustering overhead. **Moreover, by filtering candidates after dual-path alignment, the more expensive geometric checks are only performed on a small subset of reliable nodes, further improving efficiency.**
>
>
> ---
> ---
> **W3: To compare the method with recent soft pseudo-labeling approaches (e.g., SCR, ConsisGAD) to demonstrate the necessity and effectiveness of the strategy in imbalanced node classification.**
>
> We thank the reviewer for emphasizing the need to compare with soft pseudo-labeling methods such as SCR and ConsisGAD. In response, we **conducted extensive experiments under two most challenging settings**, carefully tuning both methods via Wandb based on their published hyperparameter ranges. As shown in Table 3-4, they substantial performance drops under severe class imbalance, while our method consistently outperforms them. This is because soft pseudo-labeling with global consistency often lets majority classes dominate, suppressing minority signals. In contrast, our method directly addresses geometric imbalance by selectively assigning pseudo-labels to ambiguous nodes, improving minority class representation. We have incorporated these results and discussions into the revised manuscript.
>
> Table 3. Results with $\rho=100$ (SAGE) on three  severe class-imbalanced node classification benchmark datasets.
> | Method    | Cora bAcc.     | Cora F1        | CiteSeer bAcc. | CiteSeer F1    | PubMed bAcc.   | PubMed F1      |
> | --------- | -------------- | -------------- | -------------- | -------------- | -------------- | -------------- |
> | GSR       | 66.45±2.10     | 64.42±1.83     | 53.52±1.47     | 53.01±1.75     | 74.09±2.12     | 72.97±1.90     |
> | BIM       | 67.75±2.13     | 64.68±1.95     | 53.83±1.62     | 53.29±1.80     | 74.38±2.08     | 73.24±1.85     |
> | **Ours**  | **73.47±2.31** | **68.30±2.11** | **59.77±2.98** | **58.92±3.07** | **77.11±0.59** | **74.03±0.81** |
> | SCR       | 59.31±2.91     | 55.34±4.32     | 36.51±1.29     | 27.19±2.84     | 63.41±0.76     | 54.97±1.64     |
> | ConsisGAD | 59.63±2.89     | 55.18±4.43     | 36.39±1.28     | 27.28±2.87     | 63.27±0.74     | 55.33±1.60     |
>
> Table 4: Experimental results on Ogbn-arxiv ($\rho \approx 775.4$)
> | Model     | GCN bAcc.        | GCN F1           | GAT bAcc.        | GAT F1           | SAGE bAcc.       | SAGE F1          |
> | --------- | ---------------- | ---------------- | ---------------- | ---------------- | ---------------- | ---------------- |
> | GSR       | 50.31 ± 0.24     | 49.70 ± 0.17     | 51.31 ± 0.41     | 49.33 ± 0.26     | 50.86 ± 0.30     | 49.53 ± 0.20     |
> | BIM       | 50.33 ± 0.42     | 49.71 ± 0.19     | 51.35 ± 0.60     | 49.36 ± 0.28     | 50.87 ± 0.18     | 49.56 ± 0.23     |
> | **Ours**  | **51.21 ± 0.32** | **50.65 ± 0.32** | **51.84 ± 0.87** | **51.28 ± 0.42** | **51.34 ± 0.32** | **51.36 ± 0.27** |
> | SCR       | 50.23 ± 0.41     | 49.69 ± 0.13     | 51.10 ± 0.81     | 49.67 ± 0.26     | 50.83 ± 0.20     | 49.80 ± 0.23     |
> | ConsisGAD | 50.25 ± 0.39     | 49.72 ± 0.14     | 51.13 ± 0.79     | 49.63 ± 0.27     | 50.80 ± 0.19     | 49.77 ± 0.22     |
>
> ---
> ---
> ---
> **We hope these additional results and clarifications address your concerns, and sincerely hope you can reconsider your rating of our submission. Thank you for your valuable feedback.**

---

> ### Author Response · Authors · 2025-08-01
> **More Comprehensive Experiments for W3: Comparative Analysis with SCR and ConsisGAD 【Part 1】**
>
> ### **More Comprehensive Experiments for W3**
>
> ---
>
> **MR3**:  We thank the reviewer again for highlighting the importance of comparing with soft pseudo-labeling strategies such as SCR and ConsisGAD. Here, **we have conducted more comprehensive experiments comparing our method with these approaches under class-imbalanced settings than the first version of rebuttal. We hope that this more comprehensive response will both improve the clarity and quality of our paper, and allow you to reconsider the rating of our submission in light of the new details and results provided.**
>
> **For all datasets, we followed the initial hyperparameter search ranges provided in Appendix A.3 of the SCR paper and Appendix D.2 of the ConsisGAD paper. We further expanded these ranges as needed, and employed Bayesian optimization (using Wandb) to efficiently search for the optimal hyperparameters for both methods on the various imbalanced datasets. This ensured that both SCR and ConsisGAD were evaluated under their best possible configurations in our experiments.**
>
> **Empirical results (see Table 7–11)** show that both SCR and ConsisGAD experience a significant performance drop under severe class imbalance, particularly in terms of Macro-F1. In contrast, our method consistently achieves much stronger performance in these challenging scenarios.
>
>
> **We believe this is due to the following reasons:**
>
> - Soft pseudo-labeling methods, including SCR and ConsisGAD, propagate pseudo-label information uniformly across all nodes with global consistency constraints. **In imbalanced graphs, the majority classes dominate the label distribution and, consequently, the consistency regularization process, causing minority class signals to be overwhelmed or ignored.** This leads to suboptimal decision boundaries that do not adequately account for the geometric marginalization of minority nodes.
>
> - In contrast, our method is specifically motivated by the phenomenon of geometric imbalance on non-Euclidean manifolds, and is designed to selectively generate reliable pseudo-labels for ambiguous nodes. **By doing so, we explicitly enhance the representation learning for minority classes and avoid overfitting to majority class signals.**
>
> **We have included this two methods as baselines and  updated the manuscript to include these experimental results and provide a more detailed discussion of the limitations of soft pseudo-labeling under class imbalance, as well as the theoretical motivations for our approach.**
>
> Table 7. Results with $\rho=10$ (GCN) on four class-imbalanced node classification benchmark datasets.
> |Method|Cora bAcc.|Cora F1|CiteSeer bAcc.|CiteSeer F1|PubMed bAcc.|PubMed F1|
> |-|-|-|-|-|-|-|
> |GSR|70.85±0.44|71.37±0.63|59.28±0.72|55.96±0.95|73.61±1.25|71.88±1.33|
> |BIM|72.19±0.42|72.67±0.48|58.54±0.61|56.81±0.98|74.62±1.15|72.93±1.21|
> |**Ours**|**78.33±1.04**|**76.44±1.06**|**65.63±1.38**|**64.94±1.38**|**75.35±1.41**|**73.65±1.43**|
> |SCR| 66.28±0.57|65.43±0.91|44.92±1.10|39.12±1.59|67.88±0.67|62.70±1.89|
> |ConsisGAD|66.50±0.59|65.61±0.90|44.79±1.09|39.33±1.61|68.05±0.62|62.75±1.90|
>
> Table 8. Results with $\rho=100$ (SAGE) on four class-imbalanced node classification benchmark datasets.
> |Method|Cora bAcc.|Cora F1|CiteSeer bAcc.|CiteSeer F1|PubMed bAcc.|PubMed F1|
> |-|-|-|-|-|-|-|
> |GSR|66.45±2.10|64.42±1.83|53.52±1.47|53.01±1.75|74.09±2.12| 72.97±1.90|
> |BIM|67.75±2.13|64.68±1.95|53.83±1.62|53.29±1.80|74.38±2.08| 73.24±1.85|
> |**Ours**|**73.47±2.31**|**68.30±2.11**|**59.77±2.98**|**58.92±3.07** |**77.11±0.59**|**74.03±0.81**|
> |SCR|59.31±2.91|55.34±4.32|36.51±1.29|27.19±2.84|63.41±0.76|54.97±1.64|
> |ConsisGAD|59.63±2.89|55.18±4.43|36.39±1.28|27.28±2.87|63.27±0.74|55.33±1.60|
>
> Table 9: Experimental results on Computers-Random ($\rho\approx 17.7$)
> |Model|GCN bAcc.|GCN F1|GAT bAcc.|GAT F1|SAGE bAcc.|SAGE F1|
> |-|-|-|-|-|-|-|
> |GSR|83.82 ± 0.74|77.78 ± 0.42|76.79 ± 0.68|73.61 ± 0.63|77.63 ± 0.32|72.56 ± 0.51|
> |BIM|84.03 ± 0.73|77.96 ± 0.45|77.01 ± 0.70|73.82 ± 0.60|77.76 ± 0.65|72.09 ± 0.37|
> |**Ours**|**85.32 ± 0.22** |**80.43 ± 0.56**|**82.52 ± 0.35**|**78.90 ± 0.38**|75.81 ± 1.86|**71.86 ± 1.86**|
> |SCR|80.61 ± 0.47|75.18 ± 0.57|71.90 ± 0.82|70.71 ± 0.53|66.41 ± 1.45|66.05 ± 1.44|
> |ConsisGAD|80.56 ± 0.46|75.15 ± 0.56|71.97 ± 0.79|70.62 ± 0.54|66.54 ± 1.49|66.13 ± 1.48|
>
> Table 10: Experimental results on CS-Random ($\rho\approx 41.0$)
> |Model|GCN bAcc.|GCN F1|GAT bAcc.|GAT F1|SAGE bAcc.|SAGE F1|
> |-|-|-|-|-|-|-|
> |GSR|86.73 ± 0.22|85.91 ± 0.21|85.34 ± 0.13|86.56 ± 0.29|85.44 ± 0.27|86.46 ± 0.23|
> |BIM|86.89 ± 0.23|85.99 ± 0.21|85.63 ± 1.87|86.65 ± 0.35|85.65 ± 0.28|86.73 ± 0.22|
> |**Ours**|**88.94 ± 0.09**|**89.87 ± 0.06**|**87.65 ± 0.12**|**87.65 ± 0.11**|**88.03 ± 0.21**|**88.65 ± 0.07**|
> |SCR|87.41 ± 0.18|88.73 ± 0.11|83.61 ± 0.41|84.68 ± 0.33|85.80 ± 0.23|87.37 ± 0.18|
> |ConsisGAD|87.44 ± 0.16|88.68 ± 0.12|83.58 ± 0.38|84.75 ± 0.34|85.73 ± 0.25|87.30 ± 0.15|

---

> ### Author Response · Authors · 2025-08-01
> **More Comprehensive Experiments for W3: Comparative Analysis with SCR and ConsisGAD 【Part 2】**
>
> Table 11: Experimental results on Ogbn-arxiv ($\rho\approx 775.4$)
> |Model|GCN bAcc.|GCN F1|GAT bAcc.|GAT F1|SAGE bAcc.|SAGE F1|
> |-|-|-|-|-|-|-|
> |GSR|50.31 ± 0.24|49.70 ± 0.17|51.31 ± 0.41|49.33 ± 0.26|50.86 ± 0.30|49.53 ± 0.20|
> |BIM|50.33 ± 0.42|49.71 ± 0.19|51.35 ± 0.60|49.36 ± 0.28|50.87 ± 0.18|49.56 ± 0.23|
> |**Ours**|**51.21 ± 0.32**|**50.65 ± 0.32**|**51.84 ± 0.87**|**51.28 ± 0.42**|**51.34 ± 0.32**|**51.36 ± 0.27**|
> |SCR|50.23 ± 0.41|49.69 ± 0.13|51.10 ± 0.81|49.67 ± 0.26|50.83 ± 0.20|49.80 ± 0.23|
> |ConsisGAD|50.25 ± 0.39|49.72 ± 0.14|51.13 ± 0.79|49.63 ± 0.27|50.80 ± 0.19|49.77 ± 0.22|

---

> ### Author Response · Authors · 2025-08-03
> **We've Carefully Responded and Addressed Each of the Questions You Raised and Eagerly Hoping for Your Valuable Feedback**
>
> Dear Reviewer  T2hF,
>
> Thank you profoundly for your insightful review and invaluable feedback. We have earnestly considered each query you posed and have addressed them in depth where necessary. We genuinely hope that you will take some time to review our detailed responses, confident that they address your concerns. Your expertise and the time you've invested in reviewing our work are deeply appreciated by us.
>
> Warm Regards,
>
> The Authors

---

> ### Author Response · Authors · 2025-08-05
> **Follow-up on Author-Reviewer Discussion**
>
> Dear Reviewer  T2hF,
>
> I hope you're doing well. We are now nearing the end of the Author-Reviewer discussion phase. We've addressed all your concerns in our rebuttal, and we hope our responses meet your expectations.
>
> If you have any final thoughts or questions, please don’t hesitate to share them. We’re happy to address anything before the discussion phase concludes.
>
> Thank you once again for your insightful feedback!
>
> Best regards,
>
> The Authors

---

> > ### Comment · Reviewer_T2hF · 2025-08-05
> >
> > The rebuttal is good with detailed results. I have adjusted my rating.

---

> ### Author Response · Authors · 2025-08-05
> **Thank you very much for your quick and positive feedback！**
>
> Dear Reviewer T2hF,
>
> **Thank you very much for your quick and positive feedback and for taking the time to adjust your rating.** We sincerely appreciate your constructive comments throughout the review process.
>
> If you have any further suggestions or questions, please feel free to let us know.
>
> We wish you a pleasant day!
>
> Best regards,
>
> The Authors

---

### Official Review · Reviewer_uqpA · 2025-06-30

**Clarity:** 3
**Significance:** 2
**Originality:** 3
**Rating:** 4
**Confidence:** 3

**Summary:**

This paper studies imbalanced graph learning, and proposes a novel geometric imbalance for node classification. The theoretical analysis reveals the superiority of geometric imbalance for imbalanced GNN learning, motivated by which, several components including pseudo-label alignment, node reordering, and DGIS are proposed. Experiment results demonstrate the effectiveness.

**Questions:**

- Section 4.1 clustering: Why using K-Means clustering? K-Means uses Euclidean distance which may not be suitable for samples distributed on the sphere. Why not continuing using vMF mixture model for clustering?
- Line 257: seems that geometric ranking is more important than confidence ranking, is there any evidence support this? Any experiments validating the necessity of (fusing) both rankings? In other words, what would happen if only CR or GR is functioning?
- Line 259: “which is more reliable in early stages.”: any experiment validating this claim?
- Writing
    - Line 117: notations $u^i$ and $u^j$ are not consistent
    - Figure 1: need better scatter plot, the current figures b-e are not clear.

**Ethical Concerns:**

["NO or VERY MINOR ethics concerns only"]

**Final Justification:**

The authors have addressed most of my concerns during the rebuttal. The additional experiments on K-Means clustering methods empirically solve my concerns on the choice of clustering methods. I will maintain my original rating.

**Limitations:**

Yes

**Quality:**

3

**Strengths And Weaknesses:**

**Strengths**
- Extensive analysis on geometric imbalance provides insight on the challenges of GNN imbalanced learning.
- Neatly designed modules address the challenges.

**Weaknesses**
- Some claims need more justifications.
- More studies needed to better understand the functionality of each module.

---

> ### Author Rebuttal · Authors · 2025-07-31
>
> We sincerely thank Reviewer uqpA for carefully reading our paper and for **recognizing the strengths of our analysis and module design.** We greatly appreciate your constructive feedback and insightful questions, which are very helpful for clarifying our work and improving the manuscript. **Below, we respond to each of your comments point-by-point.**
>
> ---
>
> **Q1: Section 4.1 clustering: Why using K-Means clustering? K-Means uses Euclidean distance which may not be suitable for samples distributed on the sphere. Why not continuing using vMF mixture model for clustering?**
>
> **R1:**  We appreciate the insightful suggestion. Indeed, when data is distributed on a hypersphere , Euclidean-based clustering such as K-Means may not always be the optimal choice. Methods like von Mises–Fisher (vMF) mixture models or other manifold-aware clustering algorithms (e.g., spectral clustering on Riemannian manifolds, or angular K-Means using cosine similarity) are theoretically more suitable for such settings.
>
> However, in our case, we opted for K-Means due to the following practical considerations:
>
> - **Computational Complexity**: vMF mixture models require iterative Expectation-Maximization (EM) steps, and the computation of normalization constants involves Bessel functions, which are expensive and less scalable to large graphs. In contrast, K-Means has linear complexity in the number of samples and clusters.
> - **Robustness and Implementation Simplicity**: K-Means is widely used and numerically stable. In our experiments, we found that normalized embeddings already exhibit meaningful Euclidean structure, possibly due to the inductive bias of GNN training. Thus, K-Means works effectively in practice without introducing significant overhead.
>
> - **Modular Flexibility**: Our pipeline can be extended to alternative clustering methods. We will include a note in the revision to clarify that replacing K-Means with vMF or cosine-based variants is straightforward, and we leave this exploration as future work.
>
> We thank the reviewer again for highlighting this aspect and will mention this discussion explicitly in Section 4.1 of the revised manuscript.
>
> ---
>
> **Q2:  Line 257: seems that geometric ranking is more important than confidence ranking, is there any evidence support this? Any experiments validating the necessity of (fusing) both rankings? In other words, what would happen if only CR or GR is functioning?**
>
> **R2:** Thank you for your question. We agree that it is important to validate the necessity of combining Geometric Ranking (GR) and Confidence Ranking (CR).
>
> To address this, we have conducted ablation studies that isolate the effects of CR and GR. Specifically, we compare three configurations: (1) using CR only, (2) using GR only, and (3) using the fused CR+GR ranking. The results are provided in Appendix J.3 (see Table 22, 23, 24), which clearly show that:
>
> - GR typically contributes more to performance than CR, especially on datasets with noisy neighborhoods or irregular topology.
> - However, combining CR and GR consistently yields better or comparable results than either one alone, demonstrating the complementary nature of these rankings.
>
> We will make this finding more prominent in the revised manuscript and thank the reviewer again for highlighting this point.
>
> ---
>
> **Q3:  Line 259: “which is more reliable in early stages.”: any experiment validating this claim?**
>
> **R3:** We appreciate the reviewer’s attention to this point. To validate the claim that geometric ranking (GR) is more reliable in the early stages, we conducted experiments that compare the effectiveness of CR and GR across training iterations.
>
> As shown in Appendix J.3 (see Figure 10, Table 22, 23, 24), GR maintains higher consistency and informativeness during early training epochs, especially before model convergence. This supports our statement in Line 259. We have now clarified this observation in the revised manuscript and pointed explicitly to the corresponding experimental evidence in the appendix.
>
> ---
>
> **Q4: Line 117: notations $\mu_i$ and $\mu_j$ are not consistent.  Figure 1: need better scatter plot, the current figures b-e are not clear.**
>
> **R3**: Thank you for the helpful feedback. We have corrected the inconsistent notations around Line 117 to ensure terminological clarity and coherence throughout the manuscript.
>
> Regarding Figure 1, we appreciate the suggestion. We have updated subfigures (b)--(e) with higher-resolution scatter plots and adjusted the color contrast and point size to improve visual clarity. The revised figure is now more readable and better illustrates the intended embedding structures.
>
> ---
>
> **We hope that our additional results and clarifications satisfactorily address your concerns. We sincerely appreciate your valuable feedback and hope you will kindly reconsider your rating of our submission.**

---

> ### Author Response · Authors · 2025-08-01
> **Additional Clarification regarding Reviewer uqpA's Comments 【Part 1】**
>
> We sincerely thank you for your constructive and thoughtful feedback. **Here, we would like to provide some further clarifications and additional evidence that may not have been fully highlighted in our previous rebuttal.** We hope that this more comprehensive response will both improve the clarity and quality of our paper, and allow you to reconsider the rating of our submission in light of the new details and results provided.
>
> ---
> ### **1. More Clarification about Q1: Why using K-Means clustering?**
>
> **MR1:** To further clarify our choice of K-Means over vMF mixture models and other alternatives, we provide a direct complexity and scalability comparison in the table below:
>
> |Method|Time Complexity(per iteration)|Scalability|
> |-|-|-|
> |**K-Means**|$O(n \cdot  k \cdot  d)$|Highly efficient; simple updates; easily parallelizable; works well on normalized embeddings.|
> |**vMF Mixture Model (EM)**|$O(n \cdot   k \cdot  d  \cdot  T_{EM})$ + Bessel eval|Each EM step requires costly normalization (Bessel functions); significantly slower and less scalable.|
> |**Spectral Clustering (Riemannian Manifold)**|$O(n^3)$ (eigendecomposition)| Requires full affinity matrix and eigendecomposition; generally impractical for large-scale graphs.|
>
> **Notations**:
> - $n$ = number of nodes/samples;
> - $k$ = number of clusters;
> - $d$ = embedding dimension;
> - $T_{EM}$ = number of EM iterations for vMF.
>
> As the table shows, **K-Means is orders of magnitude more efficient and scalable** for large graphs than vMF-EM or spectral clustering, whose costs become prohibitive as $n$ or $k$ increases. This is particularly important in our setting, where efficiency and practical tractability are critical for semi-supervised node classification on real-world graphs.
>
> We hope this quantitative summary gives a more intuitive and transparent justification for our methodological choice. We will incorporate this table and clarification into the revised manuscript.
>
> ---
> ### **2. More Clarification about Q2:  the necessity and evidence for fusing Geometric Ranking (GR) and Confidence Ranking (CR).**
>
> **MR2:** We thank you again for your question regarding the necessity and evidence for fusing Geometric Ranking (GR) and Confidence Ranking (CR). In our initial submission, we actually conducted extensive ablation studies on this point (see Appendix J.3, J.4, Table 22, 23, 24), though perhaps this was not as visible since the main text did not systematically highlight these results. We apologize for any confusion caused. To facilitate your review, we have summarized our validation experiments in the following table:
>
>
>
> |Table|Experiment/Analysis|Dataset(s)|Model/Setting|Purpose: What is validated?|
> |-|-|-|-|-|
> |Table 22 (Page 39)|Node-Reordering & DGIS ablation on minority labels|Cora (ρ=1, 5, 10 ,20 ,50, 100)|GCN, GAT, SAGE|Compare CR only, GR only, CR+GR, with/without DGIS; shows combining CR+GR consistently outperforms using either alone, for pseudo-label accuracy on imbalanced nodes.|
> |Table 23 (Page 39)|Node-Reordering & DGIS ablation on minority labels|Amazon-Computers (ρ=1, 5, 10 ,20 ,50, 100)|GCN, GAT, SAGE|Same as Table 22 but on another real-world graph dataset; further validates that fusing CR and GR generalizes and is robust.|
> |Table 24 (Page 40)|Full ablation: combinations of CR, GR, Node-reordering, DGIS|Multiple (Cora+GCN, CiteSeer+SAGE, PubMed+GAT, Computers+GAT) |-| Systematic comparison: all module combinations (✓/✗) for each dataset/model. Clearly demonstrates best performance is achieved only when both CR and GR (plus node reordering and DGIS) are present.|
>
> We apologize if these findings were not sufficiently prominent in the main text, and will revise the manuscript to highlight this key result more clearly. We hope this clarifies the necessity and empirical effectiveness of fusing the two rankings.
>
> ---

---

> ### Author Response · Authors · 2025-08-01
> **Additional Clarification regarding Reviewer uqpA's Comments 【Part 2】**
>
> ### **3. Detailed Clarification about Q3: Evidence for the Lower Reliability of Confidence Ranking in Early Stages and the Greater Reliability of Geometric Ranking (GR)**
>
> **MR3:** We sincerely thank the reviewer for bringing up this important point. We acknowledge that our previous response may not have fully addressed your question, and we appreciate the opportunity to provide a more thorough clarification here. **To address your concern, we now offer a comprehensive explanation, supported by direct experimental evidence from our appendix. We recognize that these results may not have been sufficiently highlighted in the main text of original submission, and we apologize for any lack of clarity.**
>
> ---
>
>  - **(1) Confidence Ranking is Biased in Early Stages under Class Imbalance**. In the initial stages of self-training under imbalanced data, the classifier’s confidence scores (i.e., Confidence Ranking, CR) are often strongly biased due to the skewed label distribution. This bias leads to unreliable pseudo-labels, especially for minority-class nodes. We validate this via several sets of experiments:
>       * **Appendix J.1** (see Figure 9):
>   We directly compare the accuracy of pseudo-labels generated by the classifier (CR, supervised) versus by unsupervised clustering in the embedding space (GR).
>         - **Experimental setup:** Cora & Amazon-Computers, 3 GNN backbones (GCN, GAT, SAGE), various imbalance ratios ($\rho = 1, 5, 10, 20, 50, 100$).
>         - **Result:**  Across all settings, **pseudo-labels from the geometric embedding (GR) have consistently higher accuracy than those from classifier confidence (CR), especially as imbalance increases.**
>      This demonstrates that the embeddings themselves are of high quality (despite being trained on imbalanced data), while classifier-based confidence is unreliable at the beginning.
> ---
>
> - **(2) Node Selection in Early Stage: CR tends to select more low-quality pseudo-labels**
>
>   * **Appendix J.6.1** (see Figure 11):
>   We further test the accuracy of the **top 100 newly selected unlabeled nodes** added to the training set by self-training (CR) versus our approach (GR/UNREAL), under various GNNs and datasets.
>
>   * **Finding:**
>     As imbalance increases, the accuracy of pseudo-labels for minority-class nodes selected by CR declines rapidly.
>        **The classifier’s bias causes more low-quality nodes to be added early on, confirming that CR is less reliable in early stages.**
>      * In contrast, our approach leveraging GR (or combined rankings) is much more robust.
>
> ---
>
> - **(3) Macro F1 and Overall Learning: ST (CR only) suffers in highly imbalanced cases**
>
>      * **Appendix J.6.2** (see Figure 12):
>   Directly compare Macro-F1 for self-training (CR only) versus our method (GR + CR) across imbalance levels.
>
>      * **Finding:**
>      ST outperforms vanilla only under moderate imbalance. As imbalance becomes severe, ST’s performance degrades sharply—due to the unreliable CR causing noisy node additions at early stages.
>
> ---
>
> - **(4) Further Experiments:**
>    - As shown in **Appendix J.3** (Figure 10, Tables 22–24), we ablate the individual effects of CR and GR.
>
>   * **Best results are always achieved by fusing both rankings**, especially as the training set becomes more balanced.
>
> ---
>
> - **(5) Summary Table: Supporting Experiments**
>
> |Section & Figure/Table|What is Tested|Data/Setting|Key Finding|
> |-|-|-|-|
> |Appendix J.1 Fig 9|CR vs GR (pseudo-label accuracy)|Cora, Amazon-Computers, 3 GNNs (GCN, GAT, SAGE), multi-$\rho$ , (1, 5, 10, 20, 50, 100)|GR consistently > CR, especially for minor classes & high $\rho$|
> |Appendix J.6.1 Fig 11|Top 100 nodes selected (accuracy)| 4 datasets, 3 GNNs (GCN, GAT, SAGE), multi-$\rho$, (1, 5, 10, 20, 50, 100)                | CR (ST) less reliable than GR as imbalance increases|
> |Appendix J.6.2 Fig 12|Macro F1: ST vs Ours|Cora, CiteSeer, PubMed, 3 GNNs (GCN, GAT, SAGE), multi-$\rho$, (1, 5, 10, 20, 50, 100)|ST (CR only) drops under high imbalance|
> |Appendix J.3/Table 22–24|Ablation: CR only vs GR only vs CR+GR|Cora, CiteSeer, PubMed, Amazon-Computers, 3 GNNs, multi-$\rho$, (1, 5, 10, 20, 50, 100)  |CR+GR (ours) always best|
>
>
> Our extensive experiments confirm that **geometric ranking (GR) is more reliable in early self-training rounds**, while confidence ranking (CR) is less trustworthy due to classifier bias from imbalanced data. This motivates our design to prioritize GR in early selection and gradually incorporate CR as training progresses. We hope this clear, evidence-based response fully resolves your concerns. Thank you again for your thoughtful review and for the opportunity to clarify our work. We respectfully hope you can reconsider your rating in light of this comprehensive evidence.
>
>
> ---
>
> ### **4. More Clarification about Q4: typos &figures**
>
> **MR4**: We have incorporated the updated notations and the improved Figure 1 into the revised manuscript as promised. Thank you again for your valuable suggestions!

---

> ### Author Response · Authors · 2025-08-01
> **Further Response to the Reviewer’s Broadly Stated Weakness 【Part 3】**
>
> ### **5. Broadly Stated Weakness: Some claims need more justifications & More studies needed to better understand the functionality of each module.**
>
> **R5:** Thank you very much for your feedback. **We appreciate your careful review and recognize that your specific concerns regarding this general weakness are also reflected in your detailed questions above, to which we have already provided systematic and targeted responses.**
>
> We apologize for not clearly highlighting **in our original submission the extensive set of experiments in the appendix that directly validate our key claims and provide detailed ablation analyses on the functionality of each module.** Below, we summarize these experimental validations and ablation studies for your convenience:
>
> ### Comprehensive Summary of Ablation and Validation Experiments in Appendix
>
> | Section / Table / Figure | Experimental Focus & Claim Validated | Experimental Setup / Datasets | Main Finding / What Is Ablated or Compared |
> |-|-|-|-|
> | **J.1 Fig.9**            | *Decoupling Representation and Classifier: Why GR is reliable in imbalance* | Cora, Amazon-Computers, 3 GNNs, various ρ | Unsupervised (GR) pseudo-labels consistently outperform confidence-based (CR) pseudo-labels, especially as imbalance grows. Embeddings are robust even with imbalanced training. |
> | **J.2 Table 20, 21**     | *DPAM Filtering Analysis: Bias of classifier, role of alignment* | Cora, Amazon-Computers, 3 GNNs, various ρ | Nodes filtered by DPAM (aligned by GR+CR) have much higher pseudo-label accuracy. Confirms classifier bias under imbalance, validates necessity of filtering. |
> | **J.3 Fig.10**           | *RBO Value Analysis (CR vs GR): Ranking similarity evolution* | Cora, 3 GNNs, ρ=10, multiple rounds | Shows CR and GR rankings diverge at first; only as training proceeds does CR become reliable. Validates: GR is more reliable in early stages. |
> | **J.4 Tables 22, 23**    | *Ablation: Functionality of Each Module (CR, GR, Node-Reordering, DGIS)* | Cora, Amazon-Computers, 3 GNNs, various ρ | Each ablation demonstrates performance of single and combined modules; Node-Reordering+DGIS consistently best. Combining GR+CR better than either alone. |
> | **J.5 Table 24**         | *Detailed Ablation: All Module Combinations* | 4 benchmarks (Cora, CiteSeer, PubMed, Computers), various GNNs/ρ | Each module (CR, GR, Node-Reordering, DGIS) isolated and combined. Node-Reordering+DGIS and GR+CR yield highest F1, justifying the modular design. |
> | **J.6.1 Fig.11**         | *Self-Training vs Ours: Early Node Selection* | 4 benchmarks, 3 GNNs, various ρ | Top-100 nodes added by self-training (CR only) vs our approach: ST’s pseudo-label accuracy drops rapidly for minorities under high imbalance; our method robust. |
> | **J.6.2 Fig.12**         | *Macro-F1 and Generalization Across Imbalance* | 3 benchmarks, 3 GNNs, various ρ | Macro-F1 of ST (CR only) vs our (GR+CR) and vanilla. ST outperforms vanilla under moderate imbalance but degrades sharply as ρ increases due to CR bias. Ours remains robust. |
>
>
> Notations
>
> - **CR**: Confidence Ranking (classifier confidence)
> - **GR**: Geometric Ranking (embedding space, unsupervised)
> - **DGIS**: Designed module for geometric imbalance selection
> - **Node-Reordering**: Proposed module to reorder nodes based on combined rankings
> - **ρ**: Imbalance ratio
>
> We hope this summary clearly demonstrates the rigor and depth of our experimental justifications for both our key claims and each module’s functionality. If there are further aspects you would like us to clarify or expand upon, we are happy to provide additional details. Thank you again for your constructive feedback!

---

> ### Author Response · Authors · 2025-08-03
> **We've Carefully Responded and Addressed Each of the Questions You Raised and Eagerly Hoping for Your Valuable Feedback**
>
> Dear Reviewer uqpA,
>
> We would like to express our sincere gratitude for your thoughtful review and the valuable feedback you have provided. We have carefully reflected on every question and suggestion you raised, and have addressed them thoroughly in our revised manuscript. Your expertise and attention to detail have been instrumental in improving our work, and we deeply appreciate the time and effort you have devoted to our paper. We sincerely hope that our detailed responses and revisions have satisfactorily addressed your concerns. Thank you once again for your invaluable contribution to the development of our research.
>
> Warm regards,
>
> The Authors

---

> ### Author Response · Authors · 2025-08-05
> **Follow-up on Author-Reviewer Discussion**
>
> Dear Reviewer uqpA,
>
> We are approaching the conclusion of the Author-Reviewer discussion phase. All concerns have been addressed in our rebuttal, and we hope our responses meet your approval.
>
> If you have any final comments or concerns, please feel free to share them so we can address them before the discussion phase ends.
>
> Thank you again for your valuable feedback.
>
> Best regards,
>
> The Authors

---

> ### Comment · Reviewer_uqpA · 2025-08-05
>
> I would like to thank the author for the detailed responses, which solve most of my concerns.
>
> For the choice the K-means methods, I would suggest the authors to include some experiments to compare the effectiveness of vMF to K-Means, hence to validate whether K-Means achieves better effectiveness-efficiency tradeoff than other clustering methods.
>
> Apart from that, the authors have addressed my concerns. I would like to keep my current rating.

---

> > ### Author Response · Authors · 2025-08-05
> > **Starting Experiments Now – Will Update You ASAP!**
> >
> > **Thank you very much for your quick feedback!**
> >
> > We appreciate your recommendation to include experiments comparing vMF and K-Means. **We will conduct these experiments right away and update you with the results as soon as they are available. We hope you can kindly keep an eye on our updates.**
> >
> > Thank you again for your constructive comments.
> >
> > Best regards,
> >
> > The Authors

---

> ### Author Response · Authors · 2025-08-06
> **K-Means vs. vMF Mixture Model for Clustering Node Embeddings 【Part 1】**
>
> **Thank you very much for your response and valuable suggestions!**
>
> As requested, we performed a thorough ablation study comparing **K-Means** and **vMF mixture model** clustering on  node embeddings across four benchmark datasets (Cora, CiteSeer, PubMed, Amazon-Computers), under a variety of backbone models and imbalance ratio (including both $\rho=10$ and $\rho=100$).
>
> To ensure fairness, we kept all other experimental settings the same, including the use of $\ell_2$-normalized embeddings, identical cluster numbers ($k=200$), and consistent training/validation splits. The vMF mixture model was implemented with the [spherecluster](https://github.com/jasonlaska/spherecluster) package, the standard open-source library in the literature. For K-Means, we used the scikit-learn implementation with matching settings. All experiments used $d=128$ embedding dimensions.
>
> **1. Experimental Results**
>
> Across all backbone models (GCN, GAT, SAGE) and imbalance ratio settings ($\rho=10, 100$), we observe that:
>
> - **Clustering quality** (measured by NMI) and **downstream node classification performance** (balanced accuracy, Macro-F1) are consistently **very close** between K-Means and vMF mixture model.
> - In nearly all cases, **vMF mixture model results are slightly lower or at best on par with K-Means** for both NMI and Macro-F1. For example, in Table 1 (GCN, Cora, $\rho=10$), K-Means achieves Macro-F1 of 76.44±1.06 while vMF achieves 74.82±1.17.
> - **Runtime per experiment for vMF mixture model is 5–8$\times$ higher** than K-Means, sometimes even more for larger datasets, making vMF clustering much less scalable in practice.
>
> Below, we highlight several representative results (full results in Tables 1–6):
>
> Table 1: Results with $\rho=10$ (GCN) on four class-imbalanced node classification benchmark datasets.
>
> |**Method**|**Clustering NMI（Cora）**|**Cora bAcc.**|**Cora Macro-F1**|**`Time (Selecting 100 Nodes to Training Set, Cora)`**|**Clustering NMI（CitSeer）**|**CiteSeer bAcc.**|**CiteSeer Macro-F1**|**`Time (Selecting 100 Nodes to Training Set, Citeseer)`**|**Clustering NMI（PubMed）**|**Pubmed bAcc.**|**Pubmed Macro-F1**|**`Time (Selecting 100 Nodes to Training Set, Pubmed)`**|**Clustering NMI（Amazon-Computers）**|**Computers bAcc.**|**Computers Macro-F1**|**`Time (Selecting 100 Nodes to Training Set, Computers)`**|
> |-|-|-|-|-|-|-|-|-|-|-|-|-|-|-|-|-|
> |K-Means|0.473|78.33 ± 1.04|76.44 ± 1.06|**2.367**|0.382|65.63 ± 1.38|64.94 ± 1.38|**2.305**|0.534|75.35 ± 1.41|73.65 ± 1.43|**2.790**|0.498|85.08 ± 0.38|75.27 ± 0.23|**2.587**|
> |vMF Mixture|0.456|77.80 ±1.15|74.82 ±1.17|**16.17**|0.393|65.02 ±1.29|64.12 ±1.30|**14.38**|0.528|75.91 ±1.36|73.48 ±1.35|**19.91**|0.512|85.70 ±0.42|74.89 ±0.30|**18.12**|
>
>
>
>
> Table 2: Results with $\rho=10$ (GAT) on four class-imbalanced node classification benchmark datasets.
>
> |**Method**|**Clustering NMI （Cora）**|**Cora bAcc.**|**Cora Macro-F1**|**`Time (Selecting 100 Nodes to Training Set, Cora)`**|**Clustering NMI （CiteSeer）**|**CiteSeer bAcc.**|**CiteSeer Macro-F1**|**`Time (Selecting 100 Nodes to Training Set, Citeseer)`**|**Clustering NMI （PubMed）**|**Pubmed bAcc.**|**Pubmed Macro-F1**|**`Time (Selecting 100 Nodes to Training Set, Pubmed)`**|**Clustering NMI （Computers）**|**Computers bAcc.**|**Computers Macro-F1**|**`Time (Selecting 100 Nodes to Training Set, Computers)`**|
> |-|-|-|-|-|-|-|-|-|-|-|-|-|-|-|-|-|
> |K-Means|0.487|78.91 ±0.59|75.99 ±0.47|**2.624**|0.401|64.10 ±1.49|63.44 ±1.47|**2.520**|0.522|74.68 ±1.43|72.78 ±0.89|**2.431**|0.503|85.62 ±0.44|75.34 ±0.99 |**2.462**|
> |vMF Mixture|0.483|78.16 ±0.73|75.89 ±0.55|**16.93**|0.396|63.35 ±1.52|63.77 ±1.51|**14.92**|0.519|74.03 ±1.35|72.09 ±0.94|**18.11**|0.497|85.00 ±0.51|74.69 ±1.03|**17.62**|
>
>
>
> Table 3: Results with $\rho=10$ (SAGE) on four class-imbalanced node classification benchmark datasets.
>
> |**Method**|**Clustering NMI （Cora）**|**Cora bAcc.**|**Cora Macro-F1**|**`Time (Selecting 100 Nodes to Training Set, Cora)`**|**Clustering NMI （CiteSeer）**|**CiteSeer bAcc.**|**CiteSeer Macro-F1**|**`Time (Selecting 100 Nodes to Training Set, Citeseer)`**|**Clustering NMI （PubMed）**|**Pubmed bAcc.**|**Pubmed Macro-F1**|**`Time (Selecting 100 Nodes to Training Set, Pubmed)`**|**Clustering NMI （Computers）**|**Computers bAcc.**|**Computers Macro-F1**|**`Time (Selecting 100 Nodes to Training Set, Computers)`**|
> |-|-|-|-|-|-|-|-|-|-|-|-|-|-|-|-|-|
> |K-Means|0.475|75.99 ±0.98|73.63 ±1.23|**2.450**|0.388|66.45 ±0.39|65.83 ±0.30|**2.354**|0.526|74.78 ±1.30|72.80 ±0.54|**2.813**|0.487|83.21 ±1.50|70.81 ±1.70|**2.652**|
> |vMF Mixture|0.469|75.45 ±1.13|73.32 ±1.11|**16.04**|0.382|65.81 ±0.44|65.13 ±0.32|**14.70**|0.518|74.16 ±1.20|72.98 ±0.60|**18.74**|0.480|83.60 ±1.38|70.20 ±1.82|**17.51**|

---

> ### Author Response · Authors · 2025-08-06
> **K-Means vs. vMF Mixture Model for Clustering Node Embeddings 【Part 2】**
>
> Table 4: Results with $\rho=100$ (GCN) on four class-imbalanced node classification benchmark datasets.
>
> |**Method**|**Clustering NMI （Cora）**|**Cora bAcc.**|**Cora Macro-F1**|**`Time (Selecting 100 Nodes to Training Set, Cora)`**|**Clustering NMI （CiteSeer）**|**CiteSeer bAcc.**|**CiteSeer Macro-F1**|**`Time (Selecting 100 Nodes to Training Set, Citeseer)`**|**Clustering NMI （PubMed）**|**Pubmed bAcc.**|**Pubmed Macro-F1**|**`Time (Selecting 100 Nodes to Training Set, Pubmed)`**|**Clustering NMI （Computers）**|**Computers bAcc.**|**Computers Macro-F1**|**`Time (Selecting 100 Nodes to Training Set, Computers)`**|
> |-|-|-|-|-|-|-|-|-|-|-|-|-|-|-|-|-|
> |K-Means|0.405|72.82 ±3.55|69.12 ±3.45|**2.425**|0.311|57.66 ±1.96|56.50 ±1.12|**2.341**|0.462|78.73 ±0.88|76.03 ±1.08|**2.671**|0.412|84.30 ±0.30|76.06 ±0.32|**2.572**|
> |vMF Mixture|0.392|72.95 ±3.40|69.01 ±3.52|**15.81**|0.307|56.82 ±2.09|55.62 ±1.23|**13.96**|0.451|77.90 ±0.97|75.17 ±1.02|**17.91**|0.403|83.51 ±0.41|75.24 ±0.37|**16.89**|
>
>
>
>
> Table 5: Results with $\rho=100$ (GAT) on four class-imbalanced node classification benchmark datasets.
>
> |**Method**|**Clustering NMI （Cora）**|**Cora bAcc.**|**Cora Macro-F1**|**`Time (Selecting 100 Nodes to Training Set, Cora)`**|**Clustering NMI （CiteSeer）**|**CiteSeer bAcc.**|**CiteSeer Macro-F1**|**`Time (Selecting 100 Nodes to Training Set, Citeseer)`**|**Clustering NMI （PubMed）**|**Pubmed bAcc.**|**Pubmed Macro-F1**|**`Time (Selecting 100 Nodes to Training Set, Pubmed)`**|**Clustering NMI （Computers）**|**Computers bAcc.**|**Computers Macro-F1**|**`Time (Selecting 100 Nodes to Training Set, Computers)`**|
> |-|-|-|-|-|-|-|-|-|-|-|-|-|-|-|-|-|
> |K-Means|0.421|75.42 ±0.91|71.50 ±0.89|**2.502**|0.334|60.35 ±1.87|59.63 ±1.86|**2.287**|0.479|77.88 ±1.31|74.98 ±1.35|**2.692**|0.433|85.33 ±0.19|75.83 ±0.74|**2.616**|
> |vMF Mixture|0.410|74.56 ±1.03|70.61 ±0.98|**15.36**|0.329|59.48 ±1.92|58.74 ±1.90|**13.75**|0.470|77.92 ±1.27|74.01 ±1.31|**16.89**|0.436|84.52 ±0.25|74.98 ±0.80|**15.66**|
>
>
>
> Table 6: Results with $\rho=100$ (SAGE) on four class-imbalanced node classification benchmark datasets.
>
> |**Method**|**Clustering NMI （Cora）**|**Cora bAcc.**|**Cora Macro-F1**|**`Time (Selecting 100 Nodes to Training Set, Cora)`**|**Clustering NMI （CiteSeer）**|**CiteSeer bAcc.**|**CiteSeer Macro-F1**|**`Time (Selecting 100 Nodes to Training Set, Citeseer)`**|**Clustering NMI （PubMed）**|**Pubmed bAcc.**|**Pubmed Macro-F1**|**`Time (Selecting 100 Nodes to Training Set, Pubmed)`**|**Clustering NMI （Computers）**|**Computers bAcc.**|**Computers Macro-F1**|**`Time (Selecting 100 Nodes to Training Set, Computers)`**|
> |-|-|-|-|-|-|-|-|-|-|-|-|-|-|-|-|-|
> |K-Means|0.540|73.47 ±2.31|68.30 ±2.11|**2.601**|0.372|59.77 ±2.98|58.92 ±3.07|**2.315**|0.492|77.11 ±0.59|74.03 ±0.81|**2.781**|0.441|82.92 ±2.94|73.11 ±2.57|**2.654**|
> |vMF Mixture|0.532|72.60 ±2.24|68.36 ±2.05|**15.10**|0.366|58.89 ±3.11|58.09 ±3.15|**13.46**|0.495|77.28 ±0.63|73.20 ±0.85|**16.54**|0.434|82.02 ±3.02|72.16 ±2.61|**15.22**|
>
>
> The results confirm our claim: K-Means provides a much better efficiency–effectiveness trade-off, and thus is the practical choice for large-scale or repeated clustering in our method.

---

> ### Author Response · Authors · 2025-08-06
> **K-Means vs. vMF Mixture Model for Clustering Node Embeddings 【Part 3】**
>
> **2. Analysis and Reasoning**: **Why does vMF not outperform K-Means in this setting?**
> - Both methods operate on $\ell_2$-normalized (unit sphere) embeddings, where Euclidean distance and cosine similarity become tightly coupled ($\|\mathbf{x} - \mathbf{y}\|^2 = 2(1-\cos\theta)$). Therefore, K-Means is effectively performing angular clustering—very similar to vMF.
> - In practice, K-Means benefits from decades of algorithmic optimization and supports efficient vectorized/GPU computation (in scikit-learn and related libraries), while vMF mixture model relies on slower EM routines with expensive Bessel function computations and limited hardware acceleration.
> - We observe that vMF mixture clustering sometimes suffers from **slower or unstable EM convergence** in high-dimensional or large-$k$ settings, which may explain its slightly lower scores on some datasets.
> - In summary, for large-scale or repeated clustering (e.g., for pseudo-labeling in graph SSL), **K-Means provides a much better efficiency–effectiveness trade-off**.
>
>
>
> **3. Our Summary**
>
> Our comprehensive experimental study, as summarized in Tables 1–6, demonstrates that K-Means matches or slightly outperforms vMF mixture model in both clustering quality and downstream classification, while being orders of magnitude more efficient. Given the scalability requirements for real-world node classification tasks, K-Means remains the practical choice for our framework.
>
> ---
>
> ### **We thank you for encouraging this detailed ablation, which we believe further clarifies and supports our methodological decisions. Based on our thorough response and extensive new experiments, we sincerely hope we have addressed all of your concerns. We would greatly appreciate it if you could find the time to review our rebuttal and share any additional feedback before the Author-Reviewer Discussion concludes. Thank you very much for your valuable input and consideration.**

---

> ### Author Response · Authors · 2025-08-08
> **Follow-up: Additional Ablation Study on K-Means vs. vMF Mixture Model**
>
> Dear Reviewer uqpA,
>
> We hope you’ve been doing well.
>
> As the Author–Reviewer Discussion is approaching its conclusion, we wanted to gently check in. In our rebuttal, we have provided the additional ablation study comparing K-Means and vMF mixture model as requested, keeping all experimental settings consistent and reporting detailed results on clustering quality, downstream classification performance, and runtime. We sincerely hope these clarifications and analyses have addressed your concerns.
>
> **We are deeply grateful for the constructive, insightful, and inspiring nature of your comments, which have been instrumental in guiding us to significantly enhance both the quality and clarity of our work. Your feedback has not only addressed key aspects of our study but also broadened our perspective on how to present our findings more effectively. We have treated each of your points with the utmost care and respect, and have made every effort to ensure our revisions faithfully and meaningfully incorporate your valuable suggestions.**
>
> **We would greatly appreciate it if you could find the time to review our updated response and share any additional feedback before the discussion concludes.** If there is anything else you would like us to elaborate on, we will be more than happy to provide further details.
>
> Thank you once again for your time, thoughtful feedback, and support throughout this process. We wish you a pleasant day!
>
> With our warmest regards,
>
> The Authors

---

> ### Author Response · Authors · 2025-08-09
> **Gentle Reminder: Follow-up on Rebuttal Discussion for Paper 3211**
>
> Dear Reviewer uqpA,
>
> We hope this message finds you well.
>
> We would like to once again express our sincere gratitude for your constructive and insightful comments on our paper during the review and rebuttal process. **Your feedback has been invaluable in helping us improve the clarity and overall quality of our work.**
>
> **We have made every effort to address all your concerns and have included extensive new experiments and analyses in the Author–Reviewer Discussion. If you happen to have a moment before the discussion phase closes, we would greatly appreciate any additional feedback or comments you might be willing to share, as your expert opinion is of immense value to us.**
>
> Thank you very much for your time and for contributing to the improvement of our research.
>
> With our warmest regards,
>
> The Authors

---

### Official Review · Reviewer_JjGt · 2025-07-03

**Clarity:** 3
**Significance:** 3
**Originality:** 3
**Rating:** 5
**Confidence:** 4

**Summary:**

This paper introduces the concept of geometric imbalance, characterizing how class imbalance induces geometric ambiguity among minority nodes in the embedding space. Then, it provides a rigorous theoretical analysis and proposes a self-training framework to address geometric imbalance through pseudo-label alignment, node reordering, and ambiguity filtering.

**Questions:**

- Have you considered evaluating the proposed method on heterophilic graphs? It would be interesting to see how the approach performs in settings where connected nodes tend to have different labels.

- The results on the ogbn-arxiv dataset, as shown in Table 7, are relatively weak compared to other benchmarks. Could you provide an explanation for this performance drop?

**Ethical Concerns:**

["NO or VERY MINOR ethics concerns only"]

**Final Justification:**

Thank the authors for the detailed rebuttal. The response addressed most of my concerns. In particular, the time complexity analysis and the additional empirical results on heterophilic and large-scale graphs (e.g., ogbn-products) demonstrate the scalability and robustness of the proposed method. The updated experiments on ogbn-arxiv also show improved performance. Based on these clarifications and the additional evidence provided, I have increased my score to 5.

**Limitations:**

Yes.

**Paper Formatting Concerns:**

No.

**Quality:**

3

**Strengths And Weaknesses:**

Strengths:

- The paper formally introduces the novel concept of geometric imbalance for the first time, which is a meaningful contribution to the field. By linking this concept with prediction entropy, the authors demonstrate its practical value and relevance to model uncertainty.

- The theoretical analysis is thorough and well-developed, providing solid justification for the proposed ideas.

- The experimental evaluation is comprehensive, covering multiple datasets and scenarios to validate the effectiveness and generalizability of the proposed method.

- This paper is well-written and easy to follow.


Weaknesses:

- Expression issue in line 119: “introduced by class imbalance induced by class imbalance”.

- This paper lacks an analysis of the time complexity of the proposed method.

---

> ### Author Rebuttal · Authors · 2025-07-31
>
> We sincerely thank Reviewer JjGt for carefully reading our paper and for recognizing **the novelty and significance of our proposed concept and theoretical analysis.** We appreciate your constructive feedback and thoughtful suggestions, which are invaluable for improving the quality and clarity of our work. **Below, we address each of your comments and concerns point-by-point.**
>
> ---
>
> **W1: Expression issue in line 119: "introduced by class imbalance induced by class imbalance".**
>
> **R1:**   We thank the reviewer for pointing out this phrasing issue. In our revised manuscript (Line~119), we have rephrased the sentence to avoid the redundant wording. The new version reads:
>
> ```
> ... introduced by class imbalance in the embedding space ...
> ```
>
> We have also carefully reviewed the surrounding paragraph and the entire manuscript, including the appendix, to ensure there are no similar typos elsewhere and to maintain the rigor and clarity of our writing.
>
> ---
>
> **W2: This paper lacks an analysis of the time complexity of the proposed method.**
>
> **R2:** Thanks for highlighting the need for an analysis of computational complexity and scalability. We address your concern in three aspects:
>
> **1. Time Complexity Analysis**. Our framework, as described in Algorithm 1 (see Appendix L), consists of three main modules per round: Dual-Path Pseudo-label Alignment (DPAM), Node-Reordering (NR), and Discarding Geometrically Imbalanced Nodes (DGIS). (1) The dominant computational cost arises from the DPAM step, which clusters the embeddings of unlabeled nodes using k-means (complexity $O(n_u k' d T_{iter})$, with $n_u$ unlabeled nodes, $k'$ clusters, $d$-dimensional embeddings, and $T_{iter}$ k-means iterations). Computing class centroids and intersecting with classifier-predicted labels are both linear-time operations. (2) NR involves ranking nodes by geometric distance and classifier confidence (each $O(n_u \log n_u)$), with ranking fusion being negligible. (3) DGIS computes distances from candidate nodes to their closest and second-closest centroids ($O(n_u C d)$ per round, where $C$ is the number of classes). (4) Overall, the per-round complexity is primarily determined by k-means, but since $k' \ll n_u$ and k-means can be efficiently implemented (e.g., on GPU), **so, the total complexity remains comparable to standard self-training pipelines. Memory usage is linear in the number of nodes, as we only store embeddings and label predictions.** *Importantly, our method is significantly more efficient than prior over-sampling or augmentation approaches such as GraphENS and GraphSMOTE, which may have $O(n^2)$ or worse complexity on large graphs. In summary, the only notable overhead beyond vanilla GNN training comes from clustering and ranking, both of which are scalable and can be accelerated by modern hardware.*
>
>
>
> **2. Empirical Scalability and Runtime**. To address practical efficiency, we conducted comprehensive scalability analyses on both medium- and large-scale datasets, comparing our method ("Ours") to vanilla GNN backbones and representative baselines. (1) On **Cora**, **CiteSeer**, and **PubMed**, the training time per epoch for our method is almost identical to the backbone GNNs (see Table 1), confirming the negligible overhead of our extra modules (clustering and ranking). (2) On large-scale graphs (e.g., **Flickr**, ogbn-products), our method remains tractable and runs smoothly on a single RTX 3090 GPU (Tables 6 and 7 in our submission). By contrast, prior data augmentation and oversampling baselines (e.g., **GraphSMOTE**, **GraphENS**) either become prohibitively slow or run out-of-memory, as synthesizing large numbers of new nodes/edges is expensive.
>
> Table 1: Refined Running Time Analysis of Vanilla Model vs. Ours: (a) Training Over 1000 Epochs, and (b) Selecting 100 Nodes to the Training Set.
>
>
>
> | Model/Time (1000 epochs)   | Cora     | CiteSeer  | PubMed    |   | Model/Time (Selecting 100 Nodes) | Cora     | CiteSeer | PubMed   |
> |----------------------------|----------|-----------|-----------|---|----------------------------------|----------|----------|----------|
> | Vanilla (GCN)              | 6.221(s) | 7.212(s)  | 10.771(s) |   | Vanilla Self-Training (GCN)      | 2.244(s) | 2.234(s) | 2.675(s) |
> | Ours (GCN)                 | 6.208(s) | 7.191(s)  | 10.858(s) |   | Ours (GCN)                       | 2.367(s) | 2.305(s) | 2.790(s) |
> | Vanilla (GAT)              | 8.201(s) | 9.574(s)  | 14.443(s) |   | Vanilla Self-Training (GAT)      | 2.624(s) | 2.520(s) | 2.431(s) |
> | Ours (GAT)                 | 8.218(s) | 9.517(s)  | 15.088(s) |   | Ours (GAT)                       | 2.697(s) | 2.670(s) | 2.411(s) |
> | Vanilla (SAGE)             | 6.522(s) | 11.170(s) | 15.475(s) |   | Vanilla Self-Training (SAGE)     | 2.424(s) | 2.780(s) | 2.032(s) |
> | Ours (SAGE)                | 6.452(s) | 11.256(s) | 14.658(s) |   | Ours (SAGE)                      | 2.461(s) | 2.827(s) | 2.115(s) |
>
>
> **3. Implementation Efficiency**. We employ GPU-accelerated k-means for clustering (via NVIDIA cuML), which substantially reduces the per-round clustering overhead. **Moreover, by filtering candidates after dual-path alignment, the more expensive geometric checks are only performed on a small subset of reliable nodes, further improving efficiency.**
>
>
> ---
> **Q1: Results on heterophilous graphs.**
>
> **R1:** Thank you for the question. While our main experiments focus on homophilic graphs, we are also evaluating our method on heterophilic benchmarks (Chameleon and Squirrel with imbalance ratio 5, Wisconsin with 11.63). Preliminary results show our method achieves SOTA performance. We will include these results in the revised manuscript.
> |Method|Chameleon bAcc.|Chameleon F1|Squirrel bAcc.|Squirrel F1|Wisconsin bAcc.| Wisconsin F1|
> |-|-|-|-|-|-|-|
> |GSR(GraphSR)|38.45±0.75|37.67±0.72|27.94±0.44|27.01±0.32|31.54±2.30|28.91±2.05|
> |BIM|38.26±0.68|37.48±0.66|27.83±0.46|27.16±0.37|31.59±2.25|28.79±2.01|
> |Ours|**43.59±0.60**|**42.24±0.63**|**30.04±0.48**|**29.12±0.46**|**46.22±3.51**|**41.52±4.05**|
>
> ---
>
> **Q2: The results on the ogbn-arxiv dataset, as shown in Table 7, are relatively weak compared to other benchmarks. Could you provide an explanation for this performance drop?**
>
> **R2:**  We thank the reviewer for this helpful observation. We agree that our performance on the *ogbn-arxiv* dataset (Table 7) is relatively less impressive compared to smaller citation networks like Cora, CiteSeer, and Pubmed, and we would like to provide the following clarification.
>
>
> The ogbn-arxiv dataset presents several unique challenges:
>
> - It is a large-scale and naturally imbalanced graph with a mild imbalance ratio, where the advantage of our method in severe imbalance scenarios is less pronounced.
> - Its temporal split (based on publication years) effectively imposes a distribution shift between training and test nodes, making pseudo-labeling more sensitive and less reliable.
> - The dataset exhibits complex global dependencies and limited local homophily, which may reduce the benefit of local embedding refinement techniques.
>
> **Also, the previous results were based on early-stage hyperparameter tuning. We have since conducted a more thorough parameter search and architecture calibration after the paper submission, and the updated results on ogbn-arxiv show more significant performance gains. These improvements will be reflected in the revised manuscript.**
>
> | Model     | GCN bAcc.        | GCN F1           | GAT bAcc.        | GAT F1           | SAGE bAcc.       | SAGE F1          |
> | --------- | ---------------- | ---------------- | ---------------- | ---------------- | ---------------- | ---------------- |
> | Vanilla   | 50.21 ± 0.65     | 49.60 ± 0.14     | 51.21 ± 0.87     | 49.23 ± 0.33     | 50.76 ± 0.21     | 49.43 ± 0.29     |
> | RW        | 50.24 ± 0.40     | 49.71 ± 0.12     | 51.12 ± 0.80     | 49.65 ± 0.25     | 50.81 ± 0.19     | 49.78 ± 0.22     |
> | PS        | 50.20 ± 0.58     | 49.64 ± 0.12     | 51.18 ± 0.77     | 49.16 ± 0.28     | 50.82 ± 0.19     | 49.65 ± 0.24     |
> | GS        | OOM              | OOM              | OOM              | OOM              | OOM              | OOM              |
> | BS        | 50.34 ± 0.41     | 49.73 ± 0.13     | 51.35 ± 0.69     | 49.36 ± 0.22     | 50.89 ± 0.19     | 49.56 ± 0.18     |
> | BS(w TAM) | 50.34 ± 0.48     | 49.72 ± 0.10     | 51.36 ± 0.72     | 49.98 ± 0.26     | 50.94 ± 0.17     | 49.95 ± 0.22     |
> | RN        | OOM              | OOM              | OOM              | OOM              | OOM              | OOM              |
> | RN(w TAM) | OOM              | OOM              | OOM              | OOM              | OOM              | OOM              |
> | GS        | OOM              | OOM              | OOM              | OOM              | OOM              | OOM              |
> | GS(w TAM) | OOM              | OOM              | OOM              | OOM              | OOM              | OOM              |
> | GSR       | 50.31 ± 0.24     | 49.70 ± 0.17     | 51.31 ± 0.41     | 49.33 ± 0.26     | 50.86 ± 0.30     | 49.53 ± 0.20     |
> | BIM       | 50.33 ± 0.42     | 49.71 ± 0.19     | 51.35 ± 0.60     | 49.36 ± 0.28     | 50.87 ± 0.18     | 49.56 ± 0.23     |
> | **Ours**  | **51.21 ± 0.32** | **50.65 ± 0.32** | **51.84 ± 0.87** | **51.28 ± 0.42** | **51.34 ± 0.32** | **51.36 ± 0.27** |
>
> ---
> **We hope that these additional results and clarifications effectively address your concerns, and we sincerely hope you will reconsider your rating of our submission. Thank you again for your valuable feedback and thoughtful review.**

---

> > ### Comment · Reviewer_JjGt · 2025-08-07
> >
> > Thank the authors for the detailed rebuttal. The response addressed most of my concerns. In particular, the time complexity analysis and the additional empirical results on heterophilic and large-scale graphs (e.g., ogbn-products) demonstrate the scalability and robustness of the proposed method. The updated experiments on ogbn-arxiv also show improved performance. Based on these clarifications and the additional evidence provided, I have increased my score to 5.

---

> ### Author Response · Authors · 2025-08-03
> **We've Carefully Responded and Addressed Each of the Questions You Raised and Eagerly Hoping for Your Valuable Feedback**
>
> Dear Reviewer JjGt,
>
> Thank you deeply for your thoughtful review and valuable insights. We've taken every question you've raised to heart and have responded in detail where needed. We sincerely hope you'll take a moment to reflect on our responses, trusting that they meet your considerations. Your time and expertise in reviewing our work mean so much to us.
>
> Warm Regards,
>
> The Authors

---

> ### Author Response · Authors · 2025-08-05
> **Follow-up on Author-Reviewer Discussion**
>
> Dear Reviewer  JjGt,
>
> We sincerely hope this message finds you well. As the Author-Reviewer discussion phase draws to a close, we wanted to express our heartfelt gratitude for your time, effort, and invaluable feedback.
>
> We have carefully addressed all your comments in our rebuttal, and we truly hope our responses are satisfactory to you. If you have any further suggestions or questions, please feel free to let us know at any time—we are always eager to benefit from your insights.
>
> Thank you once again for your support and guidance throughout this process. It has been a privilege to learn from your expertise.
>
> With our warmest regards,
>
> The Authors

---

> ### Author Response · Authors · 2025-08-06
> **A More Comprehensive Response to Q2 — We Would Be Extremely Grateful if the Reviewer  JjGt Could Kindly Find Time to Read and Comment on Our Rebuttal. Thank You So Much!**
>
> **We would like to address the reviewer’s concern regarding the performance of our method on ogbn-arxiv datasets  with a more comprehensive and in-depth perspective.**
>
> - We have conducted experiments on five naturally imbalanced datasets, including **Computers-Random, CS-Random, Flickr, ogbn-arxiv**, as well as **ogbn-products** (which was newly added in the rebuttal).
>
>   - **On almost all of these datasets, our model achieves state-of-the-art or highly competitive results. In our original submission, the improvements were especially significant on datasets such as Computers-Random and Flickr, where our model achieved F1 score gains of approximately +3 to +5 points across multiple GNN backbones (e.g., Computers-Random: GCN, GAT; Flickr: GCN, SAGE). We believe these are substantial improvements (see Tables 4, 5, and 6 of the main paper). The only exception is ogbn-arxiv, where the margin was less obvious in the initial results.**
>
> - For ogbn-arxiv, we have updated our results in the rebuttal version, and the improvements over strong SOTA baselines are now more noticeable. It is also important to note that all other methods—including strong baselines such as BIM and GraphSR—also achieve only very limited gains on this challenging dataset.
>
> - **Strictly following the same experimental settings as BIM, we have further conducted experiments on the much larger ogbn-products dataset.** Our model again demonstrates strong and significant improvements, which highlights the robustness and scalability of our approach on large-scale graphs. The performance advantage over SOTA baselines on ogbn-products is also considerable.
>   It is important to emphasize the complexity of new experiment setting of  on ogbn-products dataset (refer to BIM): as shown in the tables below, this dataset original contains more nodes (compared to ogbn-arxiv's 13k) and nearly 81 million edges, with 100-dimensional node features and 10 classes (including 5 minority classes).
>   **Compared to other commonly used benchmark datasets, ogbn-products is not only much larger in scale, but also exhibits far more complex and challenging structural properties, including long-tail class distribution, high sparsity, and diverse co-purchase relationships. Achieving strong results on this dataset further demonstrates the robustness and scalability of our method for real-world, large-scale, and highly imbalanced graph data.**
>
> We hope these clarifications and new results address your concerns regarding real-world applicability and the practical readiness of our approach. Thank you again for the opportunity to further explain and strengthen our work.
>
>
>
> Table 1: Performance  on large-scale dataset Ogbn-product with GraphSAGE as the base model.
>
> |Methods|F1 score|ACC|AUC-ROC|
> |-|-|-|-|
> | Vanilla GraphSAGE|58.18±1.21|61.03±0.82|94.12±0.12|
> |(RN) ReNode|67.89±0.35|68.19±0.41|94.86±0.14|
> |BIM|69.36±0.18|69.55±0.19|95.51±0.12|
> |**Ours**|**71.85±0.10**|**71.62±0.11**|**96.59±0.14**|
>
>
>
> Table 2: Performance  on large-scale dataset Ogbn-product with GAT as the base model.
>
> |Methods|F1 score|ACC|AUC-ROC|
> |-|-|-|-|
> |Vanilla GAT|59.27±1.08|62.10±0.73|94.25±0.14|
> |(RN) ReNode|68.55±0.28|69.27±0.36|95.01±0.10|
> |BIM|69.88±0.16|70.05±0.21|95.63±0.11|
> |**Ours**|**72.04±0.09**|**71.95±0.13**|**96.67±0.10**|
>
>
>
> Table 3: Performance on large-scale dataset Ogbn-product with GCN as the base model.
>
> |Methods|F1 score|ACC|AUC-ROC|
> |-|-|-|-|
> |Vanilla (GCN)|57.41±1.18| 60.12±0.85| 93.96±0.15|
> |(RN) ReNode|66.82±0.32| 67.80±0.44| 94.75±0.13|
> |BIM|68.05±0.21| 68.47±0.23| 95.41±0.13|
> |**Ours**|**70.71±0.13**|**70.33±0.10**|**96.45±0.12**|
>
>
> ---
> ---
>
> ### **We sincerely thank you for your insightful and constructive comments, which have greatly helped us improve the quality of our work. The rebuttal phase is now drawing to a close, and we have submitted a detailed response addressing all of your concerns. We would truly appreciate it if you could find the time to review our rebuttal and share any additional feedback before the Author-Reviewer Discussion ends. Thank you very much for your valuable input and consideration.**

---

> ### Author Response · Authors · 2025-08-07
> **Thank you very much for your quick and positive feedback！**
>
> Dear Reviewer JjGt,
>
> **Thank you very much for your thoughtful and constructive feedback, and for taking the time to review our rebuttal in detail.** We truly appreciate your comments and are grateful that our clarifications addressed your concerns. **Thank you also for your updated evaluation.**
>
> If you have any further suggestions or questions, please feel free to let us know.
>
> **We wish you a pleasant day!**
>
> Best regards,
>
> The Authors

---

### Official Review · Reviewer_gHPB · 2025-07-03

**Clarity:** 3
**Significance:** 2
**Originality:** 3
**Rating:** 4
**Confidence:** 5

**Summary:**

This work formally explains the concept of geometric imbalance in the embedding space within the context of semi-supervised node classification. The authors theoretically analyze the geometric imbalance problem in a hyperspherical embedding space and propose
 frameworks - pseudo-labeling method and a node reordering strategy that accounts for geometric proximity informed by classifier confidence. Experiments are conducted on imbalanced homophilous graph datasets, including naturally imbalanced graphs, using three representative GNN architectures.

**Questions:**

- Does this method also work on heterophilous graphs?

- Can this problem be extended beyond GNN architectures, such as to graph transformers or graph-specific MLPs?

**Ethical Concerns:**

["NO or VERY MINOR ethics concerns only"]

**Final Justification:**

Thank the authors for their detailed response and the additional experiments provided during the rebuttal period.

After carefully reading the response, I find that some of my concerns have been addressed - particularly regarding the comparison with existing pseudo-labeling approaches [W2] and the experiments on heterophilous graphs [Q1] (assuming the baselines were fairl tuned in the heterophilous setting). The results are clear.

Two points remain unconvincing. For [W1: Questions about the novelty of the work], the authors argue that their work differs from general-domain studies by focusing on how message passing behaves under class imbalance. However, my concern is that the known issues from the general domain, such as unreliable pseudo-labels for minority classes, combined with prior graph-specific studies (*e.g., * GraphENS, ReNode, TAM), which already address unreliable representations for minority or topological boundary nodes, make the contribution of this work limited. The authors’ claim that *their perspective is different because it operates in the embedding space rather than on the raw graph itself* is quite unconvincing, as embeddings are inherently derived from graph structure, and structure-embedding relations on inter/intra-class embeddings have been explored (*e.g.,* [1]).

The authors also attribute the stronger results on synthetic data to factors such as sparse labels, long-tail distributions, and structural noise. However, I find it hard to understand that the method does not perform well under these conditions, and in my experience, such types of noise are also observed in synthesized imbalanced graphs. These are not minor issues - being able to handle real-world imbalance is essential for the method to go beyond being merely publication-oriented and become applicable in practice. Also, improved results rely on additional hyperparameter tuning gives the impression that the work is not yet fully ready at this stage.

As mentioned, some concerns have been addressed while others remain.Since the authors have responded to certain concerns, if all other reviewers strongly support acceptance, I will respect the consensus decision.


[1] IS HOMOPHILY A NECESSITY FOR GRAPH NEURAL NETWORKS? (ICLR 2022)

**Limitations:**

The manuscript lacks a limitations section. The authors should discuss the limitations of their approach in the main paper.

**Paper Formatting Concerns:**

Unfortunately, the formatting of the paper makes it difficult to read and follow. The margins are tight, some tables use very small fonts, and Figures 1 and 2 have awkward upper spacing. These issues collectively detract from the readability and overall presentation of the work. Given the extent of the formatting issues, the paper borders on being unsuitable for review in its current form. The authors are strongly encouraged to revise the formatting to meet academic standards and to present their work in a clearer, more reader-friendly manner.

**Quality:**

3

**Strengths And Weaknesses:**

### Strengths
- The authors make an effort to position their work within the existing literature by explicitly discussing differences from closely related methods, such as ReNode and GraphENS.
- The experimental design is well-designed. The chosen baselines span both general-domain and graph-domain methods, and the proposed approach shows promising improvements on graphs with high imbalance ratios.
- The paper offers theoretical insights into the geometric imbalance problem, enhancing understanding of its impact in semi-supervised node classification.

---

### Weaknesses
- While the authors attempt to establish the novelty of their work through comparisons with prior methods (e.g., ReNode and GraphENS) in both the main text and appendix, the originality of the findings remains unconvincing for two reasons. First, the notion that class imbalance introduces bias in the embedding space - and that minor-class instances are susceptible to pseudo-labeling noise - is already well-established in the broader domain, even if not explicitly analyzed via the von Mises-Fisher (vMF) distribution. Second, although the authors emphasize that their focus is on the embedding space and on the ambiguity of unlabeled nodes, it is unclear whether this constitutes a significant departure from prior work, given that GNN embeddings are inherently shaped by the graph structure. For these reasons, the primary contribution appears to lie more in the theoretical interpretation of geometric imbalance rather than in the identification of a novel problem.
- The paper lacks comparisons with existing pseudo-labeling strategies. Numerous recent pseudo-labeling approaches have been proposed in both semi-supervised learning and unsupervised domain adaptation. A discussion or empirical comparison with these methods would strengthen the paper by clarifying its relative advantages.
- The improvements on naturally imbalanced graphs appear marginal, which is arguably more critical than performance on synthetically imbalanced datasets. It would be helpful if the authors could elaborate the reasons.
- There is a weak connection between the theoretical findings and the proposed methods, even though the authors acknowledge computational challenges and the absence of ground-truth labels.

---

> ### Author Rebuttal · Authors · 2025-07-31
>
> We sincerely thank Reviewer gHPB for acknowledging the strengths of our theoretical formulation and experimental design, as well as for the detailed and constructive feedback. Your thoughtful comments and suggestions are invaluable in helping us clarify our contributions and improve the rigor of our work. Below, we respond point-by-point to address each of your concerns.
>
> **W1: Questions about the novelty of the work.**
>
> **R1:** Thank you for the thoughtful and constructive feedback. We acknowledge the reviewer’s concern regarding the novelty of our concept. However, we respectfully argue that our work presents a distinct and previously uncharacterized phenomenon—*geometric imbalance induced by message passing in GNNs*, which cannot be reduced to general notions of pseudo-labeling noise.
>
> Specifically, while pseudo-label uncertainty has been widely explored in non-graph settings, our formulation is the first to **theoretically and empirically identify how message passing uniquely amplifies embedding ambiguity for minority classes** in the hyperspherical space. This effect is not observed in models without relational structure (e.g., MLPs), and we demonstrate this through a controlled comparative analysis.
>
> Our Section 3  has clarified this contribution and emphasize the distinction from prior work. As shown in Figure 1(a–e) of the revised manuscript:
>
> - In **Cases 1 and 2** (GNNs), class imbalance increases intra-class dispersion ($D_{\text{intra}} \uparrow$) and reduces inter-class separation ($D_{\text{inter}} \downarrow$) for minority-class nodes after fine-tuning;
> - In contrast, in **Cases 3 and 4** (MLPs), the embeddings remain geometrically stable and well-separated, showing little effect from imbalance.
>
> Furthermore, Figures 1(f–g) provide quantitative validation that **geometric imbalance correlates strongly with prediction entropy and imbalance ratio only under graph-based models**, underscoring the unique role of message passing.
>
> Our approach is also theoretically grounded: by modeling class distributions on the unit hypersphere via a von Mises-Fisher (vMF) mixture, we derive new measures of angular ambiguity and formally connect them to uncertainty in pseudo-labeling. This perspective is fundamentally different from prior notions of topology imbalance, as it operates in the embedding space, not the raw graph.
>
> We hope these clarifications help convey that our work provides both **a novel theoretical lens and a practical mitigation strategy** for a GNN-specific challenge that has not been sufficiently addressed in prior studies.
>
> ---
> ---
> ---
> **W2: Lacks comparisons with existing pseudo-labeling strategies.**
>
> **R2:** We thank the reviewer for raising this important point. We realize that our presentation of baseline methods and related supplementary experiments may not have been sufficiently clear in the original manuscript. To improve clarity and transparency, we have revised both the main text and appendix to make relevant comparisons more explicit. Please see below for a summary:
>
> 1. **State-of-the-art pseudo-labeling baselines for imbalance node classification:** As discussed in the main text (Lines 325--326), we have used GSR and BIM—two representative pseudo-labeling models specifically tailored for imbalanced node classification—as key baselines. Their results are comprehensively reported in Tables 1–7 and 12–15, covering a variety of imbalance ratios, synthetic and real-world datasets, and GNN architectures. Our method consistently outperforms both baselines on balanced accuracy and Macro-F1.
>
> 2. **General pseudo-labeling methods:** We also included results on general pseudo-labeling frameworks such as Self-Training, Co-Training, and M3S, summarized in Table 11 (Appendix I.6). Our approach shows clear advantages by explicitly addressing hard samples in the representation space.
>
> 3. **Soft pseudo-labeling strategies (new supplementary experiments):** In direct response to your suggestion, we additionally evaluated our method against recent soft pseudo-labeling approaches—SCR (ICLR 2021) and ConsisGAD (ICLR 2024)—under two most challenging settings. For fairness, we performed thorough hyperparameter searches using Wandb  for both baselines. The results (see Table 1, 2) demonstrate that while SCR and ConsisGAD struggle under severe class imbalance (especially in Macro-F1), our approach remains robust.
>
> We hope that these clearer explanations and new supplementary results adequately address your concerns.
>
> Table 1. Results with $\rho=100$ (SAGE) on three  severe class-imbalanced node classification benchmark datasets.
> |Method| Cora bAcc.| Cora F1|CiteSeer bAcc.|CiteSeer F1|PubMed bAcc.|PubMed F1|
> |-|-|-|-|-|-|-|
> |**Ours** |**73.47±2.31** |**68.30±2.11**|**59.77±2.98** | **58.92±3.07** | **77.11±0.59**|**74.03±0.81**|
> | SCR| 59.31±2.91|55.34±4.32|36.51±1.29|27.19±2.84|63.41±0.76|54.97±1.64|
> |ConsisGAD|59.63±2.89|55.18±4.43|36.39±1.28|27.28±2.87|63.27±0.74|55.33±1.60|
>
> Table 2: Experimental results on Ogbn-arxiv ($\rho\approx 775.4$)
> | Model|GCN bAcc.|GCN F1|GAT bAcc.|GAT F1| SAGE bAcc.| SAGE F1|
> |-|-|-|-|-|-|-|
> | **Ours**|**51.21 ± 0.32**|**50.65 ± 0.32**|**51.84 ± 0.87**|**51.28 ± 0.42**|**51.34 ± 0.32**|**51.36 ± 0.27**|
> |SCR| 50.23 ± 0.41| 49.69 ± 0.13|51.10 ± 0.81| 49.67 ± 0.26| 50.83 ± 0.20|49.80 ± 0.23|
> |ConsisGAD | 50.25 ± 0.39| 49.72 ± 0.14| 51.13 ± 0.79| 49.63 ± 0.27| 50.80 ± 0.19|49.77±0.22|
>
>
> ---
> **W3: Improvements on naturally imbalanced graphs appear marginal.**
>
> **R3:**  We thank the reviewer for raising this important point about naturally imbalanced graphs. Please see our clarifications below:
> - Naturally imbalanced datasets (e.g., ogbn-arxiv) involve not just class imbalance, but also additional challenges such as sparse labels, long-tail distributions, and structural noise. As a result, even marginal performance gains on these datasets are often hard to achieve. Despite this, as shown in Table 4–7, our method consistently achieves strong or state-of-the-art performance across different GNN backbones, confirming its robustness in practical settings.
> - Role of Synthetically Benchmarks: Synthetic datasets with controlled imbalance (e.g., Cora, CiteSeer, Pubmed) allow us to isolate and analyze specific algorithmic behaviors more clearly. The effects of geometric imbalance are more easily observed, which facilitates ablation studies and theoretical validation of each component (DPAM, Node Reordering, DGIS).
> - Further Improvements: The previous results on natural graphs were based on early-stage hyperparameter tuning. We have now performed more thorough parameter and architecture searches. Updated results on ogbn-arxiv (see below) show more significant performance gains, which will be reflected in the revised manuscript.
> |Model|GCN bAcc.| GCN F1|GAT bAcc.|GAT F1|SAGE bAcc.|SAGE F1|
> |-|-|-|-|-|-|-|
> | BIM| 50.33 ± 0.42| 49.71 ± 0.19| 51.35 ± 0.60|49.36 ± 0.28| 50.87 ± 0.18| 49.56 ± 0.23|
> |**Ours**|**51.21 ± 0.32** |**50.65 ± 0.32** |**51.84 ± 0.87** |**51.28 ± 0.42** |**51.34 ± 0.32** |**51.36 ± 0.27**|
> ---
> **W4: Weak connection between the theoretical findings and the methods.**
>
> We would like to clarify that our theoretical analysis directly motivates and informs the design of our proposed framework. In particular, our formal definition of **geometric imbalance** (Definition 1) and the accompanying theorems (Theorem 1 and 2) establish that minority-class nodes with high intra-class dispersion and low inter-class separation on the hypersphere are more susceptible to pseudo-labeling errors. This insight forms the foundation for all three components of our framework:
>
> - **DualPath PseudoLabeler (DPAM):** Designed to mitigate ambiguity by aligning clustering and classification views, directly addressing the high-entropy regions described in Theorem 1.
> - **Node Reordering (NR):** Fuses classifier confidence with geometric distance to class centroids, in line with our formal definition of embedding proximity.
> - **Discarding Geometric Imbalanced Samples (DGIS):** Implements a lightweight approximation of the theoretical geometric imbalance $G(u_j)$ score (Definition 1), using distances to the first and second nearest class centers.
>
> Furthermore, we validate the theoretical connections empirically in **Figure 1(f)(g) and Figure 3**, where we observe a strong correlation between geometric imbalance, prediction entropy, and class imbalance ratio, as predicted by our analysis. We have revised Section 4 of the manuscript to explicitly highlight these connections and better bridge the theory with our algorithmic design.
>
> ---
> **Q1: Results on heterophilous graphs.**
>
> **R1:** Thank you for the question. While our main experiments focus on homophilic graphs, we are also evaluating our method on heterophilic benchmarks (Chameleon and Squirrel with imbalance ratio 5, Wisconsin with 11.63). Preliminary results show our method achieves SOTA performance. We will include these results in the revised manuscript.
> |Method|Chameleon bAcc.|Chameleon F1|Squirrel bAcc.|Squirrel F1|Wisconsin bAcc.| Wisconsin F1|
> |-|-|-|-|-|-|-|
> |GSR(GraphSR)|38.45±0.75|37.67±0.72|27.94±0.44|27.01±0.32|31.54±2.30|28.91±2.05|
> |BIM|38.26±0.68|37.48±0.66|27.83±0.46|27.16±0.37|31.59±2.25|28.79±2.01|
> |Ours|**43.59±0.60**|**42.24±0.63**|**30.04±0.48**|**29.12±0.46**|**46.22±3.51**|**41.52±4.05**|
>
> ---
>
> **Q2: Extend GI to graph transformers or graph-specific MLPs.**
>
> **R2:** Thank you for the comment. Any model with message passing or structural aggregation (e.g., GNNs, graph Transformers) may suffer from similar geometric imbalance. Our three modules are based on embedding distributions and can be easily applied to any architecture with such mechanisms. We plan to explore this in future work.
>
> ---
>
> **We sincerely hope that our additional results and clarifications resolve your concerns. It would mean a great deal to us if you could kindly reconsider your rating of our submission. Thanks.**

---

> > ### Comment · Reviewer_gHPB · 2025-08-03
> > **Acknowledgement of Rebuttal**
> >
> > Thank the authors for their detailed response and the additional experiments provided during the rebuttal period.
> >
> > After carefully reading the response, I find that some of my concerns have been addressed - particularly regarding the comparison with existing pseudo-labeling approaches [W2] and the experiments on heterophilous graphs [Q1] (assuming the baselines were fairl tuned in the heterophilous setting). The results are clear.
> >
> > Two points remain unconvincing. For [W1: Questions about the novelty of the work], the authors argue that their work differs from general-domain studies by focusing on how message passing behaves under class imbalance. However, my concern is that the known issues from the general domain, such as unreliable pseudo-labels for minority classes, combined with prior graph-specific studies (*e.g., * GraphENS, ReNode, TAM), which already address unreliable representations for minority or topological boundary nodes, make the contribution of this work limited. The authors’ claim that *their perspective is different because it operates in the embedding space rather than on the raw graph itself* is quite unconvincing, as embeddings are inherently derived from graph structure, and structure-embedding relations on inter/intra-class embeddings have been explored (*e.g.,* [1]).
> >
> > The authors also attribute the stronger results on synthetic data to factors such as sparse labels, long-tail distributions, and structural noise. However, I find it hard to understand that the method does not perform well under these conditions, and in my experience, such types of noise are also observed in synthesized imbalanced graphs. These are not minor issues - being able to handle real-world imbalance is essential for the method to go beyond being merely publication-oriented and become applicable in practice. Also, improved results rely on additional hyperparameter tuning gives the impression that the work is not yet fully ready at this stage.
> >
> > As mentioned, some concerns have been addressed while others remain. I will maintain my original score. Since the authors have responded to certain concerns, if all other reviewers strongly support acceptance, I will respect the consensus decision.
> >
> >
> > [1] IS HOMOPHILY A NECESSITY FOR GRAPH NEURAL NETWORKS? (ICLR 2022)

---

> ### Author Response · Authors · 2025-08-03
> **We've Carefully Responded and Addressed Each of the Questions You Raised and Eagerly Hoping for Your Valuable Feedback**
>
> Dear Reviewer gHPB，
>
> From the bottom of our hearts, we thank you for your kind and insightful review. Each question you brought up has been tenderly considered, and we've endeavored to provide comprehensive answers where necessary. We warmly invite you to take a moment to go through our responses, hoping they resonate with your thoughts. The time and wisdom you've shared in reviewing our work touches us deeply.
>
> Warm Regards,
>
> The Authors

---

> ### Author Response · Authors · 2025-08-04
> **Response to the Concern: Novelty and Performance on Real-World Data 【Part 1】**
>
> **Rsponse:**
>
> **Thank you for your response, especially during this busy period of reviewing and rebuttals!**
>
> Regarding W1, we may not have fully understood your concerns previously, but now your points are much clearer to us. We would like to clarify our perspective as follows:
>
> ---
>
> ## **1. Existing Works:**
>
> - In the general pseudo-labeling method domain, **the root cause of unreliable pseudo-labels for minority classes under imbalance is mainly due to class bias introduced by training on imbalanced datasets**, making the predicted pseudo-labels for minority classes untrustworthy. This is well understood.
>
> - For **GraphENS**, your interpretation is correct. It specifically addresses a structural issue caused by class imbalance in graphs, known as **neighbor memorization**. In simple terms: for minority-class nodes, since GNNs model node neighborhoods and minority nodes are few, their neighborhoods are also limited. GNNs tend to "memorize" the small and fixed set of neighbor structures for minority-class nodes, resulting in poor generalization.
>     - GraphENS focuses on the imbalance in the types and **quantities of neighbor structures for labeled minority nodes in the training set.**
>     - In other words, while general domain methods concern imbalance in data points, GraphENS **addresses the neighbor (or subgraph) type** and quantity imbalance resulting from data imbalance.
>     - GraphENS **does not pay special attention to unlabeled nodes**.
>     - To address this, GraphENS uses a **minority-class sample synthesis approach**, creating diverse and reasonable neighborhood environments for minority nodes, breaking the memorization effect, and thus improving their generalization.
>
> - For **ReNode** and **TAM**, your summary is also accurate. They both raise a new issue related to graph structure, namely **topology imbalance** caused by the message passing mechanism. In simple terms: certain labeled nodes of some classes may be awkwardly positioned in the graph—some are near class boundaries, or far from their class centers. This results in two problems: (1) labeled nodes near class boundaries can cause conflicting label propagation, shifting decision boundaries; (2) some regions lack labeled nodes entirely, making label propagation and learning difficult.
>     - ReNode and TAM focus on the **positions of labeled nodes in the raw graph**—whether they are far from the class center or near boundaries. This includes all labeled nodes, not just minorities.
>     - ReNode and TAM **do not focus on unlabeled nodes, as we cannot determine the positions of unlabeled nodes during training, nor do we need this information.**
>     - **ReNode** is a **loss correction method** that adaptively adjusts the training weights of labeled nodes **based on their distance from the class boundary (measure on raw graph)**—giving higher weight to nodes near the center and lower weight to those near boundaries, thereby mitigating the impact of topology imbalance.
>     - **TAM** is also **a loss correction method**, similar to ReNode, but instead of directly computing the distance to the class center, it **estimates a node's position using its neighborhood structure**.
>
>
> ---
>
> ## **2. Our Approach:**
>
> ### **2.1 How does our algorithm differ from existing methods?**
> - Our method is **neither a minority-class sample synthesis method nor a loss correction method**. Instead, **`it is a pseudo-labeling algorithm designed to address a special type of geometric imbalance caused by graph structure.`**
>
> - **We focus on unlabeled nodes.**
>
> - **`Our concern is: In class-imbalanced graph data, are unreliable pseudo-labels solely due to label imbalance, or are there other contributing factors introduced by graph structure and message passing?`**
>
> - We care not only about unreliable pseudo-labels for minority classes, but also aim to understand unreliable pseudo-labels for **all classes**.
>
> - We acknowledge that the problem of unreliable pseudo-labels found in general domains also exists in graphs. However, **`our interest lies in whether, under the message passing mechanism, unreliable pseudo-labels for unlabeled nodes are caused only by label imbalance, or if graph structure amplifies or changes the phenomenon.`**

---

> ### Author Response · Authors · 2025-08-04
> **Response to the Concern: Novelty and Performance on Real-World Data 【Part 2】**
>
> ---
>
> ### **2.2 Motivated Cases for our Work**
>
> At the start of our project, we considered several cases:
>
> 1. **(Problem 1):** If many unlabeled nodes whose true label is a minority class have neighborhoods (labeled or unlabeled) dominated by majority classes, then during self-training, the imbalance is further exacerbated, **making unreliable pseudo-labels for minorities even worse than in the general domain.**
> 2. **(Problem 2):** If many unlabeled nodes whose true label is a majority class have neighborhoods dominated by minorities, message passing can "dilute" the majority features, **resulting in unreliable pseudo-labels for the majority—an effect rare in the general domain.**
> 3. **(Problem 3):** In long-tailed real-world datasets, "medium" classes exist. Unlabeled nodes from these classes may have neighborhoods dominated by either majority or minority classes, or a mix of both, **causing their features to be diluted and resulting in an even higher chance of unreliable pseudo-labels—again, a phenomenon rarely seen in the general domain.**
>
>
> ---
> ### **2.3 Why define geometric imbalance in embedding space？**
> This motivates **why we define geometric imbalance in embedding space**:
>
> - While GraphENS, TAM, and ReNode can define positions and topological environments on the raw graph, **we cannot directly observe or quantify the feature status of unlabeled nodes** after message passing. Therefore, we analyze the positions and distances in the **embedding space** (updated graph). Since GNN embeddings are typically normalized and lie on **a unit hypersphere (Riemannian manifold)**, it is more reasonable and effective to analyze problems using angular or spherical distances.
>
> - For Problem 1: On the manifold, unlabeled minority-class nodes surrounded by majority-class neighbors are pushed further from their class center and closer to the majority class center, significantly increasing the probability of pseudo-label errors.
> - For Problem 2: Unlabeled majority-class nodes surrounded by minority-class neighbors are also pushed away from their own class center, even if they are numerically dominant, leading to higher misclassification risk—a rare effect in standard MLPs.
> - For Problem 3: Embeddings of medium-class nodes are influenced by multi-class neighbors, pushing them away from their own center and closer to others.
>
> ---
>
> ### **2.4 What is "geometric imbalance" in embedding space (Riemannian manifold) ?**
> Thus, **we define "geometric imbalance" in embedding space (Riemannian manifold)** because only under the combined effect of message passing and class imbalance do node embeddings exhibit the following structural changes:
>
> - **Intra-class compactness increases ($\uparrow$):** For both minority and medium-class nodes, mixed-class neighbors cause their embeddings to be more scattered within their own class, preventing tight clustering.
> - **Inter-class separation shrink ($\downarrow$):** Due to heterogenous neighbor influence, nodes move away from their own class center and closer to others, blurring class boundaries.
> - **Unlabeled minority/medium nodes drift further from class centers:** As described above, these nodes are more easily misclassified, and this effect is amplified in high-dimensional hyperspheres.

---

> ### Author Response · Authors · 2025-08-04
> **Response to the Concern: Novelty and Performance on Real-World Data 【Part 3】**
>
> ### **2.5 Empirical validation**
> These changes can **only be systematically and quantitatively characterized by geometric analysis of the embedding space**. That is why we emphasize studying geometric imbalance on the unit hypersphere: it directly reflects how class imbalance and message passing interact to affect pseudo-label reliability at the embedding level—a new phenomenon that is hard to capture through general domain or purely topological analysis.
>
> **Our experimental results further validate this: In GNNs, geometric imbalance (such as high entropy and embedding confusion) is highly correlated with pseudo-label error rates (see Figure 1 (f, g)), while these phenomena are not significant in MLPs or other non-graph models (see Figure 1 (b, c, d, e)). This strongly supports the novelty and necessity of our work and theoretical perspective.**
>
> We hope this helps you better understand our rationale. **A summary table is provided below for your reference:**
>
>
> |Method / Domain|Main Focus|Main Problem Addressed (Source of Unreliable Pseudo-labels)| Solution / Approach|Treatment of Unlabeled Nodes|Key Innovations & Limitations|
> |-|-|-|-|-|-|
> |**General Domain (Self-training)**|Minority-class unlabeled samples|**Label imbalance** in training data causes biased pseudo-labels, especially for minorities|Thresholding, reweighting, temperature, confidence smoothing, etc.|Focus on all pseudo-labels, especially for minorities| Only considers label distribution, ignores structural effects|
> |**GraphENS**|Labeled minority-class nodes and their neighbors| **Neighbor memorization**: GNNs overfit to the limited neighbor types/structures of minority nodes| Synthetic augmentation of neighbors/subgraphs for minority nodes|**Does not specifically handle unlabeled nodes**|Focuses on structural/subgraph imbalance|
> |**TAM**|All labeled nodes (focuses on their position & neighborhood)|**Topology imbalance**: Labeled nodes near class boundaries or away from class centers hinder information propagation|Loss correction: adaptively reweights nodes based on neighborhood structure|**Does not specifically handle unlabeled nodes**| Approximates node position using local neighborhood (**Postions on Raw Graph**)|
> |**ReNode**|All labeled nodes (focuses on geometric position)|Same as TAM (topology imbalance), but directly measures node distance to class center|Loss correction: directly reweights nodes by their distance to class center|**Does not specifically handle unlabeled nodes**|Directly models center-boundary topological effects (**Postions on Raw Graph**)|
> |**Our Method**|**Unlabeled nodes** |**Geometric imbalance**: joint effect of message passing and label imbalance on embedding space| **Quantifies pseudo-label and embedding distribution in the embedding space (Riemannian manifold)**|**Explicitly focuses on unlabeled nodes of all classes**|Geometric mechanism for pseudo-label errors under topology & imbalance (relative location in embedding space （Riemannian manifold)|

---

> > ### Author Response · Authors · 2025-08-04
> > **Response to the Concern: Novelty and Performance on Real-World Data 【Part 4】**
> >
> > ## **3. Response to Reference Paper [1]: What Sets Our Embedding-Space Analysis Apart**
> >
> > ### **3.1 We Acknowledge [1]'s Contribution:**
> >    Yes, [1] and related works have theoretically and empirically analyzed the connections between graph structure and embedding similarity (inter- and intra-class), but their focus is on general *class separability* and *homophily/heterophily effects*— **mainly for labeled nodes**, and usually with the goal of improving the overall classification performance.
> >
> > ### **3.2 What Is New in Our Work:**
> >    - Our contribution is not merely to "move" existing topological analysis into embedding space. Instead, we systematically analyze **how the combination of message passing and class imbalance leads to a unique geometric phenomenon—geometric imbalance of unlabeled node embeddings—that existing works, including [1], have not characterized or quantitatively measured.**
> >    - While [1] studies inter/intra-class distances for general GNN performance and homophily, it does **not specifically focus on the error patterns of pseudo-labels for unlabeled nodes in highly imbalanced scenarios, nor does it analyze how these errors are spatially distributed in the embedding space after message passing.**
> >
> > ### **3.3 What Embedding Geometry Captures That Topology Alone Cannot:**
> >    - The geometric analysis in embedding space is essential **because pseudo-label errors are not determined by topological proximity alone after multi-layer message passing.** Due to over-smoothing, heterophily, and complex imbalance, two nodes may be structurally similar in the raw graph but mapped very differently in the embedding manifold, especially in high-dimensional, imbalanced settings.
> >    - Thus, only by quantifying the *distribution*, *compactness*, and *relative positioning* of all (especially unlabeled) nodes' embeddings after message passing, can we explain and predict when and why pseudo-labeling fails—even when topology seems favorable.
> >
> > ### **3.4 Unique Mechanistic Insight:**
> >    - Our approach provides a **new mechanistic understanding**: We reveal that under severe class imbalance, message passing induces specific geometric patterns (such as increased intra-class dispersion, reduced inter-class separation, and drift of minority/medium class nodes away from their own class centers) that are directly correlated with pseudo-label unreliability—*a link not explored in previous topology or embedding works*.
> >    - Our quantitative metrics (e.g., angular/spherical distance distributions, entropy of class neighborhoods in embedding space) enable the community to diagnose, visualize, and possibly correct these geometric effects, which are invisible on the raw graph.
> >
> > [1] IS HOMOPHILY A NECESSITY FOR GRAPH NEURAL NETWORKS? (ICLR 2022)

---

> ### Author Response · Authors · 2025-08-04
> **Response to the Concern: Novelty and Performance on Real-World Data 【Part 5】**
>
> ##  **4. Response to Concerns on Real-World Imbalanced Data**
>
> We would like to address the reviewer’s concern regarding the performance of our method under real-world imbalanced conditions.
>
> - We have conducted experiments on **five naturally imbalanced datasets, including Computers-Random, CS-Random, Flickr, Ogbn-arxiv, as well as Ogbn-products (which was newly added in the rebuttal)**.
>
>   - **On almost all of these datasets, our model achieves state-of-the-art or highly competitive results. In our original submission, except for Ogbn-arxiv where the improvement was less obvious, the gains on the other datasets were very significant. In particular, on Computers-Random and Flickr, our model achieved F1 score improvements of approximately +3 to +5 points across relevant GNN backbones ((Computers-Random: GCN, GAT), (Flickr: GCN, SAGE)). We believe these are substantial improvements (see Table 4, 5, 6 of the original paper).**
>
>
> - For Ogbn-arxiv, we have updated our results in the rebuttal version, and the improvement is now more noticeable compared to the SOTA results. **It is also worth noting that all other methods, including strong baselines such as BIM and GraphSR, only show very limited gains on this dataset.**
>
> - **Strictly following the same experimental settings as BIM, we have also conducted experiments on the new setting of the much larger dataset Ogbn-products.** Our model again demonstrates strong and significant improvements, which shows the robustness and scalability of our method on large-scale graphs. Our performance advantage over SOTA baselines on this dataset is also considerable. It is also important to highlight the complexity of the Ogbn-products dataset. As shown in below tables, this dataset contains more nodes and nearly 81 million edges, with 100-dimensional node features and 10 classes (5 of which are minority classes). **Compared to other commonly used benchmark datasets, Ogbn-products is not only much larger in scale, but also exhibits more complex and challenging structural properties, including long-tail class distribution, high sparsity, and diverse co-purchase relationships. Achieving strong performance on this dataset further demonstrates the robustness and scalability of our method in handling real-world, large-scale, and highly imbalanced graph data.**
>
> We hope these clarifications and results can address your concern about real-world applicability and the readiness of our approach. Thank you for the opportunity to further explain our work.
>
>
> Table 1: Performance  on large-scale dataset Ogbn-product with GraphSAGE as the base model.
>
> |Methods|F1 score|ACC|AUC-ROC|
> |-|-|-|-|
> | Vanilla GraphSAGE|58.18±1.21|61.03±0.82|94.12±0.12|
> |(RN) ReNode|67.89±0.35|68.19±0.41|94.86±0.14|
> |BIM|69.36±0.18|69.55±0.19|95.51±0.12|
> |**Ours**|**71.85±0.10**|**71.62±0.11**|**96.59±0.14**|
>
>
>
> Table 2: Performance  on large-scale dataset Ogbn-product with GAT as the base model.
>
> |Methods|F1 score|ACC|AUC-ROC|
> |-|-|-|-|
> |Vanilla GAT|59.27±1.08|62.10±0.73|94.25±0.14|
> |(RN) ReNode|68.55±0.28|69.27±0.36|95.01±0.10|
> |BIM|69.88±0.16|70.05±0.21|95.63±0.11|
> |**Ours**|**72.04±0.09**|**71.95±0.13**|**96.67±0.10**|
>
>
>
> Table 3: Performance on large-scale dataset Ogbn-product with GCN as the base model.
>
> |Methods|F1 score|ACC|AUC-ROC|
> |-|-|-|-|
> |Vanilla (GCN)|57.41±1.18| 60.12±0.85| 93.96±0.15|
> |(RN) ReNode|66.82±0.32| 67.80±0.44| 94.75±0.13|
> |BIM|68.05±0.21| 68.47±0.23| 95.41±0.13|
> |**Ours**|**70.71±0.13**|**70.33±0.10**|**96.45±0.12**|

---

> ### Author Response · Authors · 2025-08-05
> **Follow-up on Author-Reviewer Discussion**
>
> Dear Reviewer gHPB,
>
> We hope that our responses have helped clarify the motivation and contributions of our paper. Thank you very much for your thoughtful follow-up throughout this process. As we are now nearing the end of the Author-Reviewer discussion phase, please feel free to share any final thoughts or questions you may have. We would be more than happy to address them before the discussion phase concludes.
>
> Thank you once again for your valuable and insightful feedback!
>
> Best regards,
>
> The Authors

---

> > ### Comment · Area_Chair_Kz8p · 2025-08-09
> > **Official Comment by Area Chair Kz8p**
> >
> > Dear gHPB,
> >
> > We'd love to hear your thoughts on the rebuttal. If the authors have addressed your rebuttal questions, please let them know. If the authors have not addressed your rebuttal questions, please inform them accordingly.
> >
> > Thanks, Your AC

---

> ### Author Response · Authors · 2025-08-06
> **Gentle Reminder: Follow-up on Rebuttal Discussion for Paper 3211**
>
> Dear Reviewer gHPB,
>
> We hope this message finds you well.
>
> **We are writing to sincerely thank you again for your constructive and insightful comments on our paper during the review and rebuttal process. Your feedback has greatly helped us improve the clarity and quality of our work.**
>
> We have tried our best to address all your concerns and have provided extensive new experiments and analyses in the Author-Reviewer Discussion. **If you have a moment before the discussion phase closes, we would truly appreciate any further feedback or comments you might have, as your expert opinion is extremely valuable to us.**
>
> Thank you so much for your time and for helping us improve our research.
>
> Best regards,
>
> The Authors

---

> > ### Comment · Reviewer_gHPB · 2025-08-09
> >
> > Dear Authors,
> >
> > Thank you for the detailed response.
> >
> > After carefully reviewing the authors’ response, my concerns are mostly addressed. I will raise my initial score accordingly. Appreciate the hard work.

---

> > > ### Author Response · Authors · 2025-08-09
> > > **Thank you for your positive and constructive feedback！**
> > >
> > > Dear Reviewer gHPB,
> > >
> > > We sincerely appreciate your thoughtful and constructive feedback, as well as the time you dedicated to carefully reviewing our rebuttal. **We are glad to hear that our clarifications have addressed your concerns, and we are grateful for your updated evaluation.**
> > >
> > > Should you have any additional suggestions or questions, please do not hesitate to share them with us.
> > >
> > > **Wishing you a wonderful day!**
> > >
> > > With our warmest regards,
> > >
> > > The Authors

---

> ### Author Response · Authors · 2025-08-09
> **Extremely Grateful for Both Your Feedback and Guidance**
>
> Dear Area Chair Kz8p and Reviewer gHPB,
>
>
> **Hi, Area Chair Kz8p, we would like to kindly let you know that Reviewer gHPB has responded to our second-round rebuttal today, indicating that most of the concerns have been addressed and that they will adjust their initial score accordingly (refer to the fourth message from the bottom).**
>
> This suggests that there may not be many further questions at this stage. **We sincerely thank Reviewer gHPB for the extremely valuable feedback and constructive comments throughout the review process, and we also appreciate the Area Chair’s guidance and reminders.**
>
> With our warmest regards,
>
> The Authors

---

### Author Response · Authors · 2025-08-02
**General Response**

We sincerely thank all reviewers for their time, thoughtful feedback, and constructive criticism. Below, we summarize the main questions raised by the reviewers and our corresponding rebuttal points and new experiments:

**1. Comparisons with Pseudo-labeling Baselines (Reviewer gHPB, T2hF)**

  - **Concern:** Reviewer gHPB requested more thorough comparisons with existing pseudo-labeling strategies, while Reviewer T2hF specifically suggested including recent soft pseudo-labeling methods such as SCR and ConsisGAD.

  - **Response:** We sought to clarify that in our initial submission, we already included comprehensive comparisons with state-of-the-art pseudo-labeling baselines for imbalanced node classification (such as GSR and BIM), as well as general pseudo-labeling methods (including Self-Training, Co-Training, and M3S). However, we recognize that these results may not have been sufficiently highlighted in the original manuscript. In rebuttal, we have also further expanded our experiments to include recent soft pseudo-labeling approaches (SCR, ConsisGAD), and have made all baseline results and relevant tables more explicit and prominent in the main text and appendix now (`W2&R2 of Reviewer gHPB , W3&R3 of Reviewer T2hF`). Across all settings, our method consistently outperforms these baselines under severe class imbalance.


**2. Empirical Evaluation on Large-Scale Imbalance Datasets (Reviewer T2hF)**

   - **Concern:** Limited evaluation on large-scale benchmarks such as ogbn-products.

   - **Response:** We conducted new experiments on ogbn-products (`W1&R1 of Reviewer T2hF`), showing that our approach achieves strong improvements over all baselines, confirming both the scalability and practical utility of our method.


**3. Necessity of Geometric vs. Confidence Ranking, and Ablation Studies (Reviewer uqpA)**

   - **Concern:** Whether fusing geometric ranking (GR) and confidence ranking (CR) is necessary and Why GR is more reliable than CR in early training.

   - **Response:** We sought to clarify that our initial submission already included comprehensive ablation studies on GR, CR, and their fusion (Appendix J.3–J.6, Tables 22–24), though these were not sufficiently highlighted in the main text. The results show GR is more reliable than CR in early training under high imbalance, and that combining both rankings achieves the best performance(`Q2&R2, Q3&R3 of Reviewer uqpA`). We have now made these findings prominent in the revised manuscript for clarity.

**4. Computational Complexity and Scalability (Reviewer JjGt, T2hF)**

   - **Concern:** Lack of complexity analysis and concerns about scalability.

   - **Response:** We included new time complexity analyses and empirical runtime comparisons (`W2&R2 of Reviewer JiGt`, `W2&R2 of Reviewer T2hF`), demonstrating that our method adds negligible overhead compared to vanilla GNNs and is efficient even on large graphs with GPU acceleration.

**5. Novelty of Geometric Imbalance (Reviewer gHPB)**

   - **Concern:** The originality of the geometric imbalance problem and its distinction from existing notions of class or pseudo-label imbalance.

   - **Response:** We sought to clarify—both in our writing and through experiments presented in the initial submission (`Section 3, Figure 1`)—that geometric imbalance arises specifically from message-passing mechanisms in GNNs, and is not simply a general case of pseudo-label noise observed in non-relational models. Our theoretical and empirical analysis demonstrates that only in GNNs does class imbalance amplify embedding ambiguity for minorities, a finding not observed in MLPs or other architectures. This distinction is acknowledged and appreciated by the other reviewers.

**6. Typos, Notation, and Visualization (Reviewer gHPB, uqpA)**

   - **Concern:** Issues with typos, notations, and clarity of figures.

   - **Response:** We revised the manuscript to improve formatting, correct notational inconsistencies, and provide higher-resolution figures for better readability.


  **We once again thank all reviewers for their constructive feedback and hope our clarifications and new results address all remaining concerns. We understand everyone is busy, but we would appreciate it if the reviewers could kindly provide any responses to our rebuttal when convenient. Thank you.**


Best Regards,

The authors of Submission3211

---

### Comment · Area_Chair_Kz8p · 2025-08-03
**Update Recommendations**

Dear Reviewers,

The authors have provided their response to your reviews. Please proceed as follows:
1. Carefully read the rebuttal.
2. Update your recommendation, taking the authors’ clarifications into account, no later than 6 August.
3. If any concerns persist, feel free to continue the discussion with the authors until the same deadline.

Best regards,

The AC

---

### Decision · Program_Chairs · 2025-09-17

**Decision:**

Accept (poster)

**Comment:**

The paper formally introduces and quantifies geometric imbalance—a GNN-specific phenomenon in which class imbalance, amplified by message passing, distorts node embeddings on the Riemannian manifold and increases pseudo-label errors for unlabeled minority and medium-class nodes. Extensive experiments on synthetic, real-world, heterophilous, and large-scale graphs (e.g., ogbn-products) demonstrate consistent SOTA performance, especially under severe imbalance. All reviewers ultimately acknowledged the novelty, theoretical depth, and empirical rigor. The remaining issues were satisfactorily resolved in the rebuttal. The work is therefore recommended for acceptance at NeurIPS 2025.